# CAUSAL ORDER: THE KEY TO LEVERAGING IMPERFECT EXPERTS IN CAUSAL INFERENCE

**Aniket Vashishtha**[1][‡] **Abbavaram Gowtham Reddy**[2][§] **Abhinav Kumar**[3],
**Saketh Bachu**[4]**, Vineeth N Balasubramanian**[5][¶]**Amit Sharma**[5]
[1]UIUC, [2]CISPA Helmholtz Center for Information Security, Germany,
[3]MIT, [4]IIT Hyderabad, India, [5]Microsoft Research, India
`aniketv2@illinois.edu, gowtham.abbavaram@cispa.de,`
`akumar03@mit.edu, sakethvnit@gmail.com,`
`vineeth.nb@microsoft.com, amshar@microsoft.com`

## ABSTRACT

Large Language Models (LLMs) have been used as *experts* to infer causal graphs, often by repeatedly applying a pairwise prompt that asks about the causal relationship of each variable pair. However, such experts, including human domain experts, cannot distinguish between direct and indirect effects given a pairwise prompt. Therefore, instead of the graph, we propose that *causal order* be used as a more stable output interface for utilizing expert knowledge. Even when querying a *perfect* expert with a pairwise prompt, we show that the inferred graph can have significant errors whereas the causal order is always correct. In practice, however, LLMs are imperfect experts and we find that pairwise prompts lead to multiple cycles. Hence, we propose the triplet method, a novel querying strategy that introduces an auxiliary variable for every variable pair and instructs the LLM to avoid cycles within this triplet. It then uses a voting-based ensemble method that results in higher accuracy and fewer cycles while ensuring cost efficiency. Across multiple real-world graphs, such a *triplet*-based method yields a more accurate order than the pairwise prompt, using both LLMs and human annotators. The triplet method enhances robustness by repeatedly querying an expert with different auxiliary variables, enabling smaller models like Phi-3 and Llama-3 8B to surpass GPT-4 with pairwise prompting. For practical usage, we show how the expert-provided causal order from the triplet method can be used to reduce error in downstream graph discovery and effect inference tasks.

## 1 INTRODUCTION

Based on evidence that LLMs' domain knowledge, even if imperfect, can be used to decide the direction of causal relationship between a pair of variables (Kıcıman et al., 2023; Willig et al., 2022), recent years have seen the use of LLMs for inferring the entire causal graph for a given problem domain. This is done by typically invoking a pairwise prompt—of the form: "*does variable A cause variable B?*"—multiple times for different pairs of variables (Long et al., 2022; Antonucci et al., 2023; Kıcıman et al., 2023; Cohrs et al., 2023). In other related efforts, causal graphs or edges obtained from LLMs are used as a prior (Takayama et al., 2024) or constraint (Long et al., 2023; Khatibi et al., 2024; Ban et al., 2023a) for causal discovery algorithms, showing that LLM-derived graphs enhance downstream graph discovery accuracy.

However, we highlight a key limitation of using graphs as the *output interface* for such domain knowledge inferred from LLMs, or for that matter, even other imperfect experts (e.g., humans). Obtaining the complete graph requires distinguishing between direct and indirect effects among variables. Given only a pair of variables, it is not possible to decide whether an edge exists or is

---

[‡]Work primarily done as a Research Fellow at MSR India, with additional contributions made at UIUC.

[§]Work primarily done at IIT-Hyderabad, with additional contributions made at CISPA.

[¶]Work primarily done at IIT Hyderabad, with additional contributions made at MSR India.

Code: `https://github.com/AniketVashishtha/Causal_Order_Imperfect_Experts`.

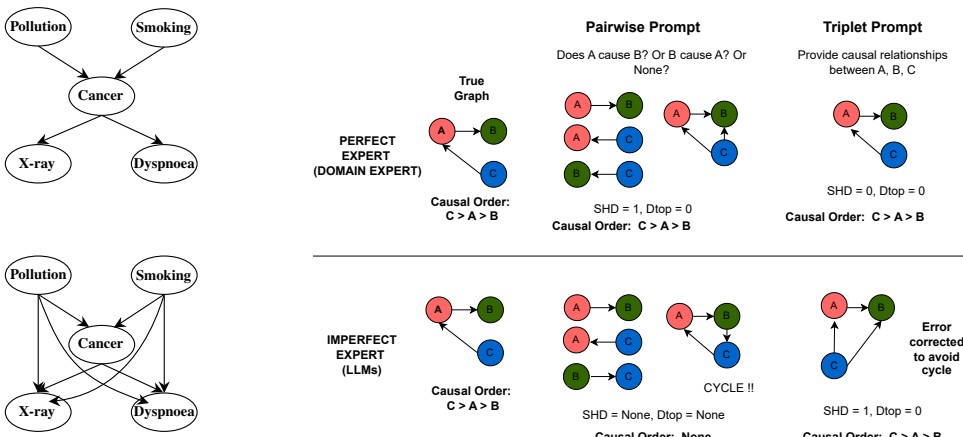

Figure 1: **Cancer dataset** (Scutari & Denis, 2014): **Top:** True causal graph. **Bottom:** Expert-estimated causal graph. Note that the latter, while not correct wrt. the true graph, yields the correct causal order.

Figure 2: **Top:** Using the pairwise prompt, even under a perfect expert (e.g., domain expert), the estimated graph may not be correct ($SHD = 1$). Causal order, however, is correct ($D_{top} = 0$) and hence a better metric. **Bottom**: under imperfect experts such as LLMs, pairwise prompts may not lead to valid order, creating cycles. The proposed triplet prompting strategy alleviates this issue to provide better estimates of causal order ($D_{top} = 0$).

mediated by another variable, even for a *perfect* human expert—the existence of an edge depends on which other variables are considered to be a part of the node set in the query. For example, consider the true data-generating process, *Smoking → Lung Damage → Respiratory Diseases*. If an expert is asked whether there should be a direct causal edge from *Smoking* to *Respiratory Diseases*, they would answer "*Yes*", which may not capture the true process. However, if they are told that the set of observed variables additionally includes *Lung Damage*, then the correct answer would be to not create a direct edge between *Smoking* and *Respiratory Diseases*, but rather create edges mediated through *Lung Damage*. In large graphs, keeping track of the different variables that can affect a given pairwise decision can be cumbersome.

As another example showing the subjectivity of deciding direct or indirect edges, consider the scenario in Fig 1 with the variables: *Pollution Exposure, Cancer, Dyspnoea, Smoking History* and *Positive X-ray*. When queried only for the presence of a causal edge from *Pollution* to *Dyspnoea* (shortness of breath), an expert may answer *"Yes"*. However, if one has to provide a complete graph, it may be non-trivial for an expert to decisively agree on adding a direct edge from *Pollution* to *Dyspnoea*, creating edges mediating through *Cancer*, or both.

**Causal Order: Significance and Utility.** We instead propose *causal order* as a more stable approach to obtain experts' domain knowledge. *Causal order* is defined as the topological ordering over graph variables. Since the causal order does not distinguish between direct and indirect effects, in both examples above, the causal order is unique and unambiguous. In the first example, *Smoking ≺ Respiratory Diseases* is a valid causal order ($a \prec b$ indicates that $a$ occurs before $b$ in a casual process). Similarly, in the second example, the causal order, *Pollution ≺ Dyspnoea* holds true in all three cases considered above by the expert. Formally, we show that for an (optimal) perfect expert that is given only a pair of variables at a time, the predicted causal graph can be incorrect but the predicted causal order is always correct. As a result, the standard practice of obtaining a causal graph from LLMs and crowd-sourced annotators (using pairwise questions) may introduce errors in inferred edges, which can be mitigated by using causal order in downstream causal algorithms. Order is a stable causal construct, independent of other variables present in the query, making it more generalizable. Though simpler than a full graph, it aids tasks like effect inference and graph discovery. We show that the correct causal order is sufficient for identifying a valid backdoor set for any pair of treatment and outcome variables. Moreover, a causal order-based metric, topological divergence ($D_{top}$), correlates better with effect estimation accuracy than commonly used graph metrics such as structural hamming distance (*SHD*). Specifically, $D_{top} = 0$ if and only if the causal order provides a valid backdoor adjustment set. Causal order enhances effect inference and improves graph discovery. We provide simple algorithms to integrate it into existing causal discovery methods.

**Causal Order: Eliciting from Experts.** In practice, obtaining causal order from experts is still a challenge because we need to account for *imperfect experts* such as human annotators and LLMs. Using the standard method (Kıcıman et al., 2023; Long et al., 2022) of iterating with a pairwise prompt/question over a set of variables, while a perfect expert would always predict the correct causal order, we find that using LLMs as experts leads to many cycles. To reduce the number of cycles from LLM output, we propose a novel *triplet* method for obtaining causal order (see Figure 2). Rather than asking questions about a pair of variables, the triplet prompt asks about the causal relationship between the pair and an auxiliary variable, and instructs the LLM to obey acyclicity for the triplet. We theoretically show that given an imperfect expert with an error $\epsilon$ on each prediction, using the triplet-based prompt results in an error less than $\epsilon$, which is less than the error of the pairwise prompt. Moreover, since each variable pair occurs in more than one triplet, the repetition allows for ensembling strategies for a more reliable order. Using human annotators and LLMs as imperfect experts, the triplet method provides more accurate causal order than the pairwise prompt, especially in large graphs. It also enables small models like Phi-3 and Llama-3 8B to outperform GPT-4's pairwise prompt.

## 2 RELATED WORK

**Domain Expertise-aided Causal Discovery.** Prior knowledge has been used in causal discovery literature (Hasan & Gani, 2022; Constantinou et al., 2023; Heckerman & Geiger, 2013; Teshima & Sugiyama, 2021; O'Donnell et al., 2006; Wallace et al., 1996). These methods rely on prior knowledge such as domain experts' opinions and documented knowledge from randomized controlled trials. Various priors have been studied in literature, such as *edge existence*, *forbidden edge* (Meek, 1995), and *ancestral constraints* (Constantinou et al., 2023; Ban et al., 2023b). Recent advances in LLMs have led to more attention on how LLMs may act as imperfect experts and provide causal knowledge based on metadata such as variable names (Kıcıman et al., 2023; Ban et al., 2023b; Long et al., 2023; Willig et al., 2022). Early methods (Kıcıman et al., 2023; Willig et al., 2022; Long et al., 2022) rely on LLMs to predict the complete causal structure, which is evaluated using metrics for full graph structure such as Structural Hamming Distance (SHD). Recent methods however use LLM's output to improve accuracy of graph discovery algorithms. The key idea is that LLM can provide information about edges in the graph, which can then be added as a prior or constraint (Long et al., 2023; **?**) to improve the accuracy for a causal discovery algorithm. For example, (Long et al., 2023) use LLMs to improve output of a constraint-based algorithm for full graph discovery by orienting undirected edges in the CPDAG. Most of these works, however, depend on obtaining correct edge information from LLMs and evaluate LLMs' quality by full graph metrics (Naik et al., 2023; Zhang et al., 2024) such as SHD (Kıcıman et al., 2023; Long et al., 2023). We observe that imperfect experts (LLMs or humans) cannot reliably provide edge information given a pair (or subset) of variables. Causal order may be a more appropriate causal structure to elicit from experts. For the same reason, the quality of an imperfect expert's output for such tasks is better evaluated on the accuracy of causal order, rather than the full graph structure.

**LLM-based Prompting Strategies.** Existing LLM-based algorithms for graph discovery (Kıcıman et al., 2023; Long et al., 2022; Ban et al., 2023b; Antonucci et al., 2023) use a pairwise prompt, essentially asking "*does A cause B?*" with varying levels of prompt complexity. Going beyond this line of work, we propose a triplet-based prompt that provides more accurate answers through aggregation and provides an uncertainty score for each edge to aid in cycle removal. As a result, our triplet-based prompt may be of independent interest for causal tasks.

## 3 CAUSAL ORDER: A STABLE INTERFACE FOR EXPERTS' KNOWLEDGE

**Preliminaries.** Let $\mathcal{G}(\mathbf{X}, \mathbf{E})$ be a causal directed acyclic graph (DAG) consisting of a set of variables $\mathbf{X} = \{X_1, \ldots, X_n\}$ and a set of directed edges $\mathbf{E}$ among variables in $\mathbf{X}$. A directed edge $X_i \rightarrow X_j \in \mathbf{E}$ denotes the *direct* causal influence of $X_i$ on $X_j$. Let $pa(X_i) = \{X_k | X_k \rightarrow X_i\}$, $de(X_i) = \{X_k | X_k \leftarrow \cdots \leftarrow X_i\}$, $ch(X_i) = \{X_k | X_i \rightarrow X_k\}$ denote the set of *parents*, *descendants* and *children* of $X_i$ respectively. If a variable $X_k$ is a descendant of $X_i$ (but they are not connected by a direct edge), then $X_i$ is said to have an *indirect* effect on $X_k$. Average causal effect (Pearl, 2009) (ACE) of a variable $X_i$ on a variable $X_j$ is defined as: $ACE_{X_i}^{X_j} = \mathbb{E}[X_j | do(X_i = x_i)] - \mathbb{E}[X_j | do(X_i = x_i^*)]$, where $X_i$ is called the *treatment*, $X_j$ is called the *target*, and $do(X_i = x_i)$ denotes an external intervention to the variable $X_i$ with the value $x_i$. If a set of variables $\mathbf{Z}$ satisfies the backdoor criterion (Defn. B.1) relative to $(X_i, X_j)$,

$\mathbb{E}[X_j|do(X_i = x_i)]$ can be computed as: $\mathbb{E}[X_j|do(X_i = x_i)] = \mathbb{E}_{\mathbf{z}\sim\mathbf{Z}}\mathbb{E}[X_j|X_i = x_i, \mathbf{Z} = \mathbf{z}]$ (Thm. 3.3.2 of Pearl (2009)); and $\mathbf{Z}$ is called a valid *adjustment set*. We now define the causal (topological) order and the topological divergence metric (Rolland et al., 2022) that measures the goodness of a given causal order wrt. the ground-truth graph.

**Definition 3.1.** *Topological Order. Given a causal graph $\mathcal{G}(\mathbf{X}, \mathbf{E})$, a sequence (or ordered permutation) $\pi$ of variables $\mathbf{X}$ is a topological order iff for each edge $X_i \rightarrow X_j \in \mathbf{E}$, $\pi_i < \pi_j$.*

**Definition 3.2.** *The topological divergence of an estimated order $\hat{\pi}$ with ground truth adjacency matrix $A$, denoted by $D_{top}(\hat{\pi}, A)$, is defined as: $D_{top}(\hat{\pi}, A) = \sum_{i=1}^{n}\sum_{j:\hat{\pi}_i > \hat{\pi}_j} A_{ij}$ where $A_{ij} = 1$ if there is a directed edge from node $i$ to $j$ else $A_{ij} = 0$. $D_{top}(\hat{\pi}, A)$ counts the number of ground-truth edges that cannot be recovered due to the estimated topological order $\hat{\pi}$.*

Structural Hamming Distance (SHD) is also a popular metric for assessing the goodness of a predicted DAG. Given a true DAG $\mathcal{G}$ and an estimated DAG $\hat{\mathcal{G}}$, SHD counts the number of missing, falsely detected, and falsely directed edges in $\hat{\mathcal{G}}$. $D_{top}$ acts as a lower-bound on SHD (Rolland et al., 2022).

### 3.1 CAUSAL ORDER FROM A PERFECT EXPERT IS ALWAYS ACCURATE, BUT GRAPH IS NOT

The predominant approach to extract causal knowledge from LLMs is to use a pairwise prompt (Kıcıman et al., 2023; Long et al., 2022; Choi et al., 2022) to determine the existence of an edge and then aggregate to build a causal graph. We highlight a key limitation of pairwise prompts for inferring edges and causal graphs, even with a hypothetical perfect expert, as LLMs are imperfect.

Revisiting the two graphs in Fig. 1, the second graph is estimated by asking pairwise questions to a *perfect* expert that (hypothetically) knows about all cause-effect relationships in a domain (see Defn. 3.3 for a formal definition). The difference in edge predictions is introduced due to the existence of direct and indirect effects. For example, when asked about the relationship between *Pollution* and *Dyspnoea*, it may be valid to draw a direct edge if the expert is not aware of the *Cancer* node. As a result, if we compare the estimated graph in Fig. 1 using standard graph comparison metrics such as SHD, we may find that that the estimated graph is significantly different from the true graph and (incorrectly) conclude that the expert's knowledge was insufficient. Instead, if we compute the causal order using Def. 3.1 for the predicted graph (Fig. 1 right), we obtain $\{Smoking, Pollution\} \prec Cancer \prec \{Dyspnoea, X\text{-}ray\}$. This order is fully consistent with the true graph (Fig. 1 left), and thus is a valid causal order. We could thus correctly validate the expert's knowledge as perfect. In particular, using causal order as the *output interface* of the expert-estimated graph ensures that no incorrect constraints are added. If the expert was asked to output the entire graph, erroneous edge constraints such as $Pollution \rightarrow Dyspnoea$ may be added to a downstream discovery algorithm. However, causal order only constrains that some path exists from $Pollution$ to $Dyspneoa$, and allows the downstream algorithm to learn the correct edges from data.

Note that the limitation is not about using a pairwise prompt, but using its output to infer edges in a graph. As stated earlier, given a pair of variables, it is not possible to determine whether an edge exists between them, without knowing whether potential mediators between the two variables exist. By not explicitly inferring edges, causal order instead corresponds to an ancestor-descendant relationship between a pair of variables which can be objectively decided given only the two variables. One can view our approach as formalizing the intuition in Ban et al. (2023b) who consider an LLM's pairwise answer to represent ancestor relationship between a pair of variables. We now formally show that causal order is a more accurate measure of an expert's knowledge. All proofs are in Appendix B.

**Definition 3.3.** *Perfect Expert. A perfect expert is an entity with access to the full ground-truth DAG $\mathcal{G}(\mathbf{X}, \mathbf{E})$. Given two variables two variables, $X_i, X_j \in \mathbf{X}$, and (optionally) an auxiliary set of nodes $\mathbf{O}_{ij} \subset \mathbf{X}$ (note that rest of the variables in set $\mathbf{U} = \mathbf{X} \setminus \mathbf{O}_{ij} \bigcup \{X_i, X_j\}$ need not be known), the expert can provide information on the existence of a causal edge between $X_i$ and $X_j$ ("does $X_i$ cause $X_j$") as follows:*

- *$X_i \rightarrow X_j$: If there is directed edge from $X_i$ to $X_j$ ($X_i \rightarrow X_j \in \mathbf{E}$), or if a directed path exists from $X_i$ to $X_j$ such that it does not contain any node $Z \in \mathbf{O}_{ij}$.*

- *$X_j \rightarrow X_i$: If there is directed edge from $X_j$ to $X_i$ ($X_j \rightarrow X_i \in \mathbf{E}$), or if a directed path exists from $X_j$ to $X_i$ such that it does not contain any node $Z \in \mathbf{O}_{ij}$.*

- *Otherwise, output no edge.*

**Definition 3.4.** *Level Order. Given a causal DAG $\mathcal{G}(\mathbf{X}, \mathbf{E})$, its level order is the systematic assignment of levels to variables, beginning with level $0$ to the set of variables $\{X_i|pa(X_i) = \emptyset\}$.*

*Subsequently, each remaining variable is assigned a level such that for each variable at a given level $i$, the length of the longest directed path from one/more variables in level 0 is $i$.*

**Proposition 3.1.** *Let the true causal DAG be $\mathcal{G}(\mathbf{X}, \mathbf{E})$ with ground-truth adjacency matrix $A$. Consider a procedure to estimate a graph $\hat{G}$ by querying a Perfect Expert (as in Def. 3.3) with pairwise queries $X_i$, $X_j$ with auxiliary set $\mathbf{O}_{ij}$, followed by subsequent aggregation of predicted edges from each query (i.e. from a total of $^{|\mathbf{X}|}C_2$ queries). The causal order of the graph $\hat{G}$ thus estimated is correct, i.e. $D_{top}(\pi(\hat{\mathcal{G}}), A) = 0$ for all values of the sets $\mathbf{O}_{ij}$. As a corollary, the causal graph thus estimated can however have errors. In other words, when $\mathbf{O}_{ij} = \phi \; \forall i, j$, $D_{top}(\pi(\hat{\mathcal{G}}), A) = 0$ whereas Structural Hamming Distance (SHD) between $\mathcal{G}$ and $\hat{\mathcal{G}} = \sum_{i=1}^{|\mathbf{X}|} |de(X_i)| - |ch(X_i)|$.*

Figure 3 illustrates the result of the proposition using empirical simulation. Given a fixed number of nodes, we sample a graph at random as the 'ground truth' and then consider all graph orientations of the same size (number of nodes) such that $D_{top} = 0$ w.r.t. the ground truth graph. These are potentially the graphs outputted by a Perfect Expert with different values of the auxiliary set $\mathbf{O}$. For this set of graphs, we compute SHD w.r.t the ground truth graph. Notice the variance in SHD, despite $D_{top}$ being 0. For graphs with six nodes, SHD can vary from 0 to 14 even as $D_{top} = 0$.

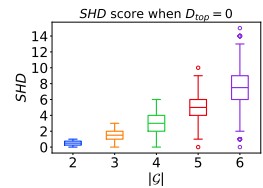

Figure 3: Variability of SHD for various graph sizes with $D_{top} = 0$ within each graph.

The above observations indicate that SHD can be high even when we obtain information from a Perfect Expert, but $D_{top}$ is always 0. This result is of significance since most estimated graphs (including those that are LLM-generated (Ban et al., 2023b; Long et al., 2023)) are evaluated using graph metrics such as SHD. Rather than the graph, it motivates us to posit the use of causal order as a more accurate output interface for experts' domain knowledge, since it allows objective evaluation of the expert's output using the topological divergence metric (Defn 3.2).

## 3.2 DOWNSTREAM UTILITY OF CAUSAL ORDER: DISCOVERY AND EFFECT INFERENCE

While the causal order is a more stable measure of experts' knowledge than the full graph, a natural question is whether it is a useful measure by itself. We now show the utility of causal order for effect estimation and causal discovery, which is also demonstrated by our experimental results in Sec 5. Specifically, we show that causal order is sufficient to find a valid backdoor set and $D_{top}$ is an ideal metric to minimize for effect estimation, assuming no latent confounders. Effect estimation error correlates more with topological divergence than it does with SHD. Causal order is also useful as a prior or constraint to increase accuracy of graph discovery algorithms.

**Correct topological order is necessary and sufficient for finding a valid backdoor set.** We first present the (known) result that a correct causal order is sufficient for identifying a backdoor set. We assume there are no unobserved variables in the underlying causal graph.

**Proposition 3.2.** *(Pearl, 2009; Cinelli et al., 2022) Under the no latent confounding assumption, for a pair of treatment and target variables $(X_i, X_j)$ in a DAG $\mathcal{G}$, $\mathbf{Z} = \{X_k | \pi_k < \pi_i\}$ is a valid adjustment set relative to $(X_i, X_j)$ for any topological order $\pi$ of $\mathcal{G}$.*

Proofs of all propositions are provided in App. § B. Propn 3.2 states, in simple words, that all variables that precede the treatment variable in a topological order $\pi$ of $\mathcal{G}$ constitute a valid adjustment set. Note that the set $\mathbf{Z}$ may contain variables that are not necessary to adjust for (e.g., ancestors of only the treatment or target variables). For statistical efficiency purposes, ancestors of the target variable are helpful for precise effect estimation, whereas ancestors of treatment variable can be harmful (Cinelli et al., 2022). In practical scenarios, however, it is recommended to adjust for all available adjustment variables since one cannot possibly rule out unknown confounding factors (Sauer et al., 2013; VanderWeele & Shpitser, 2011), which aligns with the set obtained using the causal order in Propn 3.2 (see App. C for a discussion).

We now show that $D_{top}$ is an optimal metric to minimize for effect estimation. That is, $D_{top}$ being 0 for a topological order is equivalent to obtaining the correct backdoor adjustment set using Propn. 3.2. And if $D_{top} \neq 0$, there exists some treatment-target pair whose backdoor set is not correctly identified.

**Proposition 3.3.** *For an estimated topological order $\hat{\pi}$ and a true topological order $\pi$ of a causal DAG $\mathcal{G}$ with the corresponding adjacency matrix $A$, $D_{top}(\hat{\pi}, A) = 0$ iff $\mathbf{Z} = \{X_k | \hat{\pi}_k < \hat{\pi}_i\}$ is a valid adjustment set relative to $(X_i, X_j)$, $\forall \pi_i < \pi_j$.*

Empirically, the correlation of $D_{top}$ with effect estimation is shown in App. E.2 for common BNLearn datasets. As long as $D_{top}$ is zero, changing the graph has no impact on effect estimation error.

**Topological order can improve accuracy of graph discovery algorithms.** Constraints implied by the topological order can be used to reduce the search space for discovery algorithms. For instance, if $X_i \prec X_j$ in the order, then $X_i$ cannot be a descendant of $X_j$ in the corresponding causal graph.

*Using causal order with Constraint-Based Discovery Methods:* Constraint-based causal discovery algorithms usually return a Completed Partially Directed Acyclic Graph (CPDAG), from which a Markov equivalence class of graphs can be obtained. However, not all edges in a CPDAG are oriented. Given a CPDAG from a constraint-based algorithm like PC (Spirtes et al., 2000), we use the causal order $\hat{\pi}$ obtained from experts to orient the undirected edges, similar to the algorithm from Meek (1995). Iterating over undirected edges, we first check if the nodes of that edge occur in $\hat{\pi}$. If yes, we orient the edge according to $\hat{\pi}$. Since it is possible that the causal order obtained from querying experts may not include some nodes (Isolated Nodes (IN)), if either (or both) nodes of the undirected edge are not in $\hat{\pi}$, we query a superior expert (e.g. oracle) (see Sec 4) to finalize a direction between the pair. Algorithm 1 (Appendix D) outlines the specific steps for this integration.

*Using causal order with score-based discovery methods:* Score-based methods like CaMML (Wallace et al., 1996) allow the specification of prior constraints which are respected while obtaining the complete graph. We hence utilize the causal order $\hat{\pi}$ obtained from experts as a level order prior (Defn 3.4) to such methods. We handle any cycles in the expert's output by assigning all nodes in a cycle to the same level. The approach is similar to an LLM-prior approach by Ban et al. (2023b) where the output of LLM and a score-based method are combined using an ancestral constraint. This approach also allows us to provide a prior probability to control the influence of prior on the discovery method. Algorithm 2 (Appendix D) outlines the specific steps for this integration.

## 4 OBTAINING A CAUSAL ORDER FROM IMPERFECT EXPERTS

If we assume a Perfect Expert, then aggregating edge responses from the standard pairwise prompt (Kıcıman et al., 2023) can yield an accurate order. However, in practice, LLMs are imperfect experts and their answers can contain unpredictable errors. As a result, aggregating responses from the pairwise prompt leads to many cycles in the final graph (see Sec. 5, Table 2), which in turn implies that the causal order is undefined. In this section, we propose two ways to reduce the errors made by an imperfect expert such as an LLM, motivated by Prop. 3.1 that showed that adding additional context may help an expert avoid creating unnecessary edges. First, we consider strategies to add auxiliary context in the pairwise prompt. Second, we propose a strategy that adds dynamic context to each variable pair by iterating over all triplets of variables.

### 4.1 ENHANCING ACCURACY OF PAIRWISE PROMPT

One way to avoid cycles is to make the pairwise prompt more robust. Beyond the standard pairwise prompt that asks the expert to identify the causal relationship between a pair of variables (Kıcıman et al., 2023), we consider the following strategies to add contextual information (see Appendix G).

- *Iterative Context.* Here we provide the previously oriented pairs as context in the query while iteratively prompting for next pair.
- *One-hop Iterative Context.* Providing all previously oriented pairs can become prohibitive for large graphs. Therefore, in this setting, we limit the provided information to the already oriented edges connecting the node pair under inspection with their adjacent neighbors. Specifically, we only supply the orientation details for the current node pair and their neighboring nodes.
- *Chain-of-Thought (+In-context Learning).* Here we include names of all variables in the graph as additional context. Based on recent results on providing in-context examples in LLM prompts for various tasks (Brown et al., 2020), we include examples of the ordering task (viz. node pairs and their correct causal ordering), before asking the question about the given nodes.

### 4.2 THE TRIPLET METHOD FOR PROMPTING LLMS

Rather than providing a pair of variables, another way is to provide a larger set of nodes in a prompt and ask LLM to obey the acyclicity constraint while providing the edges among them. The number of total prompts used for a graph with size $|V|$ would be $O(|V|^k)$ where $k$ is the size of the subset included in each prompt. In addition, LLM's accuracy is known to reduce as the query prompt becomes more complex (Levy et al., 2024). Therefore, while the set of nodes can be of any size, we

decide to go with triplet-based prompts as they allow for adding more context with minimal increase in prompt complexity and the total number of LLM calls. Moreover, empirically, we did not see a noticeable improvement in accuracy when moving from a triplet to quadruplet prompt (see Table A9).

In effect, we move from $O(|V|^2)$ calls to $O(|V|^3)$ LLM calls. A key benefit is that for each pair of nodes, we have $n-1$ responses from the LLM, each considering a different auxiliary node as context. For large graphs, we can use a variant that considers a constant $k$ responses for each pair of nodes, leading to $O(k|V|^2)$ complexity. To aggregate the final graph, we take a majority vote on the answers from each edge, further leading to robustness.

1. From a given set of graph nodes, we generate all possible triplets of nodes.
2. We query the expert to orient nodes of each triplet group to form a DAG representing the causal relationship between the triplet's nodes. This results in multiple acyclic mini-graphs representing causal relationships for each triplet group.
3. Once we have DAGs for each triplet, we focus on merging them. This is done in two steps: (i) We iterate over all node pairs, and for each combination we obtain a majority vote on the orientation between them across all triplets containing the node pair; (ii) In case of a conflict (or a tie in the majority vote) among the three possible edge orientations (A → B; B → A; No edge between A and B), we resort to a high-cost expert for tie-breaking.
4. Finally, a causal order is extracted from the merged graph.

Our triplet prompt additionally use in-context examples and the chain-of-thought strategy from the pairwise setup. An example prompt is shown in Table A28.

**Theoretical Analysis.** Next, we analyze the triplet strategy for its impact on predicting incorrect edges. We begin by defining (imperfect) $\epsilon$-experts as in Long et al. (2023). For ease of exposition, we define the $\epsilon$-expert to have error probability *exactly* equal to $\epsilon$; this could however be generalized to have error probability at most $\epsilon$. By enforcing the acyclicity constraint for each triplet, the triplet prompt avoids errors that a pairwise prompt may make. Below we assume that the $\epsilon$-expert's predictions satisfy acyclicity for subgraphs having 3 nodes.[*]

**Definition 4.1** ($\epsilon$-Experts). *Given two nodes A and B of a graph and three options of the causal relationship between them: (i) $A \rightarrow B$, (ii) $A \leftarrow B$, and (iii) no edge between A and B (denoted as $[c_1, c_2, c_3]$), an expert $\mathcal{E}$ queried for the causal relationship between A and B is said to be an $\epsilon$-expert (denoted as $\mathcal{E}_\epsilon$) if the probability of making an error in the prediction of the causal relationship between A and B is $\epsilon$, where $\epsilon \in (0, 1)$.*

**Proposition 4.1.** *Given two nodes A and B of an underlying causal graph, access to an $\epsilon$-expert $\mathcal{E}_\epsilon$ that doesn't produce any cycles in the predicted causal graph (see Assm B.1 for formal statement) and and renormalizes the probability in case an option is not available (see Assm B.2 for formal statement), let $C \neq A \neq B$ be any other node in the graph. If $\mathcal{E}_\epsilon$ predicts causal relationship between all pairs of nodes sequentially, the marginalized probability that $\mathcal{E}_\epsilon$ makes an error in predicting the causal relationship between A and B, after it has already predicted the causal relationships between $(C, A)$ and $(C, B)$, is less than $\epsilon$, where marginalization is over all possible causal graphs that can be formed between A, B and C, with each of such graphs being equally likely.*

Thus, given two nodes A and B, a querying strategy using triplets will have error probability $< \epsilon$ on determining the causal relationship between A and B than a pairwise strategy (proof in Appendix B). Still, some cycles may be produced in the aggregated global graph, hence we use a cycle removal algorithm from Zheng et al. (2018) in the third step of our triplet method. For every edge, we leverage the votes from the triplet prompts to establish a probability distribution over edge orientations. We use this to compute entropy for each edge, removing those with higher entropy (lower confidence). To minimize $D_{top}$, we prune edges with entropy below the mean of all entropies.

## 5 EXPERIMENTS AND RESULTS

**Datasets.** We evaluate the triplet method using benchmark datasets from the BNLearn repository (Scutari & Denis, 2014): Earthquake, Cancer, Survey, Asia, Asia modified (Asia-M), and Child. Asia-M is derived from Asia by removing the node *either* since it is not a node with a semantic meaning (see

---

[*]For imperfect experts, it is possible to enforce acyclicity by removing any cycles from their triplet output. However, this step is not needed for GPT-3.5 and GPT-4 as they follow acyclicity constraint with high accuracy.

App.§ G for details). To address memorization concerns with the BNLearn datasets, we also use recently proposed datasets that require nuanced medical domain understanding: (i) **Neuropathic**: A medium-sized subset graph from a relatively less popular Neuropathic dataset (Tu et al., 2019) (see Appendix Fig A8). (ii) **Alzheimers:** This graph (refer Figure A9) provides features (such as ventricular volume, brain volume, APOE4, etc) to study the clinical and phenotype of Alzheimer's disease (Abdulaal et al., 2024). It was created by a consensus of human experts. (iii) **Covid-19:** This graph, curated by medical experts, models the pathophysiological process of SARS-CoV-2 in the respiratory system which involves outlining the various pathways from viral infection to key complications (refer Figure A10). Orienting this graph requires understanding of how nodes like Pulmonary capillary leakage, systemic inflammatory response, Virema and more influence each other (Mascaro et al., 2022). All graphs are real-world graphs constructed by human experts. We provide more details on the datasets in Table A18.

**Imperfect Experts.** We consider two types of imperfect experts: ***LLMs*** and ***human annotators***.

*Human Annotation.* We considered 15 human annotators, each with undergrad-level training in STEM but no formal experience in causality. Each annotator was randomly assigned graphs for pairwise and triplet query strategies while ensuring no annotator got the same graph to query with both strategies. To get an estimate of the upper bound of human performance, for resolving tie-breaking conflicts in the triplet method, we used a ground truth-based oracle (proxy for a human domain expert). For each dataset, three human annotators were asked to annotate the final graph and the aggregate of that was reported. For feasibility reasons, human annotations were done only for the BNLearn graphs.

*LLM-Based.* We consider two main LLM-based experts, GPT-3.5-turbo and GPT-4. For the triplet method, GPT-4 was used for tie-breaking. To understand the effect of model size, we also evaluate the pairwise and triplet methods on Phi-3 (3.8B parameters) (Abdin, 2024) and Llama3 (8B parameters) (Dubey, 2024) which are significantly smaller models than GPT-3.5-turbo and GPT-4.

**Baselines.** In addition to the pairwise prompt and its extensions from Sec. 4.1, we consider two methods based on breadth-first search from Jiralerspong et al. (2024). The first method, BFS, iterates over nodes and uses an LLM to query children of each node. The second method, BFS+Stats, uses correlation coefficient between nodes as additional context in the LLM's prompt.

## 5.1 ACCURACY OF CAUSAL ORDER WITH TRIPLET VS. PAIRWISE METHODS

We first present the accuracy of obtaining causal order using our triplet method over other pairwise query strategies. Subsequently, we present the results of using the causal order obtained from imperfect experts to downstream tasks such as causal discovery and effect inference.

***Human Experts.*** With human annotators, Table 1 shows that graphs like *Survey* and *Asia-M* result in cycles when queried pairwise. However, no cycle formations were observed across annotators when they were queried to orient causal graphs using the triplet method. Moreover, the triplet method shows consistently low $D_{top}$ and $SHD$ across all human outputs, highlighting its effectiveness.

***GPT-3.5-turbo as Expert.*** Tables A5 and A6 present the performance of various pairwise optimization strategies from Sec. 4.1. While strategies like CoT offer some gains over the base pairwise method, they often produce cycles, especially in larger graphs like Child. These findings show that our pairwise variations improve graph discovery but still fall short. Table 3 compares the base pairwise method, best pairwise variation (CoT), and the triplet method across benchmark datasets using metrics like $D_{top}$, SHD, Cycles, IN, and Total Nodes (TN). Triplet method consistently outperforms the best pairwise CoT approach, showing a significant performance gap over the base pairwise method. For larger graphs like Child, the pairwise base approach shows a more pronounced difference, with higher cycle counts and SHD. Results on the Neuropathic dataset further confirm that the triplet method yields low $D_{top}$ and significantly lower SHD than pairwise methods.

***GPT-4 as Expert.*** Table A7 shows the impact of using a more advanced model like GPT-4 for the pairwise method. Despite superior model capabilities, we observe a consistently high number of cycles in bigger, complex graphs such as Child, Neuropathic, Covid-19 and Alzheimers, indicating that simply upgrading the model is not sufficient. In comparison, upgrading to GPT-4 for orienting subgraphs for the triplet method leads to further performance improvements as shown in Table A10.

***Results with Small LMs.*** To assess the robustness of the triplet method, we use it with small LMs such as Phi-3 and LLama3-8B as experts and GPT-4 for tie-breaker. Remarkably, as shown in Table 2, the

| Dataset | Metric | Pairwise | Triplet |
|---------|--------|----------|---------|
| | Using Human Annotators | | |
| Earthquake | $D_{top}$ | **0** | **0** |
| | SHD | 4.67 | **1.67** |
| | Cycles | **0** | **0** |
| | IN | **0** | 0.33 |
| Survey | $D_{top}$ | - | **0** |
| | SHD | 6.33 | **3.67** |
| | Cycles | 0.67 | **0** |
| | IN | 0.67 | **0** |
| Cancer | $D_{top}$ | **0** | **0** |
| | SHD | 4.33 | **3.67** |
| | Cycles | **0** | **0** |
| | IN | 0.67 | **0** |
| Asia-M | $D_{top}$ | - | **1.33** |
| | SHD | 11.67 | **11.33** |
| | Cycles | 3 | **0** |
| | IN | **0** | **0** |

Table 1: Experiments with non-expert human annotators show that the triplet method consistently produces lower SHD and $D_{top}$ values.

| Dataset | Metric | Pairwise GPT-4 | Triplet Phi-3 | Triplet Llama3 |
|---------|--------|----------------|---------------|----------------|
| Asia | $D_{top}$ | 1 | **0** | 2 |
| | SHD | 18 | **13** | 17 |
| | Cycles | **0** | **0** | **0** |
| | IN/TN | **0/5** | 1/5 | **0/5** |
| Alzheimers | $D_{top}$ | - | 7 | **5** |
| | Cycles | 1 | **0** | **0** |
| | IN/TN | **0/11** | **0/11** | 1/11 |
| Child | $D_{top}$ | - | 17 | **12** |
| | SHD | 148 | **69** | 129 |
| | Cycles | »10k | **0** | **0** |
| | IN/TN | **0/20** | **0/20** | **0/20** |

Table 2: Comparison of triplet method using Phi-3/Llama3 against pairwise (base) using GPT-4. Triplet method with significantly smaller models obtains lower SHD and $D_{top}$ values while avoiding cycles.

Table 3: Results using GPT-3.5-Turbo. Performance of triplet method, best performing pairwise query strategy (Chain of Thought), standard pairwise technique (Base) on multiple benchmark datasets across diff metrics: $D_{top}$, SHD, (Num of) Cycles, IN, TN. When number of cycles>0, $\hat{\pi}$ cannot be computed, hence $D_{top}$ is given by '-'. While CoT method shows improvement over base pairwise, triplet method outperforms pairwise methods across all datasets and metrics, with significant improvements on larger graphs such as *Child* and *Neuropathic*.

| Dataset | Metric | Pairwise (Base) | Pairwise (CoT) | Triplet |
|---------|--------|-----------------|----------------|---------|
| | Using LLM | | | |
| Earthquake | $D_{top}$ | **0** | **0** | **0** |
| | SHD | 7 | 4 | 4 |
| | Cycles | **0** | **0** | **0** |
| | IN/TN | **0/5** | **0/5** | **0/5** |
| Survey | $D_{top}$ | 3 | 1 | **0** |
| | SHD | 12 | **9** | 9 |
| | Cycles | **0** | **0** | **0** |
| | IN/TN | **0/6** | 2/6 | **0/6** |
| Cancer | $D_{top}$ | **0** | - | 1 |
| | SHD | 6 | - | **6** |
| | Cycles | 0 | - | **0** |
| | IN/TN | **0/5** | - | **0/5** |
| Asia-M | $D_{top}$ | - | - | **1** |
| | SHD | 15 | 13 | **11** |
| | Cycles | 7 | 1 | **0** |
| | IN/TN | **0/7** | **0/7** | **0/7** |
| Child | $D_{top}$ | - | - | **1** |
| | SHD | 177 | 138 | **28** |
| | Cycles | »3k | »500 | **0** |
| | IN/TN | **0/20** | **0/20** | **0/20** |
| Covid | $D_{top}$ | - | **0** | 0 |
| | SHD | 41 | 27 | 30 |
| | Cycles | »1000 | **0** | **0** |
| | IN/TN | **0/20** | **0/20** | **0/20** |
| Alzheimers | $D_{top}$ | - | 6 | **4** |
| | SHD | 42 | **26** | 28 |
| | Cycles | 684 | **0** | **0** |
| | IN/TN | **0/20** | **0/20** | **0/20** |
| Neuropathic | $D_{top}$ | - | - | **3** |
| | SHD | 212 | 64 | **24** |
| | Cycles | »5k | 5 | **0** |
| | IN/TN | **0/22** | **0/22** | 13/22 |

triplet method using smaller LMs outperforms the base pairwise method using GPT-4, particularly for complex networks. Results with small LMs are shown in Table A8. The triplet method outperforms the pairwise method consistently, yielding low $D_{top}$ values for both small and large graphs.

***Comparison with BFS and BFS+Stats.*** We also compare the triplet method to recently proposed BFS-based methods on a subset of the datasets in Table A17. BFS and BFS+Stats methods obtain lower accuracy than the triplet method. Across datasets, SHD and $D_{top}$ for BFS and BFS+Stats methods (especially with GPT-3.5-turbo) are higher than the triplet method. Among the Child and Covid-19 datasets, all configurations lead to cycles in atleast one of them, except BFS with GPT-4.

***Cost Estimation Analysis: Pairwise vs Triplet for LLMs.*** The triplet method optimizes cost by using smaller models efficiently, reserving larger models for clash resolution, reducing inference costs while improving accuracy over pairwise methods. See Appendix F for a detailed cost comparison.

## 5.2 USING CAUSAL ORDER FOR DOWNSTREAM APPLICATIONS

***Causal Discovery.*** Table 4 presents the $D_{top}$ results of using the causal order obtained from the triplet method (both using LLMs and humans) to assist causal discovery methods. We compare our approach using the triplet method with well-known causal discovery methods: PC (Spirtes et al., 2000), SCORE (Rolland et al., 2022), ICA-LiNGAM (Shimizu et al., 2006), Direct-LiNGAM (Shimizu et al., 2011), NOTEARS (Zheng et al., 2018), and CaMML (Wallace et al., 1996) across five different sample sizes: $250, 500, 1000, 5000, 10000$ (complete results in Table A3). Among the discovery algorithms, we find that PC and CaMML perform the best, with the lowest $D_{top}$ across all datasets. We hence studied 4 variants of using the causal order with discovery algorithms: PC+Human,

| | Dataset | PC | SCORE | ICA LiNGAM | Direct LiNGAM | NOTEARS | CaMML | Ours (LLM+PC) | Ours (LLM+CamML) | Ours (Human+PC) | Ours (Human+CaMML) |
|---|---|---|---|---|---|---|---|---|---|---|---|
| $N = 250$ | Earthquake | 0.16±0.28 | 4.00±0.00 | 1.00±0.00 | 1.00±0.00 | 1.00±0.00 | 2.00±0.00 | **0.00±0.00** | **0.00±0.00** | **0.00±0.00** | 1.00±0.00 |
| | Cancer | **0.00±0.00** | 3.00±0.00 | 2.00±0.00 | 2.00±0.00 | 2.00±0.00 | 2.00±0.00 | **0.00±0.00** | **0.00±0.00** | **0.00±0.00** | **0.00±0.00** |
| | Survey | 0.50±0.00 | 4.00±0.00 | 2.00±0.00 | 4.00±0.00 | 4.00±0.00 | 3.33±0.94 | **0.00±0.00** | 3.33±0.94 | **0.00±0.00** | **0.00±0.00** |
| | Asia | 2.00±0.59 | 7.00±0.00 | 3.33±0.47 | 1.00±0.00 | 3.00±0.00 | 1.85±0.58 | 1.00±0.00 | **0.97±0.62** | N/A | N/A |
| | Asia-M | 1.50±0.00 | 6.00±0.00 | 1.00±0.00 | 3.00±0.00 | 3.00±0.00 | **1.00±0.00** | **1.00±0.00** | 1.71±0.45 | **1.00±0.00** | 2.00±0.00 |
| | Child | 5.75±0.00 | 12.0±0.00 | 14.33±0.47 | 16.0±0.00 | 14.0±0.00 | **3.00±0.00** | 4.00±0.00 | 3.53±0.45 | N/A | N/A |
| | Neuropathic | 4.00±0.00 | 6.00±0.00 | 13.0±6.16 | 10.0±0.00 | 9.00±0.00 | 10.4±1.95 | **3.00±0.00** | 5.00±0.00 | N/A | N/A |
| $N = 10000$ | Earthquake | **0.00±0.00** | 4.00±0.00 | 3.00±0.00 | 3.00±0.00 | 1.00±0.00 | 0.40±0.48 | **0.00±0.00** | **0.00±0.00** | **0.00±0.00** | **0.00±0.00** |
| | Cancer | 2.00±0.00 | 3.00±0.00 | 3.00±0.00 | 3.00±0.00 | 2.00±0.00 | 0.60±0.80 | 2.00±0.00 | **0.00±0.00** | 2.00±0.00 | **0.00±0.00** |
| | Survey | 2.00±0.00 | 4.00±0.00 | 5.00±0.00 | 5.00±0.00 | 3.00±0.00 | 3.60±1.35 | 2.00±0.00 | 1.83±0.00 | 2.00±0.00 | **0.00±0.00** |
| | Asia | 1.5±0.00 | 4.00±0.00 | 6.00±0.00 | 4.40±1.35 | 3.00±0.00 | 1.40±0.48 | **0.00±0.00** | 0.34±0.47 | N/A | N/A |
| | Asia-M | 1.00±0.00 | 4.00±0.00 | 8.00±0.00 | 4.80±0.39 | 3.00±0.00 | 2.00±0.00 | **0.00±0.00** | **0.00±0.00** | **0.00±0.00** | 3.00±0.00 |
| | Child | 6.00±3.04 | 3.00±0.00 | 12.2±1.46 | 11.6±0.48 | 14.4±0.48 | 2.80±0.84 | 5.00±2.64 | **1.00±0.00** | N/A | N/A |
| | Neuropathic | 10.00±0.00 | 6.00±0.00 | **1.00±0.00** | 10.0±0.00 | 10.0±0.00 | 3.00±0.00 | 10.00±0.00 | **1.00±0.00** | N/A | N/A |

Table 4: Comparison with causal discovery methods, showing mean and std dev of $D_{top}$ over 3 runs. (For the Neuropathic subgraph (1k samples), PC Algorithm returns cyclic graphs in the MEC). Human experiments not conducted for Neuropathic, Child (due to feasibility issues) and Asia; hence rows marked as N/A.

CaMML+Human, PC+LLM, and CaMML+LLM. The results show that using expert-provided causal order improves $D_{top}$ across our experiments consistently. Specifically, the improvement (reduction) in $D_{top}$ when using our approach is larger at lower sample sizes. This indicates that obtaining causal order from imperfect experts like humans and LLMs can help with causal discovery in limited sample settings. While the results on the BNLearn datasets may be impacted by memorization, we obtain consistent results on the less popular Neuropathic dataset that requires nuanced medical knowledge.

***Causal Effect Inference.*** Table 5 presents the results of using the causal order obtained from the triplet method to compute average causal effect (ACE). We report the error in ACE $\epsilon_{ACE}$ across the same set of methods and datasets as above. The obtained causal order shows unanimous improvement in performance across the studies, especially when using the causal order from CaMML+LLM. Following Proposition 3.2, we use all variables that precede the treatment variable in estimated topological order as the adjustment set. Once the adjustment set is identified, the causal effect is estimated using the DoWhy library (Sharma & Kiciman, 2020) and linear regression as the estimator. Table A16 compares the causal effects estimated using this approach versus minimal backdoor set adjustment in the Asia dataset, showing minimal differences.

| Dataset | Metric: $\epsilon_{ACE}$ (Treatment, Target) | PC | SCORE | ICA LiNGAM | Direct LiNGAM | NOTEARS | CaMML | Ours (LLM+PC) | Ours (LLM+CaMML) |
|---|---|---|---|---|---|---|---|---|---|
| Earthquake | (JohnCalls,alarm) | **0.00 ± 0.00** | 0.85 ± 0.02 | 0.63 ± 0.10 | 0.63 ± 0.10 | 0.21 ± 0.12 | 0.08 ± 0.03 | **0.00 ± 0.00** | **0.00 ± 0.00** |
| Cancer | (dyspnoea,cancer) | 0.20 ± 0.01 | 0.30 ± 0.00 | 0.30 ± 0.01 | 0.30 ± 0.01 | 0.18 ± 0.02 | 0.06 ± 0.00 | 0.30 ± 0.00 | **0.00 ± 0.00** |
| Survey | (T,E) | 0.02 ± 0.00 | 0.04 ± 0.00 | 0.05 ± 0.01 | 0.05 ± 0.01 | 0.03 ± 0.00 | 0.03 ± 0.00 | 0.02 ± 0.00 | 0.01 ± 0.01 |
| Asia | (smoke,dyspnoea) | 0.10 ± 0.00 | 0.09 ± 0.00 | 0.27 ± 0.03 | 0.27 ± 0.04 | 0.14 ± 0.01 | 0.05 ± 0.00 | 0.02 ± 0.00 | **0.00 ± 0.00** |
| Child | (Lung Parench, Lowerbody O2) | 0.22 ± 0.01 | 0.02 ± 0.00 | 0.52 ± 0.00 | 0.52 ± 0.00 | 0.52 ± 0.07 | 0.01 ± 0.00 | 0.22 ± 0.00 | **0.00 ± 0.00** |

Table 5: Comparison of causal effect inference with existing methods, showing mean and std dev of error in Average Causal Effect ($\epsilon_{ACE}$) of a variable on another, over 3 runs.

| | PC | SCORE | ICA LiNGAM | Direct LiNGAM | NOTEARS | CaMML | Ours (LLM+PC) | Ours (LLM+CaMML) |
|---|---|---|---|---|---|---|---|---|
| $N = 250$ | 4.00±0.00 | 6.00±0.00 | 13.0±6.16 | 10.0±0.00 | 9.00±0.00 | 10.4±1.95 | **3.00±0.00** | 5.00±0.00 |
| $N = 10000$ | 10.00±0.00 | 6.00±0.00 | **1.00±0.00** | 10.0±0.00 | 10.0±0.00 | 3.00±0.00 | 10.00±0.00 | **1.00±0.00** |

Table 6: Performance on causal discovery for the *Neuropathic* dataset subgraph (1k samples), showing mean and std dev of $D_{top}$ over 3 runs.

# 6 CONCLUDING DISCUSSION

Obtaining reliable knowledge from imperfect experts is challenging. We presented causal order as a suitable output interface to elicit causal knowledge from imperfect experts like LLMs and human annotators. Compared to the full graph, we showed that causal order is a more stable quantity to elicit from imperfect experts since it avoids making a distinction between direct and indirect effects. We also proposed a novel triplet-based method to query experts for obtaining the causal order.

*Limitations.* While LLMs can provide causal order for relationships over known variables, we do not expect them to be useful for completely novel nodes and causal relationships. Separately, causal order may not be sufficient for tasks such as counterfactual estimation (Janzing et al., 2019) that require the graph structure for estimating functional equations. For both these cases, a viable method may be to obtain causal order over known relationships from LLMs and use it as a prior /constraint for existing discovery methods to obtain the full causal graph.

ETHICAL IMPACT AND REPRODUCIBILITY

*Ethical Statement.* All datasets used in our work are publicly available and are accurate to the best of our knowledge. We made best efforts to compare against contemporary benchmarks in a fair manner. There may be no direct harmful impact, especially considering our causal order is only a pre-processing steps for downstream algorithms. However, since LLMs may be used in our approach, suitable prudence may be necessary to avoid ill-effects in applications.

*Reproducibility.* Our methods are fairly straightforward, and implementation details are already included in our paper descriptions. Our code is publicly available at https://github.com/AniketVashishtha/Causal_Order_Imperfect_Experts

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

## APPENDIX

In this appendix, we include the following additional information, which we could not include in the main paper due to space constraints:

- Appendix A: Illustration of our triplet query strategy
- Appendix B: Proofs of propositions
- Appendix C: Practicality of the adjustment set obtained using the Causal Order
- Appendix D: Algorithms to integrate causal order into existing discovery methods
- Appendix E: Additional results, including LLMs used in post-processing for graph discovery and a discussion of triplet vs pairwise query strategies
- Appendix F: More details and examples of our query strategies
- Appendix G: Causal graphs used in our experiments of the datasets

## A    ILLUSTRATION OF OUR TRIPLET QUERY STRATEGY

We present an intuitive illustration of our overall triplet querying framework to obtain causal order from imperfect experts in Fig A1 below.

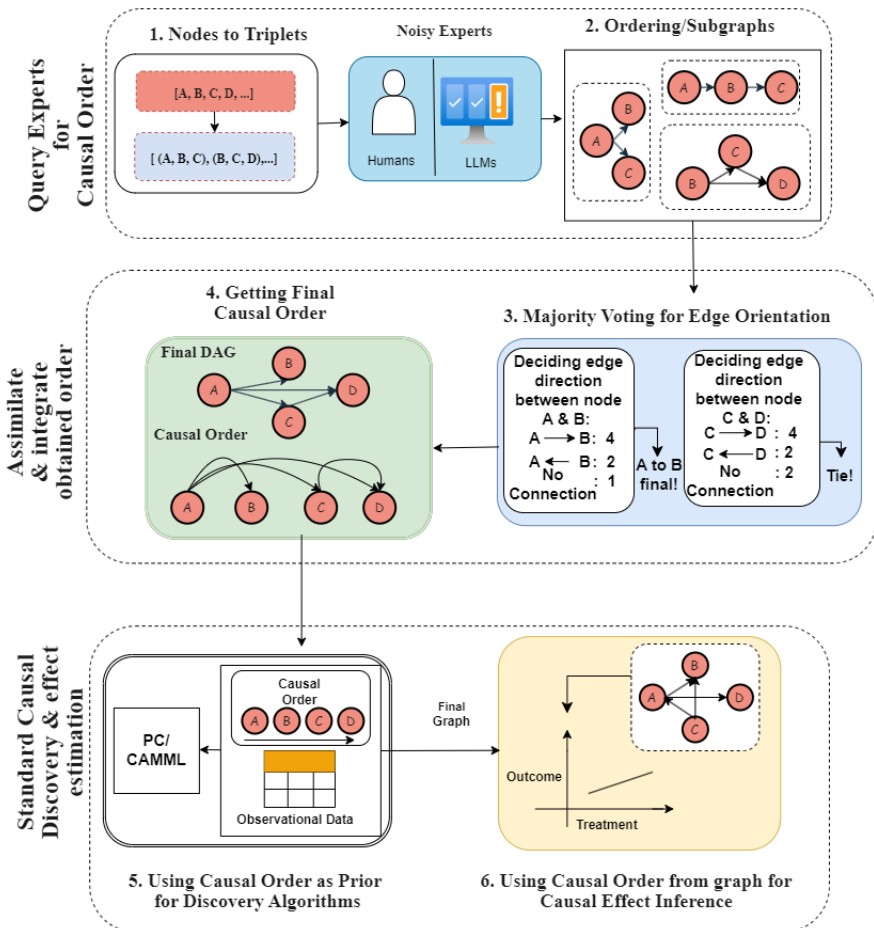

Figure A1: **Leveraging Causal Order from Imperfect Experts.** Our triplet-based querying method infers all three-variable subgraphs from imperfect experts and aggregates them (using majority voting) to produce a causal order. Ties in causal order are broken using a high-cost expert. Expert-generated causal order is integrated with discovery algorithms, before estimating causal effect.

## B    PROOFS OF PROPOSITIONS

To estimate $\mathbb{E}[X_j|do(X_i = x_i)]$ from observational data, the *backdoor adjustment* formula is used.

**Definition B.1.** *Backdoor Adjustment (Pearl, 2009). Given a DAG $\mathcal{G}$, a set of variables $\mathbf{Z}$ satisfies the backdoor criterion relative to a pair of treatment and target variables $(X_i, X_j)$ if (i) no variable in $\mathbf{Z}$ is a descendant of $X_i$; and (ii) $\mathbf{Z}$ blocks every path between $X_i$ and $X_j$ that contains an arrow into $X_i$.*

**Proposition 3.1.** *Let the true causal DAG be $\mathcal{G}(\mathbf{X}, \mathbf{E})$ with ground-truth adjacency matrix $A$. Consider a procedure to estimate a graph $\hat{G}$ by querying a Perfect Expert (as in Def. 3.3) with pairwise queries $X_i$, $X_j$ with auxiliary set $\mathbf{O}_{ij}$, followed by subsequent aggregation of predicted edges from each query (i.e. from a total of $^{|\mathbf{X}|}C_2$ queries). The causal order of the graph $\hat{G}$ thus estimated is correct, i.e. $D_{top}(\pi(\hat{\mathcal{G}}), A) = 0$ for all values of the sets $\mathbf{O}_{ij}$. As a corollary, the causal graph thus estimated can however have errors. In other words, when $\mathbf{O}_{ij} = \phi \; \forall i, j$, $D_{top}(\pi(\hat{\mathcal{G}}), A) = 0$ whereas Structural Hamming Distance (SHD) between $\mathcal{G}$ and $\hat{\mathcal{G}}$ = $\sum_{i=1}^{|\mathbf{X}|} |de(X_i)| - |ch(X_i)|$.*

*Proof.* First claim ($D_{top}(\pi(\hat{\mathcal{G}}), A) = 0$): By definition, the Perfect Expert adds new edges that are not present in the true G, but cannot miss predicting a ground truth edge. This implies that all edges between any two level $i, j$ where $i < j$ that are present in the ground truth graph $\mathcal{G}$ are also present in the estimated graph $\hat{\mathcal{G}}$. Given any two nodes $X_1^l$ and $X_2^l$ with the same level order "$l$" in the true causal graph. Since there is no directed path between $X_1^l$ and $X_2^l$, the perfect expert will never predict any edge between them (using Def. 3.3). Combining these two observations, the level order of both the graphs $\hat{\mathcal{G}}$ and $\mathcal{G}$ remains the same. Next, we will use the following lemma that states that if the level order of two graphs remains the same then the topological order remains the same thus completing the proof of the first claim.

**Lemma B.1.** *Given two DAG $\mathcal{G}_1$ and $\mathcal{G}_2$ have same level order (see Def. 3.4) then there exist two topological order $\pi(\mathcal{G}_1)$ and $\pi(\mathcal{G}_2)$ corresponding to the two DAG s.t. the ordered set $\pi(\mathcal{G}_1) = \pi(\mathcal{G}_2)$.*

*Proof.* Since the level order is the same for both the graphs, all the nodes on a given level "$l$" for both graphs $\mathcal{G}_1$ and $\mathcal{G}_2$ are the same. Now, any two nodes on the same level don't have any edges between them. Thus add all the nodes on the level in the same order to both $\pi(\mathcal{G}_1) = \pi(\mathcal{G}_2)$. Thus when we are done adding the nodes from all the levels in the topological order we get the $\pi(\mathcal{G}_1) = \pi(\mathcal{G}_2)$. $\square$

Second claim (SHD $> 0$): Recall that SHD counts the number of missing, falsely detected, and falsely directed edges in the estimated causal graph as compared to the ground truth graph. Since the perfect expert correctly predicts all the ground truth edges, there are no *falsely directed* or *missing* edges in the predicted graph. From Def. 3.3, when queried over all $^{|\mathbf{X}|}C_2$ pairs of nodes the perfect expert will add additional (falsely directed) edges between a node and all its descendants. Thus total number of falsely directed edges = $\sum_{i=1}^{|\mathbf{X}|} |de(X_i)| - |ch(X_i)|$ = SHD.

$\square$

**Proposition 3.2.** *(Pearl, 2009; Cinelli et al., 2022) Under the no latent confounding assumption, for a pair of treatment and target variables $(X_i, X_j)$ in a DAG $\mathcal{G}$, $\mathbf{Z} = \{X_k | \pi_k < \pi_i\}$ is a valid adjustment set relative to $(X_i, X_j)$ for any topological order $\pi$ of $\mathcal{G}$.*

*Proof.* Before starting the proof, we define a confounding variable. A confounder is a variable that should be casually associated with both the treatment and the target variables and is not on the causal pathway between treatment and target. An unmeasured common cause can also be a source of confounding the treatment $\rightarrow$ target relationship. Coming to the proof, we need to show that the set $\mathbf{Z} = \{X_k | \pi_k < \pi_i\}$ satisfies the conditions (i) and (ii) in Defn B.1. For any variable $X_k$ such that $\pi_k < \pi_i$, we have $X_k \notin de(X_i)$ and hence the condition (i) is satisfied. Additionally, for each $X_k \in pa(X_i)$ we have $\pi_k < \pi_i$ and hence $pa(X_i) \subseteq \mathbf{Z}$. Since $pa(X_i)$ blocks all paths from $X_i$ to $X_j$ that contains an arrow into $X_i$ (Peters & Bühlmann, 2015), $\mathbf{Z}$ satisfies condition (ii). $\square$

**Proposition 3.3.** *For an estimated topological order $\hat{\pi}$ and a true topological order $\pi$ of a causal DAG $\mathcal{G}$ with the corresponding adjacency matrix $A$, $D_{top}(\hat{\pi}, A) = 0$ iff $\mathbf{Z} = \{X_k | \hat{\pi}_k < \hat{\pi}_i\}$ is a valid adjustment set relative to $(X_i, X_j)$, $\forall \pi_i < \pi_j$.*

*Proof.* The statement of proposition is of the form $A \iff B$ with $A$ being "$D_{top}(\hat{\pi}, A) = 0$" and $B$ being "$\mathbf{Z} = \{X_k | \hat{\pi}_k < \hat{\pi}_i\}$ is a valid adjustment set relative to $(X_i, X_j), \forall i, j$". We prove $A \iff B$ by proving (i) $A \implies B$ and (ii) $B \implies A$.

(i) Proof of $A \implies B$: If $D_{top}(\hat{\pi}, A) = 0$, for all pairs of nodes $(X_i, X_j)$, we have $\hat{\pi}_i < \hat{\pi}_j$ whenever $\pi_i < \pi_j$. That is, causal order in estimated graph is same that of the causal order in true graph. Hence, from Propn 3.2, $\mathbf{Z} = \{X_k | \hat{\pi}_k < \hat{\pi}_i\}$ is a valid adjustment set relative to $(X_i, X_j), \forall i, j$.

(ii) Proof of $B \implies A$: we prove the logical equivalent form of $B \implies A$ i.e., $\neg A \implies \neg B$, the *contrapositive* of $B \implies A$. To this end, assume $D_{top}(\hat{\pi}, A) \neq 0$, then there will be at least one edge $X_i \rightarrow X_j$ that cannot be oriented correctly due to the estimated topological order $\hat{\pi}$. i.e., $\hat{\pi}_j < \hat{\pi}_i$ but $\pi_j > \pi_i$. Hence, to find the causal effect of $X_i$ on $X_l$; $l \neq j$, $X_j$ is included in the back-door adjustment set $\mathbf{Z}$ relative to $(X_i, X_l)$. Adding $X_j$ to $\mathbf{Z}$ renders $\mathbf{Z}$ an invalid adjustment set because it violates the condition (i) of Defn B.1. $\square$

**Assumption B.1** (DAG Acyclicity). *Given that $\epsilon$-expert $\mathcal{E}_\epsilon$ is used to predict a causal graph between a set of nodes, the predicted causal graph is acyclic.*

**Assumption B.2** (Error Distribution and Probability Renormalization). *Let $[c_1, c_2, c_3]$ be the three choices for a causal relationship between node A and B (see Def 4.1). Let $P(c_1), P(c_2)$ and $P(c_3)$ be the probability of selecting the corresponding three choices by the $\epsilon$-expert $\mathcal{E}_\epsilon$. We assume that the probability for the two wrong options are equally likely, i.e., equal to $\epsilon/2$. If any constraint $\mathcal{T}$ renders some of the choices as not possible i.e. $P(c_j | \mathcal{T}) = 0$ for some $j \in \{1, 2, 3\}$, then $\mathcal{E}_\epsilon$ renormalizes the posterior probability over the other choices,i.e., $P(c_i | \mathcal{T}) = \frac{P(c_i)}{\sum_{j, P(c_j | \mathcal{T}) \neq 0} P(c_j)}$*

*where the denominator is summed over $j$ s.t. $P(c_j | \mathcal{T}) \neq 0$.*

**Proposition 4.1.** *Given two nodes A and B of an underlying causal graph, access to an $\epsilon$-expert $\mathcal{E}_\epsilon$ that doesn't produce any cycles in the predicted causal graph (see Assm B.1 for formal statement) and and renormalizes the probability in case an option is not available (see Assm B.2 for formal statement), let $C \neq A \neq B$ be any other node in the graph. If $\mathcal{E}_\epsilon$ predicts causal relationship between all pairs of nodes sequentially, the marginalized probability that $\mathcal{E}_\epsilon$ makes an error in predicting the causal relationship between A and B, after it has already predicted the causal relationships between $(C, A)$ and $(C, B)$, is less than $\epsilon$, where marginalization is over all possible causal graphs that can be formed between A, B and C, with each of such graphs being equally likely.*

*Proof.* Without any additional constraint, $\epsilon$-expert ($\mathcal{E}_\epsilon$) has "$\epsilon$" probability of making incorrect prediction. But in presence of additional constraint, e.g. DAG constraint (see Assm B.1), the probability of error changes and is given by the following lemma:

**Lemma B.2.** *Suppose we have two nodes A and B and three possible choices $[c_1, c_2, c_3]$ for causal relationship between them i.e $A \rightarrow B$, $B \rightarrow A$ or no edge between them (not in any particular order). Without loss of generality, let $c_3$ be the ground truth causal relationship between node A and B. Thus, without any additional constraint, let the probability assigned to each of the three choices by $\epsilon$-expert ($\mathcal{E}_\epsilon$) is $P(c_1) = \epsilon_1, P(c_2) = \epsilon_2$ and $P(c_3) = 1 - \epsilon_1 - \epsilon_2$ respectively where $\epsilon = \epsilon_1 + \epsilon_2$. If due to additional constraint (e.g. acyclicity Assm B.1), one of the incorrect choice gets discarded, say $c_1$, then the new probability of selecting the wrong choice ($c_2$ given by $\epsilon'$) is always less than $\epsilon$. However if the correct/ground truth choice is discarded due to this additional constraint the new probability of selecting the wrong choice ($c_1$ or $c_2$) is 1. In case, no options are discarded the new probability of choosing the wrong choice remains same i.e $\epsilon$ as before.*

*Proof.* For the case when the correct/ground truth choice i.e $c_3$ is discarded due to some constraint, the only left out choices are wrong choices i.e. $c_1$ and $c2$. Thus the probability of making error in selecting the correct choice is 1. Next, for the case when one of the incorrect choice (here $c_1$ w.l.o.g) is discarded, we are left with one incorrect ($c_2$) and one correct choice ($c_3$). From Assm B.2 once a particular option is discarded, the $\epsilon$-expert renormalizes the probability proportional to their initial probability. Thus the new probability ($\tilde{P}(c_2)$) of choosing wrong option $c_2$ is:

$$\tilde{P}(c_2) = \frac{\epsilon_2}{1 - \epsilon_1 - \epsilon_2 + \epsilon_2} = \frac{\epsilon_2}{1 - \epsilon_1} = \frac{\epsilon/2}{1 - \epsilon/2} = \frac{\epsilon}{2 - \epsilon} \tag{1}$$

where $\epsilon_1 = \epsilon_2 = \epsilon/2$ from Assm B.2. Next, we can show that $\tilde{P}(c_2) < \epsilon$ completing our proof. To have $\tilde{P}(c_2) < \epsilon$ we need:

$$\tilde{P}(c_2) = \frac{\epsilon_2}{1 - \epsilon_1} < \epsilon = \epsilon_1 + \epsilon_2$$
$$\implies \epsilon_2 < \epsilon_1 + \epsilon_2 - \epsilon_1^2 - \epsilon_1 \epsilon_2 \tag{2}$$
$$\implies \epsilon_1(\epsilon_1 + \epsilon_2 - 1) < 0$$

which is always true since from Assm B.2 we have $\epsilon_1 > 0$, $\epsilon_2 > 0$ and $1 - \epsilon_1 - \epsilon_2 > 0$. $\qquad\square$

Now, give any three nodes $A$, $B$ and $C$, Table A1 summarizes all possible *partially completed* graph (henceforth partial graph) possible between those nodes. Each partially-completed DAG in Table A1 generated more DAG based on the orientation of the node $A$ and $B$. Specifically, each of the partial graph $1, 2, 3, 4, 5, 7$ and $9$ generated three graphs ($A \to B$, $B \to A$ or no edge between $A$ and $B$) and partial graph $6$ and $8$ will give two DAG (one option is not possible to maintain acyclicity constraint). Thus overall we have 25 possible graphs. Our next goal is to show that the marginal probability of choosing the wrong causal relationship for node $(A, B)$ when oriented last among is less than $\epsilon$, where marginalization is over all the causal graph depicted in Table A1 (assuming all graphs are equally likely). The expert $\mathcal{E}_\epsilon$ finds the causal relationship sequentially for all the pairs in $\{(C, A), (C, B), (A, B)\}$. We are interested in the case when $\mathcal{E}_\epsilon$ finds the causal relationship for pair $(A, B)$ in the end. Let $F, S, T$ (called first,second and third) be three binary random variable and the value $0$ represent whether the causal relationship discovered by $\mathcal{E}_\epsilon$ for first, second or last/third pair respectively is incorrect and $1$ represent it is correct. So the probability of error when finding the causal relationship between node $A$ and $B$ when oriented last/third (denoted by $P(T)$) is given by:

$$P(T = 0) = \sum_{G \in \mathcal{G}} \sum_{S, T \in \{0,1\} \times \{0,1\}} P(G) P(F, S | G) P(T = 0 | F, S, G)$$
$$= \frac{1}{25} \cdot \sum_{G \in \mathcal{G}} \sum_{S, T \in \{0,1\} \times \{0,1\}} P(F, S | G) P(T = 0 | F, S, G) \tag{3}$$

where $\mathcal{G}$ denotes the set of graphs generated by orienting the causal relationship between $A$ and $B$ for all *partial* graphs in Table A1, $|\mathcal{G}| = 25$ and all the graphs are equally likely, different configuration of $(F, S)$ shows whether the causal relationship between first two pairs $(C, A)$ and $(C, B)$ are correct or not. When orienting the first two pair of nodes i.e $(C, A)$ and $(C, B)$ there is no DAG constraint thus we have:

$$P(F, S) = \begin{cases} \epsilon^2 & \text{when } S = 0, T = 0 \\ \epsilon(1 - \epsilon) & \text{when } S = 0, T = 1 \\ \epsilon(1 - \epsilon) & \text{when } S = 1, T = 0 \\ (1 - \epsilon)^2 & \text{when } S = 1, T = 1 \end{cases} \tag{4}$$

Now based on the graph $G \in \mathcal{G}$ and the setting of $S, T$, $P(T = 0 | F, S, G)$ takes different values. Suppose that the causal relationship between the first two pairs $(C, A)$ and $(C, B)$ are already predicted by the expert. We observe that the DAG acyclcity constraint (Assm B.2) will only change the probability of error for orienting nodes $(A, B)$ ($P(T = 0 | F, S, G)$ given by Lemma B.2) when the predicted causal graphs is either $B \to C \to A$ or $A \to C \to B$ after orienting $(C, A)$ and $(C, B)$. For all the other predictions of $(C, A)$ and $(C, B)$, they don't enforce any acyclicity constant for finding the causal relationship between $(A, B)$, thus, $P(T = 0 | F, S, G) = \epsilon$ (from Lemma B.2). Table A2 summarizes of error probability for all the partial graphs in Table A1 ($P(F, S | G)$ and $P(T | F, S, G)$). The first column shows different partial graphs from Table A1. The second column then shows different causal relationships that are possible between the nodes $A$ and $B$ for a particular partial graph. Given one *true orientation* between node $A$ and $B$ we get a final ground truth graph. Thus the third column shows the probability of prediction of structure $A \leftarrow C \to B$ for a particular true graph and the fourth column shows the probability of making an error in predicting the third causal relationship i.e between $(A, B)$ given the first and second pair $(C, A)$ and $((C, B))$ is already predicted. Similarly, the fifth and sixth columns show the same thing for the predicted structure $A \to C \to B$ for each of the ground truth graphs. The partial-graph number $4, 7, 8$ is not depicted in the table but the entries for $4^{th}$ graph is the same as $2^{nd}$, $7^{th}$ is the same as $3^{rd}$ and $8^{th}$ is same as $6^{th}$ due to symmetry in the partial-structure. The value of $\epsilon' = \frac{\epsilon}{2 - \epsilon}$ in $4^{th}$ and $6^{th}$ column is given by renormalized probability Eq. 1 in Lemma B.2. Substituting the values from Table A2 in Eq. 3

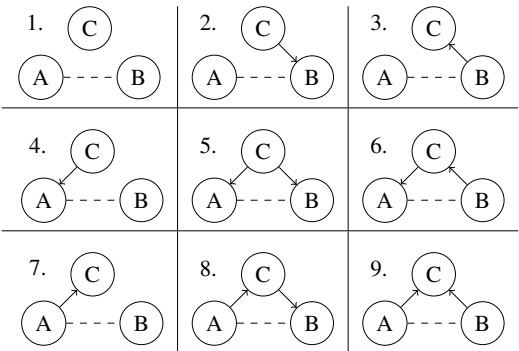

Table A1: All possible causal graph between three variables $A$, $B$ and $C$. The dashed arrow represented undecided causal relationship between node $A$ and $B$. So, the dashed arrow can take one of three choices $A \rightarrow B$, $A \leftarrow B$ or no edge between $A$ and $B$. To ensure that the graph is acyclic, some of the graphs above might not allow all three choice for causal relationship between node $A$ and $B$. Hence the causal-graph $1, 2, 3, 4$ and $7$ each have three possible graphs and $5, 6, 8$ and $9$ each have two possible graphs based on the valid choice of causal relationship between $A$ and $B$ that preserves acyclicity constraint. So overall there are 25 possible different causal graph between three variables $A$, $B$ and $C$.

and using the value $P(T|F, S, G) = \epsilon$ for the rest of the predicted structure not mentioned in the Table A2 we get:

$$
\begin{aligned}
P(T = 0) &= \frac{1}{25} \cdot \Bigg\{ 2 * \frac{\epsilon^2}{4} \Big[ 2\epsilon^{'} + 1 \Big] + \Big[ 1 - 2 * \frac{\epsilon^2}{4} \Big] \epsilon \\
&+ \left( \Big[ \frac{\epsilon^2}{4} + \frac{\epsilon(1 - \epsilon)}{2} \Big] \Big[ 2\epsilon^{'} + 1 \Big] + \Big[ 1 - \frac{\epsilon^2}{4} - \frac{\epsilon(1 - \epsilon)}{2} \Big] \epsilon \right) * 4 \\
&+ \left( 2 * \frac{\epsilon(1 - \epsilon)}{2} \Big[ 2\epsilon^{'} + 1 \Big] + \Big[ 1 - 2 * \frac{\epsilon(1 - \epsilon)}{2} \Big] \epsilon \right) * 2 \\
&+ \left( (1 - \epsilon)^2 \Big[ 2\epsilon^{'} \Big] + \frac{\epsilon^2}{4} \Big[ \epsilon^{'} + 1 \Big] + \Big[ 1 - (1 - \epsilon)^2 - \frac{\epsilon^2}{4} \Big] \epsilon \right) * 2 \Bigg\} \\
&= \frac{1}{25} \cdot \left\{ \frac{\epsilon(3\epsilon^2 - 30\epsilon + 52)}{4 - 2\epsilon} \right\}
\end{aligned}
\tag{5}
$$

Now we want to show that the error probability for the third pair $(A, B)$ given by the above equation is less than $\epsilon$. For that, we need:

$$
\begin{aligned}
\frac{1}{25} \cdot \left\{ \frac{\epsilon(3\epsilon^2 - 30\epsilon + 52)}{4 - 2\epsilon} \right\} &< \epsilon \\
3\epsilon^2 + 20\epsilon - 48 &< 0
\end{aligned}
\tag{6}
$$

The above inequality is always satisfied since $\epsilon \in (0, 1)$ and $3\epsilon^2 + 20\epsilon - 48$ is always less than $0$ in the allowed range of $\epsilon$ since the roots of the quadratic equation are $-10/3 - 2\sqrt{61}/3 = -8.5$ and $-10/3 + 2\sqrt{61}/3 = 1.87$. Thus $P(T = 0) < \epsilon$ for all values of $\epsilon \in (0, 1)$ completing our proof.

$\square$

| Partial True Graph | True Orientation $(A,B)$ | Predicted Orientation in first two steps $(F,S)$ | | | |
| --- | --- | --- | --- | --- | --- |
| | | $A \leftarrow C \leftarrow B$ | | $A \rightarrow C \rightarrow B$ | |
| | | $P(F,S\|G)$ | $P(T\|F,S,G)$ | $P(F,S\|G)$ | $P(T\|F,S,G)$ |
| 1.   C    A - - - B | no edge
$A \rightarrow B$
$A \leftarrow B$ | $\left(\frac{\epsilon}{2}\right)^2$ | $\epsilon'$
1
$\epsilon'$ | $\left(\frac{\epsilon}{2}\right)^2$ | $\epsilon'$
$\epsilon'$
1 |
| 2.   C → B   A - - - B | no edge
$A \rightarrow B$
$A \leftarrow B$ | $\left(\frac{\epsilon}{2}\right)^2$ | $\epsilon'$
1
$\epsilon'$ | $\left(\frac{\epsilon}{2}\right)(1-\epsilon)$ | $\epsilon'$
$\epsilon'$
1 |
| 3.   C ← B   A - - - B | no edge
$A \rightarrow B$
$A \leftarrow B$ | $\left(\frac{\epsilon}{2}\right)(1-\epsilon)$ | $\epsilon'$
1
$\epsilon'$ | $\left(\frac{\epsilon}{2}\right)^2$ | $\epsilon'$
$\epsilon'$
1 |
| 5.   A → C → B   A - - - B | no edge
$A \rightarrow B$
$A \leftarrow B$ | $\left(\frac{\epsilon}{2}\right)(1-\epsilon)$ | $\epsilon'$
1
$\epsilon'$ | $\left(\frac{\epsilon}{2}\right)(1-\epsilon)$ | $\epsilon'$
$\epsilon'$
1 |
| 6.   A ← C → B   A - - - B | no edge
$A \leftarrow B$ | $(1-\epsilon)^2$ | $\epsilon'$
$\epsilon'$ | $\left(\frac{\epsilon}{2}\right)^2$ | $\epsilon'$
1 |
| 9.   A → C ← B   A - - - B | no edge
$A \rightarrow B$
$A \leftarrow B$ | $\left(\frac{\epsilon}{2}\right)(1-\epsilon)$ | $\epsilon'$
1
$\epsilon'$ | $\left(\frac{\epsilon}{2}\right)(1-\epsilon)$ | $\epsilon'$
$\epsilon'$
1 |

Table A2: Summary of Error Probability for all the partial graphs in Table A1 ($P(F,S|G)$ and $P(T|F,S,G)$): The first column shows different partial graphs from Table A1. The second column then shows different causal relationships that are possible between the nodes $A$ and $B$ for a particular partial graph. Given one *true orientation* between node $A$ and $B$ we get a final ground truth graph. Now we observed in the proof of Proposition 4.1 (see Proof B), that the error probability for the prediction of causal relationship for the pair $(A,B)$ will only change when the $\epsilon$-expert predicts the the structure $A \leftarrow C \rightarrow B$ or $A \rightarrow C \rightarrow B$ for the pair of nodes $(C,A)$ and $(C,B)$ for any ground truth graph. For the rest of the possible predictions of a pair of nodes $(C,A)$ and $(C,B)$ in any ground truth graph, the error probability for $(A,B)$ remains $\epsilon$ ( see Lemma B.2). Thus the third column shows the probability of prediction of structure $A \leftarrow C \rightarrow B$ for a particular true graph and the fourth column shows the probability of making an error in predicting the third causal relationship i.e between $(A,B)$ given the first and second pair $(C,A)$ and $((C,B))$ is already predicted. Similarly, the fifth and sixth columns show the same thing for the predicted structure $A \rightarrow C \rightarrow B$ for each of the ground truth graphs. The partial-graph number $4,7,8$ is not depicted in the table but the entries for $4^{th}$ graph is the same as $2^{nd}$, $7^{th}$ is the same as $3^{rd}$ and $8^{th}$ is same as $6^{th}$ due to symmetry in the partial-structure. The value of $\epsilon' = \frac{\epsilon}{2-\epsilon}$ in $4^{th}$ and $6^{th}$ column is given by renormalized probability Eq. 1 in Lemma B.2.

## C    PRACTICALITY OF THE ADJUSTMENT SET OBTAINED USING THE CAUSAL ORDER

Including variables appearing before treatment (in the causal order) is actually a widespread practice in biomedical and social science empirical studies. In these studies, such variables are called "pre-treatment variables" and a common practice is to condition on all of them. For this reason, we do not think that our proposal is impractical. The importance of Prop. 3.2 is to show the utility of the causal order to identify such a commonly used adjustment set.

For example, refer to the Covariate selection chapter (Sauer et al., 2013) by Sauer, Brookhart, Roy and Vanderwheele in a User Guide ("Developing a Protocol for Observational Comparative Effectiveness Research"). In the section on *"Adjustment for all observed pre-treatment covariates"*, they mention

the widely used propensity score adjustment and write, "The greatest importance is often placed on balancing all pretreatment covariates." They also add that while theoretically colliders can bias the result, "in practice, pretreatment colliders are likely rarer than ordinary confounding variables.".

Further, when unobserved confounding cannot be ruled out (as is the case with most observational studies), evidence is not clear on whether we should include all pre-treatment covariates or select a few, especially because the true graph may be unknown. "Strong arguments exist for error on the side of overadjustment (adjusting for instruments and colliders) rather than failing to adjust for measured confounders (underadjustment). Nevertheless, adjustments for instrumental variables have been found to amplify bias in practice". As the last sentence suggests, note that we are not claiming that adjusting for all pre-treatment variables (variables before treatment in causal order) is always the correct approach; but rather showing that it can be practical in many situations.

Theoretically, of course, improvements to this causal order criterion are possible. Vanderweele and Shpitser (2011) (VanderWeele & Shpitser, 2011) cite the popular practice of using "all pre-treatment variables" and propose the Disjunctive Cause criterion as an improvement. This criterion states that if a pre-treatment variable causes the treatment, outcome, or both; then it should be included in the adjustment set. Note that this criterion—effectively including all pre-treatment ancestors of treatment and/or outcome—is quite close to the causal order-based criterion in our paper. Except for possibly conditioning on a collider in cases where there are unobserved variables in the graph (see Fig. 1 from (VanderWeele & Shpitser, 2011)), additional variables in the causal order adjustment superset will not have a significant effect on the estimate.

## D  ALGORITHMS FOR INTEGRATING CAUSAL ORDER IN EXISTING DISCOVERY METHODS

In continuation to the discussion in Sec 3.2, the algorithms for integrating causal order into existing constraint-based and score-based discovery methods are summarized in Algorithms 1 and 2 respectively.

---

**Algorithm 1** Integrating $\hat{\pi}$ in constraint-based methods

1: **Input:** Noisy expert topological ordering $\hat{\pi}$, Expert $\mathcal{E}$, CPDAG $\hat{\mathcal{G}}$
2: **Output:** Estimated topological order $\hat{\pi}_{\text{final}}$ of $\{X_1, \ldots, X_n\}$.
3: **for** $(i - j) \in$ undirected-edges($\hat{\mathcal{G}}$) **do**
4:     If both nodes $i$ and $j$ are in $\hat{\pi}$ and if $\hat{\pi}_i < \hat{\pi}_j$, orient $(i - j)$ as $(i \to j)$ in $\hat{\mathcal{G}}$.
5:     Otherwise, use expert $\mathcal{E}$ to orient the edge.
6: **end for**
7: $\hat{\pi}_{\text{final}} =$ topological ordering of $\hat{\mathcal{G}}$
8: return $\hat{\pi}_{\text{final}}$

---

**Algorithm 2** Integrating $\hat{\pi}$ in score-based methods

1: **Input:** Dataset $\mathcal{D}$, Variables $\{X_1, \ldots, X_n\}$, Expert $\mathcal{E}$, Score-based method $\mathcal{S}$, *Prior* probability $p$.
2: **Output:** Estimated topological order $\hat{\pi}_{\text{final}}$ of $\{X_1, \ldots, X_n\}$.
3: $\hat{\mathcal{G}} = \mathcal{E}(X_1, \ldots, X_n)$
4: $L =$ level order of $\hat{\mathcal{G}}$
5: **for** cycle $C \in \hat{\mathcal{G}}$ **do**
6:     **for** node $\in C$ **do**
7:         L(node) = min(level(c) $\forall c \in C$)
8:     **end for**
9: **end for**
10: $\hat{\mathcal{G}} = \mathcal{S}(\mathcal{D}, L, p)$ `//L is provided as prior`
11: $\hat{\pi}_{\text{final}} =$ topological ordering of $\hat{\mathcal{G}}$
12: return $\hat{\pi}_{\text{final}}$

---

## E  ADDITIONAL RESULTS

### E.1  STUDY ON DOWNSTREAM TASKS: CAUSAL DISCOVERY

In continuation to the results presented in Sec 5 of the main paper, we present the performance on the causal discovery task across all sample sizes in Table A3. Evidently, as stated in the main paper, the results show that using expert-provided causal order improves $D_{top}$ across our experiments consistently. CaMML+Human/LLM yields benefits even at higher sample sizes. At a sample size of 10000, CaMML's $D_{top}$ for Child and Asia surpasses CaMML+LLM by three and fourfold respectively. In specific datasets like *Survey* where the variables are better understood by humans, incorporating human priors to CaMML leads to consistently zero $D_{top}$, outperforming LLM output.

| | Dataset | PC | SCORE | ICA LiNGAM | Direct LiNGAM | NOTEARS | CaMML | Ours (PC+LLM) | Ours (CaMML+LLM) | Ours (PC+Human) | Ours (CaMML+Human) |
|---|---|---|---|---|---|---|---|---|---|---|---|
| N = 250 | Earthquake | 0.16±0.28 | 4.00±0.00 | 1.00±0.00 | 1.00±0.00 | 1.00±0.00 | 2.00±0.00 | **0.00±0.00** | **0.00±0.00** | **0.00±0.00** | 1.00±0.00 |
| | Cancer | **0.00±0.00** | 3.00±0.00 | 2.00±0.00 | 2.00±0.00 | 2.00±0.00 | 2.00±0.00 | **0.00±0.00** | **0.00±0.00** | **0.00±0.00** | **0.00±0.00** |
| | Survey | 0.50±0.00 | 4.00±0.00 | 2.00±0.00 | 4.00±0.00 | 4.00±0.00 | 3.33±0.94 | **0.00±0.00** | 3.33±0.94 | **0.00±0.00** | **0.00±0.00** |
| | Asia | 2.00±0.59 | 7.00±0.00 | 3.33±0.47 | 1.00±0.00 | 3.00±0.00 | 1.85±0.58 | 1.00±0.00 | 1.00±0.00 | N/A | N/A |
| | Asia-M | 1.50±0.00 | 6.00±0.00 | 1.00±0.00 | 3.00±0.00 | 3.00±0.00 | **1.00±0.00** | **1.00±0.00** | 1.71±0.45 | **1.00±0.00** | 2.00±0.00 |
| | Child | 5.75±0.00 | 12.0±0.00 | 14.33±0.47 | 16.0±0.00 | 14.0±0.00 | **3.00±0.00** | 4.00±0.00 | 3.53±0.45 | N/A | N/A |
| | Neuropathic | 4.00±0.00 | 6.00±0.00 | 13.0±6.16 | 10.0±0.00 | 9.00±0.00 | 10.4±1.95 | **3.00±0.00** | 5.00±0.00 | N/A | N/A |
| N = 500 | Earthquake | 0.75±0.25 | 4.0±0.0 | 1.0±0.0 | 1.0±0.0 | 1.0±0.0 | **0.00±0.00** | **0.00±0.00** | **0.00±0.00** | **0.00±0.00** | **0.00±0.00** |
| | Cancer | 0.16±0.28 | 3.00±0.00 | 3.40±0.48 | 3.00±0.00 | 2.00±0.00 | 1.00±0.00 | 0.33±0.57 | 1.00±0.00 | **0.00±0.00** | **0.00±0.00** |
| | Survey | 1.25±0.00 | 4.00±0.00 | 6.00±0.0 | 6.00±0.00 | 3.40±0.48 | 3.39±0.08 | 1.00±0.00 | 3.33±0.94 | 1.00±0.00 | **0.00±0.00** |
| | Asia | 3.06±0.00 | 5.00±0.00 | 5.60±0.48 | 7.00±0.00 | 3.20±0.39 | 3.81±0.39 | 1.00±0.00 | **0.97±0.62** | N/A | N/A |
| | Asia-M | 2.00±0.00 | 6.00±0.00 | 7.60±0.48 | 5.00±0.00 | 3.80±0.39 | 2.00±0.00 | 1.00±0.00 | 1.33±0.57 | | 3.00±0.00 |
| | Child | 8.09±0.00 | 6.20±1.32 | 12.2±0.74 | 10.6±1.35 | 15.4±0.48 | **2.00±0.00** | 5.00±1.73 | **2.00±0.00** | N/A | N/A |
| | Neuropathic | 7.50±0.00 | 6.00±0.00 | 9.00±1.41 | 13.0±0.00 | 11.0±0.00 | **5.32±0.57** | 8.00±0.00 | 7.49±0.64 | N/A | N/A |
| N = 5000 | Earthquake | 0.50±0.86 | 4.00±0.00 | 2.80±0.39 | 3.00±0.00 | 1.00±0.00 | 0.80±0.97 | **0.00±0.00** | **0.00±0.00** | **0.00±0.00** | **0.00±0.00** |
| | Cancer | 1.33±0.57 | 3.00±0.00 | 3.00±0.00 | 3.00±0.00 | 2.00±0.00 | 2.00±0.00 | 1.33±0.57 | **0.00±0.00** | 1.33±0.57 | **0.00±0.00** |
| | Survey | 2.00±0.00 | 4.00±0.00 | 5.00±0.00 | 5.00±0.00 | 3.00±0.00 | 3.33±0.69 | 2.00±0.00 | 2.60±0.00 | 2.00±0.00 | **0.00±0.00** |
| | Asia | 1.00±0.00 | 4.00±0.00 | 6.60±0.79 | 4.40±1.35 | 3.40±0.48 | 1.75±0.43 | **0.00±0.00** | 0.97±0.62 | N/A | N/A |
| | Asia-M | 2.00±0.00 | 4.00±0.00 | 7.60±0.48 | 4.60±0.48 | 3.20±0.39 | 1.68±0.46 | 2.00±0.00 | **0.00±0.00** | 2.00±0.00 | 2.00±0.00 |
| | Child | 8.25±0.00 | 3.00±0.00 | 12.6±0.79 | 10.8±1.72 | 14.2±0.40 | **3.00±0.00** | 7.00±0.00 | **3.00±0.00** | N/A | N/A |
| | Neuropathic | 8.62±0.00 | 6.00±0.00 | 9.33±0.94 | 10.0±0.00 | 10.0±0.00 | 4.20±0.96 | 9.00±0.00 | **1.23±0.42** | N/A | N/A |
| N = 10000 | Earthquake | **0.00±0.00** | 4.00±0.00 | 3.00±0.00 | 3.00±0.00 | 1.00±0.00 | 0.40±0.48 | **0.00±0.00** | **0.00±0.00** | **0.00±0.00** | **0.00±0.00** |
| | Cancer | 2.00±0.00 | 3.00±0.00 | 3.00±0.00 | 3.00±0.00 | 2.00±0.00 | 0.60±0.80 | 2.00±0.00 | **0.00±0.00** | 2.00±0.00 | **0.00±0.00** |
| | Survey | 2.00±0.00 | 4.00±0.00 | 5.00±0.00 | 5.00±0.00 | 3.00±0.00 | 3.60±1.35 | 2.00±0.00 | 1.83±0.00 | 2.00±0.00 | **0.00±0.00** |
| | Asia | 1.5±0.00 | 4.00±0.00 | 6.00±0.00 | 4.40±1.35 | 3.00±0.00 | 1.40±0.48 | **0.00±0.00** | 0.34±0.47 | N/A | N/A |
| | Asia-M | 1.00±0.00 | 4.00±0.00 | 8.00±0.00 | 4.80±0.39 | 3.00±0.00 | 2.00±0.00 | **0.00±0.00** | **0.00±0.00** | **0.00±0.00** | 3.00±0.00 |
| | Child | 6.00±3.04 | 3.00±0.00 | 12.2±1.46 | 11.6±0.48 | 14.4±0.48 | 2.80±0.84 | 5.00±2.64 | **1.00±0.00** | N/A | N/A |
| | Neuropathic | 10.00±0.00 | 6.00±0.00 | **1.00±0.00** | 10.0±0.00 | 10.0±0.00 | 3.00±0.00 | 10.00±0.00 | **1.00±0.00** | N/A | N/A |

Table A3: Comparison with causal discovery methods, showing mean and std dev of $D_{top}$ over 3 runs. (For the Neuropathic subgraph (1k samples), PC Algorithm returns cyclic graphs in the MEC). Human experiments not conducted for Neuropathic, Child (due to feasibility issues) and Asia; hence rows marked as N/A.

## E.2 $D_{top}$ VS SHD: BETTER MEASURE OF EFFECT ESTIMATION ERROR

As discussed in Sec 3.2 of the main paper, we show herein that $D_{top}$ has a strong correlation with effect estimation error and hence is a valid metric for effect inference.

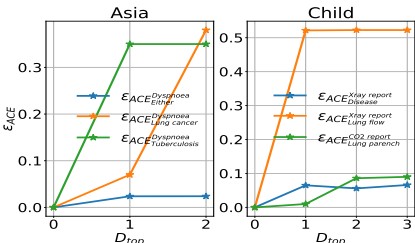

Figure A2: $D_{top}$ vs. $\epsilon_{ACE}$. $\epsilon_{ACE}$ increases as $D_{top}$ increases, aligning with theoretical observations.

| Cancer | | | | |
|---|---|---|---|---|
| $D_{top} = 0$ | | | $SHD = 2$ | |
| $SHD$ | $\epsilon_{ACE}$ | | $D_{top}$ | $\epsilon_{ACE}$ |
| 0 | 0.00 | | 0 | 0.00 |
| 2 | 0.00 | | 1 | 0.25 |
| 4 | 0.00 | | 2 | 0.50 |

| Asia | | | | |
|---|---|---|---|---|
| $D_{top} = 0$ | | | $SHD = 3$ | |
| $SHD$ | $\epsilon_{ACE}$ | | $D_{top}$ | $\epsilon_{ACE}$ |
| 0 | 0.00 | | 1 | 0.14 |
| 6 | 0.00 | | 2 | 0.22 |
| 10 | 0.00 | | 3 | 0.57 |

| Survey | | | | |
|---|---|---|---|---|
| $D_{top} = 0$ | | | $SHD = 2$ | |
| $SHD$ | $\epsilon_{ACE}$ | | $D_{top}$ | $\epsilon_{ACE}$ |
| 0 | 0.00 | | 0 | 0.00 |
| 2 | 0.00 | | 1 | 0.25 |
| 4 | 0.03 | | 2 | 0.50 |

Table A4: $\epsilon_{ACE}$ vs $SHD$ given $D_{top}$ (& $D_{top}$ given $SHD$)

### E.3 LLMs USED IN POST PROCESSING FOR GRAPH DISCOVERY

We conducted some experiments where we utilised discovery algorithms like PC for creating skeletons of the graph and employed LLMs for orienting the undirected edges. The idea was to utilise LLMs ability to correctly estimate the causal direction while leveraging PC algorithm's ability to give a skeleton which could be oriented in a post processing setup. We saw that LLM ended up giving improved results as compared to PC alone.

### E.4 TRIPLET VS PAIRWISE QUERY STRATEGIES

In continuation to the discussion in Sec 5 of the main paper, we include Tables A5 for more details. The pairwise strategy also shows flaws when LLMs are used as noisy experts. In many cases, pairwise querying yields cycles due to which $D_{top}$ cannot be computed. In particular, for the Child dataset with 20 nodes, pairwise querying of LLMs yields an extremely high number of cycles (see Table A5). LLM output tends to overconnect, resulting in high SHD. Overall, among the prompting strategies, the chain of thought prompt performs the best: it has the lowest number of cycles for Child and Neuropathic datasets. This indicates that in-context examples and chain-of-thought reasoning help to increase the accuracy of causal order output, but other contextual cues do not matter.

| Dataset | $D_{top}$ | SHD | IN/TN | Cycles |
|---|---|---|---|---|
| Base Prompt | | | | |
| Earthquake | 0 | 7 | 0/5 | 0 |
| Cancer | 0 | 6 | 0/5 | 0 |
| Survey | 3 | 12 | 0/6 | 0 |
| Asia | - | 21 | 0/8 | 1 |
| Asia-M | - | 15 | 0/7 | 7 |
| Child | - | 177 | 0/20 | >>3k |
| Neuropathic | - | 212 | 0/22 | >>5k |
| All Directed Edges | | | | |
| Earthquake | 1 | 9 | 0/5 | 0 |
| Cancer | 1 | 7 | 0/5 | 0 |
| Survey | 2 | 11 | 0/6 | 0 |
| Asia | - | 21 | 0/8 | 6 |
| Asia-M | 0 | 13 | 0/7 | 0 |
| Child | - | 139 | 0/20 | >>300 |
| Neuropathic | - | 194 | 0/22 | >>1k |
| One Hop Iteration | | | | |
| Earthquake | 0 | 8 | 0/5 | 0 |
| Cancer | 0 | 6 | 0/5 | 0 |
| Survey | 3 | 12 | 0/6 | 0 |
| Asia | - | 21 | 0/8 | 1 |
| Asia-M | 0 | 14 | 0/7 | 0 |
| Child | - | 167 | 0/20 | >>400 |
| Neuropathic | - | 204 | 0/22 | >>4k |

Table A5: Comparison of various querying strategies for only LLM-based setups, providing different contextual cues in each setup about the graph. IN: Isolated Nodes, TN:Total Nodes.

| Dataset | $D_{top}$ | SHD | IN/TN | Cycles |
|---|---|---|---|---|
| Chain of Thought | | | | |
| Earthquake | 0 | 4 | 0/5 | 0 |
| Survey | 1 | 9 | 2/6 | 0 |
| Asia | - | 18 | 0/8 | 1 |
| Asia-M | - | 13 | 0/7 | 1 |
| Child | - | 138 | 0/20 | >>500 |
| Neuropathic | - | 64 | 0/22 | 5 |
| Triplet Query | | | | |
| Earthquake | 0 | 4 | 0/5 | 0 |
| Cancer | 1 | 6 | 0/5 | 0 |
| Survey | 0 | 9 | 0/6 | 0 |
| Asia | 1 | 14 | 0/8 | 0 |
| Asia-M | 1 | 11 | 0/7 | 0 |
| Child | - | 138 | 0/20 | 391 |
| Child (+ Cycle Remover) | 1 | 28 | 10/20 | 0 |
| Neuropathic | - | 151 | 0/22 | 772 |
| Neuropathic(+ Cycle remover) | 3 | 24 | 13/20 | 0 |

Table A6: Triplet query output using variable names with their descriptions (Cancer not included since CoT prompt has examples from this graph). IN: Isolated Nodes, TN:Total Nodes. Since calculating total number of cycles in a DAG is computationally challenging (NP Hard), we find a lower bound of cycles present in each graph based on total k lenght cycles in each setting, where k=5. If k is scaled up, the number of such unique cycles in the LLM output will also scale significantly. Lower bound helps us make a comparison with number of cycles in outputs like in Triplet strategy, where numbers are comparatively smaller and can be calculated easily.

| Dataset | SHD | $D_{top}$ | Cycles | IN/TN |
|---|---|---|---|---|
| Base Prompt | | | | |
| Asia | 18 | 1 | 0 | 0 |
| Child | 148 | - | >>10k | 0 |
| Earthquake | 4 | 0 | 0 | 0 |
| Survey | 7 | - | 1 | 0 |
| Neuropathic | 178 | - | >>10k | 0 |
| Covid | 33 | - | 15 | 0 |
| Alzheimers | 30 | - | 1 | 0 |

Table A7: Final result of using performing base pairwise querying strategy with GPT-4. These results show how using a superior model in pairwise querying does not lead to complete removal of cycles, further highlighting the impact of triplet strategy.

| Dataset | Metric | Pairwise (Base) | Pairwise (CoT) | Triplet |
|---|---|---|---|---|
| | | Using Phi-3 | | |
| Asia | $D_{top}$ | - | 4 | **0** |
| | SHD | 17 | 11 | **13** |
| | Cycles | 1 | **0** | **0** |
| | IN/TN | **1/8** | **0/8** | 1/8 |
| Alzheimers | $D_{top}$ | - | - | 7 |
| | SHD | 28 | 28 | **25** |
| | Cycles | 11 | 11 | **0** |
| | IN/TN | **0/11** | **0/11** | 0/11 |
| Child | $D_{top}$ | - | - | 17 |
| | SHD | 142 | 80 | **69** |
| | Cycles | »10k | 59 | **0** |
| | IN/TN | **0/20** | **0/20** | 0/20 |
| | | Using Llama3 | | |
| Asia | $D_{top}$ | - | - | 2 |
| | SHD | 22 | 23 | **17** |
| | Cycles | 71 | 20 | **0** |
| | IN/TN | **0/8** | **0/8** | 0/8 |
| Alzheimers | $D_{top}$ | - | - | 5 |
| | SHD | 41 | 29 | **24** |
| | Cycles | 1144 | 7 | **0** |
| | IN/TN | 1/11 | **0/11** | 1/11 |
| Child | $D_{top}$ | - | - | 12 |
| | SHD | 167 | 151 | **129** |
| | Cycles | »10k | 71 | **0** |
| | IN/TN | **0/20** | **0/20** | 0/20 |

Table A8: *(Top)* Results using Phi-3 *(Bottom)* Performance of triplet method using Llama3 (8b) models vs CoT pairwise vs base pairwise query strategy on multiple benchmark datasets across diff metrics: $D_{top}$, SHD, (Num of) Cycles, IN (Isolated Nodes), TN (Total Nodes). When num of cycles>0, $\hat{\pi}$ cannot be computed, hence $D_{top}$ is given by '-'. Triplet consistently outperforms the pairwise (base as well as CoT) strategy across metrics & datasets, especially by significant amounts on larger graphs like *Child*.

| Graphs | Dtop | SHD | Cycles | Isolated Nodes | LLM Calls | Number of Nodes | Complexity |
|---|---|---|---|---|---|---|---|
| **Quadruplet** | | | | | | | |
| Asia | 1 | 6 | 0 | 0 | 70 | 8 | $O(n^3)$ |
| Covid | 1 | 19 | 0 | 0 | 330 | 11 | $O(n^3)$ |
| Alzheimers | 5 | 14 | 0 | 0 | 330 | 11 | $O(n^3)$ |
| **Triplet** | | | | | | | |
| Asia | 1 | 14 | 0 | 0 | 286 | 8 | $O(n^4)$ |
| Covid | 0 | 30 | 0 | 0 | 165 | 11 | $O(n^4)$ |
| Alzheimers | 4 | 28 | 0 | 0 | 165 | 11 | $O(n^4)$ |

Table A9: Analyzing the performance differences between using triplets and quadruplets, we found no significant difference in the quality of the final graph output. However, the number of LLM API calls more than doubles when shifting from triplets to quadruplets, leading to a substantial increase in cost.

| Dataset | Metric | Triplet GPT-4 | Triplet GPT-3.5-Turbo |
|---|---|---|---|
| Asia | $D_{top}$ | 0 | 1 |
| | SHD | 10 | 14 |
| | Cycles | 0 | 0 |
| | IN/TN | 0/8 | 0/8 |
| Alzheimers | $D_{top}$ | 4 | 4 |
| | SHD | 23 | 28 |
| | Cycles | 0 | 0 |
| | IN/TN | 0/11 | 0/11 |
| Child | $D_{top}$ | 1 | 1 |
| | SHD | 24 | 28 |
| | Cycles | 0 | 0 |
| | IN/TN | 6/20 | 10/20 |

Table A10: Results of running GPT-4 for orienting triplet subgraphs, and then re-using GPT-4 for resolving clashes during merging phase. These results cover graph discovery on Asia, Alzheimers and Child graphs. Upgrading to a superior model (GPT-4) leads to better results for all three graphs on triplet strategy.

| Graph | Sample size | Before LLM prior | After LLM Prior |
|---|---|---|---|
| **Child** | 250 | 18 | 16 |
| | 500 | 16 | 15 |
| | 1000 | 14 | 13 |
| | 5000 | 13.5 | 12 |
| | 10000 | 9.66 | 6 |
| **Earthquake** | 250 | 3.83 | 3 |
| | 500 | 3.6 | 3 |
| | 1000 | 3.6 | 3 |
| | 5000 | 1.16 | 0.66 |
| | 10000 | 0 | 0 |
| **Cancer** | 250 | 1 | 0 |
| | 500 | 3.83 | 3.83 |
| | 1000 | 2.6 | 2.6 |
| | 5000 | 2.3 | 2.3 |
| | 10000 | 2 | 2 |
| **Asia** | 250 | 7.5 | 7 |
| | 500 | 6 | 5 |
| | 1000 | 7 | 7 |
| | 5000 | 2 | 1 |
| | 10000 | 2 | 1 |
| **Asia-M** | 250 | 4.5 | 4 |
| | 500 | 4 | 4 |
| | 1000 | 5.5 | 5 |
| | 5000 | 4 | 4 |
| | 10000 | 4 | 4 |
| **Neuropathic** | 250 | 27 | 26 |
| | 500 | 31 | 29 |
| | 1000 | 41 | 40 |
| | 5000 | 55 | 53 |

Table A11: Comparison of SHD Values Before and After Incorporating LLM Priors Using the PC Algorithm Across Various Graphs

Finally, the triplet prompt provides the most accurate causal order. For small-scale graphs, it produces no cycles and consistently produces minimal $D_{top}$ (ranging from 0 to 1) while also producing no isolated nodes. Even for medium-size graphs like Child and Neuropathic, the LLM output includes significantly fewer cycles than the pairwise strategy, which were removed leading to a significant and accurate causal order used further as prior. That said, we do see that isolated nodes in the output increase after cycles are removed for medium graphs (all graphs are connected, so outputting an isolated node is an error). Considering LLMs as virtual experts, this indicates that there are some nodes on which the LLM expert cannot determine the causal order. This is still a better tradeoff than providing the wrong causal order, which can confuse downstream algorithms. Overall, we conclude that the triplet query strategy provides the most robust causal order predictions. Additional results showing the error introduced by the LLM with respect to a ground truth order are shown in two different settings in Tables A15 and A16.

| Dataset | Dtop | SHD | IN | Cycles |
|---------|------|-----|-----|--------|
| Alzheimers | 5 | 14 | 0 | 0 |
| Covid | - | 36 | 0 | 1 |

Table A12: Results of a hybrid approach where the PC algorithm integrates an LLM-derived prior (GPT-4) obtained via BFS for Alzheimer's and COVID graphs. The prior directly provides edge orientations, which guide the initial graph structure, while PC subsequently orients remaining edges. Unlike triplet that used only causal order, this approach incorporates the full graph as a prior. The PC algorithm is further supported by a large observational dataset of 10,000 samples. The results show that PC + BFS (GPT-4) is also outperformed by Triplet method (GPT-3.5). Specifically, PC+BFS yields 1 cycle and higher SHD on Covid dataset. On the Alzheimers dataset, PC+BFS is comparable: it yields higher Dtop but a lower SHD.

| Dataset | Samples | LLM | Ground Truth | PC (Average over MEC) |
|---------|---------|-----|--------------|-----------------------|
| Asia | 250 | $1.00\pm0.00$ | $0.00\pm0.00$ | $2.00\pm0.00$ |
|  | 1000 | $3.00\pm0.00$ | $2.00\pm0.00$ | $3.00\pm0.00$ |
|  | 10000 | $3.00\pm0.00$ | $3.00\pm0.00$ | $3.00\pm0.00$ |
| Child | 250 | $5.00\pm0.00$ | $5.00\pm0.00$ | $6.50\pm0.00$ |
|  | 1000 | $6.00\pm0.00$ | $6.00\pm0.00$ | $8.43\pm0.00$ |
|  | 10000 | $9.00\pm0.00$ | $9.00\pm0.00$ | $9.75\pm0.00$ |

Table A15: Comparing $D_{top}$ of final graph using LLM order vs Ground truth order as prior to PC algorithm for Child and Asia graph, averaged over 4 runs

| | 1000 samples | | | |
|---|---|---|---|---|
| Context | Base prompt | Past iteration orientations | Markov Blanket | PC (Avg. over MEC) |
| $D_{top}$ | 8.0 | 5.3 | 6.6 | 9.61 |
| SHD | 14.33 | 12.66 | 14.0 | 17.0 |
| | 10000 samples | | | |
| $D_{top}$ | 6.33 | 9.66 | 6.0 | 7.67 |
| SHD | 9.0 | 13.33 | 8.33 | 12.0 |

Table A13: PC + LLM results where LLM is used to orient the undirected edges of the skeleton PC returns over different data sample sizes. We show how LLMs can be used in a post processing setup for edge orientation besides having the capability of acting as a strong prior for different discovery algorithms.

| Dataset | Samples | $\epsilon_{ATE}(S_1)$ | $\epsilon_{ATE}(S_2)$ | $\epsilon_{ATE}(S_3)$ | $\Delta_{12}$ | $\Delta_{13}$ |
|---------|---------|------------------------|------------------------|------------------------|---------------|---------------|
| Asia | 250 | $0.70\pm0.40$ | $0.70\pm0.39$ | $0.69\pm0.39$ | $0.00\pm0.00$ | $0.00\pm0.00$ |
|  | 500 | $0.64\pm0.39$ | $0.64\pm0.39$ | $0.64\pm0.38$ | $0.00\pm0.00$ | $0.00\pm0.00$ |
|  | 1000 | $0.59\pm0.32$ | $0.59\pm0.32$ | $0.59\pm0.32$ | $0.00\pm0.00$ | $0.00\pm0.00$ |
|  | 5000 | $0.59\pm0.30$ | $0.59\pm0.30$ | $0.59\pm0.29$ | $0.00\pm0.00$ | $0.00\pm0.00$ |
|  | 10000 | $0.49\pm0.00$ | $0.49\pm0.00$ | $0.49\pm0.00$ | $0.00\pm0.00$ | $0.00\pm0.00$ |

Table A16: Results on Asia dataset. Here we test the difference in the estimated causal effect of *lung* on *dyspnoea* when the causal effect is estimated using the backdoor set $S_1$ = *{smoke}* vs. the causal effect estimated when all variables that precede treatment variable in two possible topological orders as backdoor sets: $S_2$ = *{asia, smoke}*, $S_2$= *{asia, tub, smoke}*. $\Delta_{12}, \Delta_{13}$ refers to the absolute difference between the pairs $\epsilon_{ATE}(S_1), \epsilon_{ATE}(S_2)$ and $\epsilon_{ATE}(S_1), \epsilon_{ATE}(S_3)$ respectively. From the last two columns, we observe that using the variables that come before the treatment node in a topological order as a backdoor set does not result in the deviation of causal effects from the ground truth effects.

| Dataset | BFS | | | | | | | | BFS + Statistics | | | | | | | |
|---|---|---|---|---|---|---|---|---|---|---|---|---|---|---|---|---|
| | GPT-3.5 | | | | GPT-4 | | | | GPT-3.5 | | | | GPT-4 | | | |
| | $D_{top}$ | SHD | IN | Cyc | $D_{top}$ | SHD | IN | Cyc | $D_{top}$ | SHD | IN | Cyc | $D_{top}$ | SHD | IN | Cyc |
| Asia | 2 | 7 | **0** | **0** | **0** | **1** | **0** | **0** | - | 23 | 0 | 33 | **0** | 3 | **0** | **0** |
| Alzh. | 5 | 17 | 2 | **0** | **0** | 34 | **0** | **0** | - | 27 | 1 | 17 | - | **14** | **0** | 1 |
| Child | - | 40 | **0** | 6 | 11 | 30 | **0** | **0** | - | 52 | 2 | 21 | 2 | 27 | 4 | **0** |
| Covid | - | 28 | **0** | 4 | 5 | **20** | **0** | **0** | - | 30 | **0** | 15 | - | 32 | 1 | 10 |

Table A17: Comparison of BFS and BFS+Statistics approaches using GPT-3.5-turbo and GPT-4. Datasets used: Asia, Alzheimers, Child, Covid. Metrics: topological distance ($D_{top}$), structural hamming distance (SHD), Isolated Nodes (IN), and cycle count (Cyc).

# F    QUERY STRATEGIES: MORE DETAILS AND EXAMPLES

As stated in Sec. E.4, we follow earlier efforts in studying pairwise query strategies in our experiments. Beyond the basic query strategy, we also study its augmentation with additional contextual information. In summary, we study four types of pairwise queries, which we describe below.

- **Basic prompt.** This is the simplest technique. We directly ask the expert to find the causal direction between a given pair of variables (Kıcıman et al., 2023).

- **Chain-of-Thought (+ In-context Learning).** Based on encouraging results of providing in-context examples in prompts for various LLM tasks (Brown et al., 2020), we include 3 examples of the ordering task that we expect the expert to perform on. Effectively, we provide example node pairs with their correct causal ordering before asking the question about the given nodes. Each example answer also contains an explanation of the answer, generated using a high-cost expert (GPT-4, in our experiments). Adding the explanation provides the expert with additional reasoning information when deciding the causal order (Wei et al., 2022). To avoid overfitting, we select node pairs from graphs that are not evaluated in our study, as additional input. Node pairs with and without direct edges were equally chosen for this purpose. Examples of an expert's (LLM's in this case) answers (and their explanations) using this query strategy are shown in tables below.

- **Iterative Context.** Here, we provide previously oriented pairs as context in the prompt. Since the expert has access to its previous decisions, we expect that it may avoid creating cycles through its predictions.

- **One hop iterative Context.** Providing previously oriented pairs may become prohibitive for large graphs. Here we provide the information of connections with neighbouring nodes of the node pair being inspected as additional context in the query.

**Cost Estimation Analysis: Pairwise vs. Triplet for LLMs**
Triplet method ensures scalability by optimizing most calls to a cheaper and smaller model (like GPT-3.5-Turbo) while improving performance. The triplet pipeline boosts accuracy through multiple context switches (varying the third node) for better pairwise orientation. Strategic use of GPT-4 for conflict resolution enhances effectiveness and controls costs. For a 100-node graph, pairwise orientation using GPT-4 costs an estimated $574, while our triplet strategy, leveraging both GPT-4 and GPT-3.5-Turbo, reduces costs to $55. Although our triplet method involves more calls, it optimally uses GPT-4 for error correction, significantly improving performance while keeping costs low.

**Tradeoff Between Increased Nodes: Gains vs. Complexity-Driven Errors** As we increase the number of nodes in the prompt, there is a tradeoff: Adding more nodes provides more context and thus is beneficial, but more nodes in the LLM's prompt can also lead to higher error and higher computational cost. Therefore, we tackled this question empirically by comparing pairwise, triplet, and quadruplet-based prompts. As Table A9 shows, using a quadruplet prompt slightly increases accuracy but leads to a significant increase in the number of LLM calls. In contrast, the increase in accuracy (especially cycle avoidance) is substantial when moving from pairwise to the triplet method. Given these considerations, we decided to go with the Triplet prompt, as it allows for adding more context with minimal increase in prompt complexity and total number of LLM calls. Note that future iterations of language models might be able to handle longer context better with more improvements, therefore the  will vary with model size, architecture and data the model is trained on. Since we don't have the information about this, it will be difficult to model  accurately. However, with the LLMs that we have tried (GPT-4, GPT-3.5, Phi-3 and LLama3), we do not see an increased error when using the triplet prompt compared to the pairwise prompt.

# G    CAUSAL GRAPHS USED IN EXPERIMENTS

Figures A3-A7 show the causal graphs and details we considered from BNLearn repository (Scutari & Denis, 2014).

| Dataset | Graph | Data for Variables |
|---------|-------|--------------------|
| BN Learn Datasets (Asia, Cancer, Earthquake, Survey, Child) | Real-world graphs from scientific studies | Synthetic data generation based on bnlearn library |
| Neuropathic Pain | Real-world graph constructed with consensus from medical experts Tu et al. (2019). Includes domain-specific variables as Right L1 Radiculopathy, Topical Dysfunction, DLS L5-S1, etc. (see Fig. A8) | Synthetic data generation based on Tu et al. (2019) |
| Alzheimers Dataset | Real-world graph constructed with consensus from medical experts Abdulaal et al. (2023). Constructed in 2023, after the training cutoff date of GPT-3.5 and GPT-4 models used. | No data is available |
| Covid-19 Dataset | Real-world graph constructed by experts to understand effect of Covid-19 on respiratory system Mascaro et al. (2023). Constructed in 2022, after the training cutoff date of GPT-3.5 and GPT-4 models used. | No data is available. |

Table A18: Details about the datasets used for evaluation.

| Dataset | Number of Nodes | Number of Edges | Description (used as a context) |
|---------|-----------------|-----------------|---------------------------------|
| Asia | 8 | 8 | Model the possible respiratory problems someone can have who has recently visited Asia and is experiencing shortness of breath |
| Cancer | 5 | 4 | Model the relation between various variables responsible for causing Cancer and its possible outcomes |
| Earthquake | 5 | 5 | Model factors influencing the probability of a burglary |
| Survey | 6 | 6 | Model a hypothetical survey whose aim is to investigate the usage patterns of different means of transport |
| Child | 20 | 25 | Model congenital heart disease in babies |
| Neuropathic Pain Diagnosis (subgraph) | 22 | 25 | For neuropathic pain diagnosis |

Table A19: Overview of datasets used

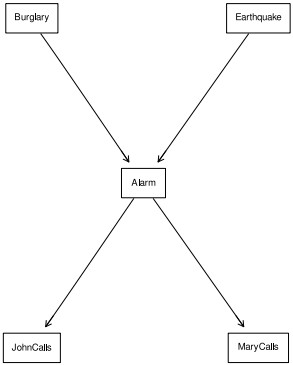

Figure A3: Earthquake Bayesian network. Abbreviations/Descriptions: Burglary: *burglar entering*, Earthquake: *earthquake hitting*, Alarm: *home alarm going off in a house*, JohnCalls: *first neighbor to call to inform the alarm sound*, Marycalls: *second neighbor to call to inform the alarm sound*.

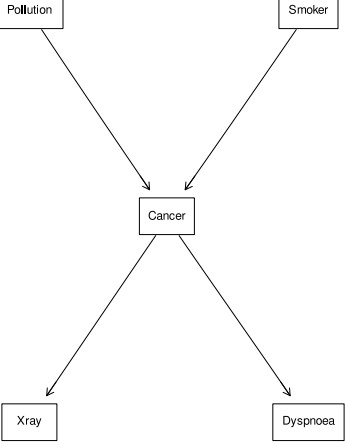

Figure A4: Cancer Bayesian network. Abbreviations/Descriptions: Pollution: *exposure to pollutants*, Smoker: *smoking habit*, Cancer: *Cancer*, Dyspnoea: *Dyspnoea*, Xray: *getting positive xray result*.

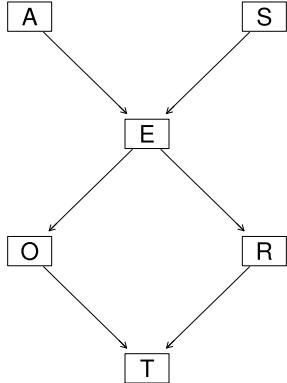

Figure A5: Survey Bayesian network. Abbreviations: A=*Age/Age of people using transport*, S=*Sex/male or female*, E=*Education/up to high school or university degree*, O=*Occupation/employee or self-employed*, R=*Residence/the size of the city the individual lives in, recorded as either small or big*, T=*Travel/the means of transport favoured by the individual*.

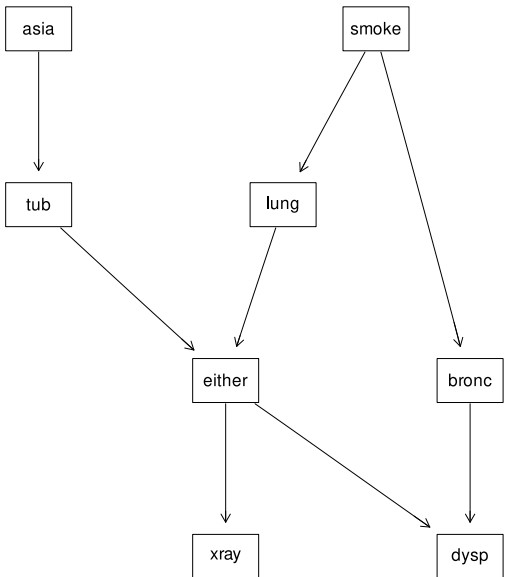

Figure A6: Asia Bayesian network. Abbreviations/Descriptions: asia=*visit to Asia/visiting Asian countries with high exposure to pollutants*, smoke=*smoking habit*, tub=*tuberculosis*, lung=*lung cancer*, either=*either tuberculosis or lung cancer*, bronc=*bronchitis*, dysp=*dyspnoea*, xray=*getting positve xray result*.

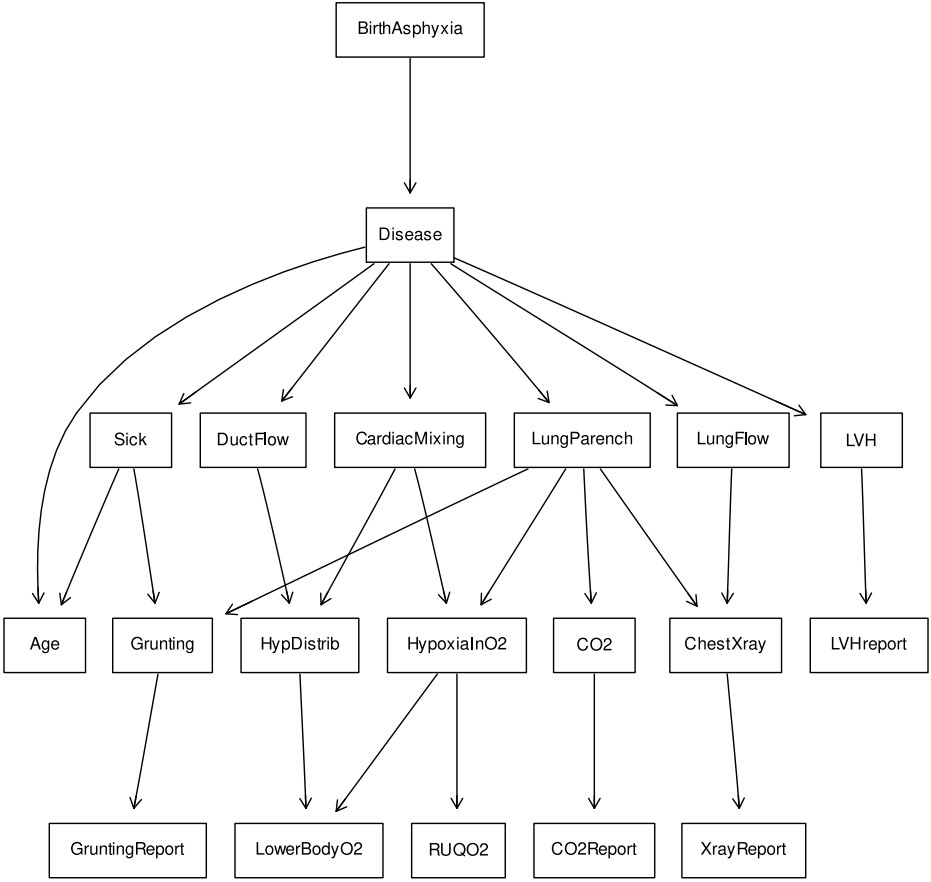

Figure A7: Child Bayesian network. Abbreviations: BirthAsphyxia: *Lack of oxygen to the blood during the infant's birth*, HypDistrib: *Low oxygen areas equally distributed around the body*, HypoxiaInO2: *Hypoxia when breathing oxygen*, CO2: *Level of carbon dioxide in the body*, ChestXray: *Having a chest x-ray*, Grunting: *Grunting in infants*, LVHreport: *Report of having left ventricular hypertrophy*, LowerBodyO2: *Level of oxygen in the lower body*, RUQO2: *Level of oxygen in the right upper quadricep muscle*, CO2Report: *A document reporting high levels of CO2 levels in blood*, XrayReport: *Report of having a chest x-ray*, Disease: *Presence of an illness*, GruntingReport: *Report of infant grunting*, Age: *Age of infant at disease presentation*, LVH: *Thickening of the left ventricle*, DuctFlow: *Blood flow across the ductus arteriosus*, CardiacMixing: *Mixing of oxygenated and deoxygenated blood*, LungParench: *The state of the blood vessels in the lungs*, LungFlow: *Low blood flow in the lungs*, Sick: *Presence of an illness*

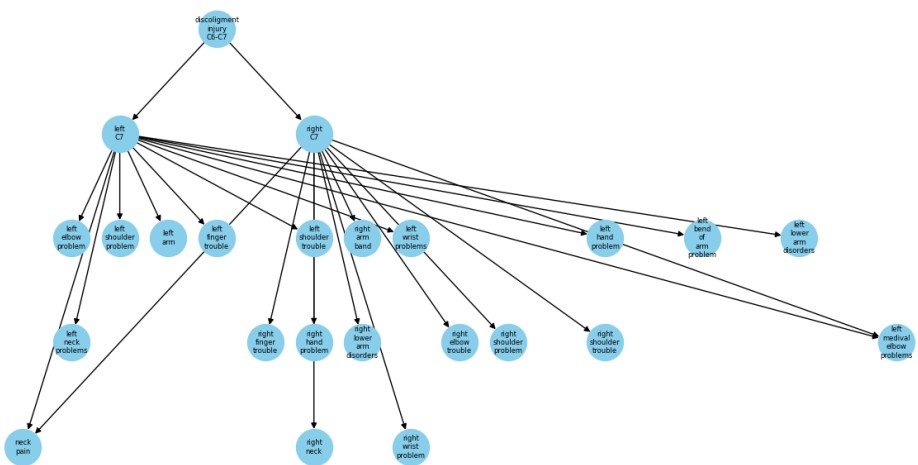

Figure A8: For Neuropathic dataset, we consider a sub-graph induced by one of the root nodes, containing the following 22 nodes and corresponding edges taken from https://observablehq.com/@turuibo/the-complete-causal-graph-of-neuropathic-pain-diagnosis: 'right C7', 'right elbow trouble', 'left shoulder trouble', 'left bend of arm problem', 'right shoulder trouble', 'right hand problem', 'left medival elbow problems', 'right finger trouble', 'left neck problems', 'left wrist problems', 'left shoulder problem', 'right neck', 'right wrist problem', 'right shoulder problem', 'discoligment injury C6 C7', 'left hand problem', 'left C7', 'right arm band', 'left lower arm disorders', 'neck pain', 'left finger trouble', 'left arm'. We did not use descriptions for the nodes of Neuropathic graph.

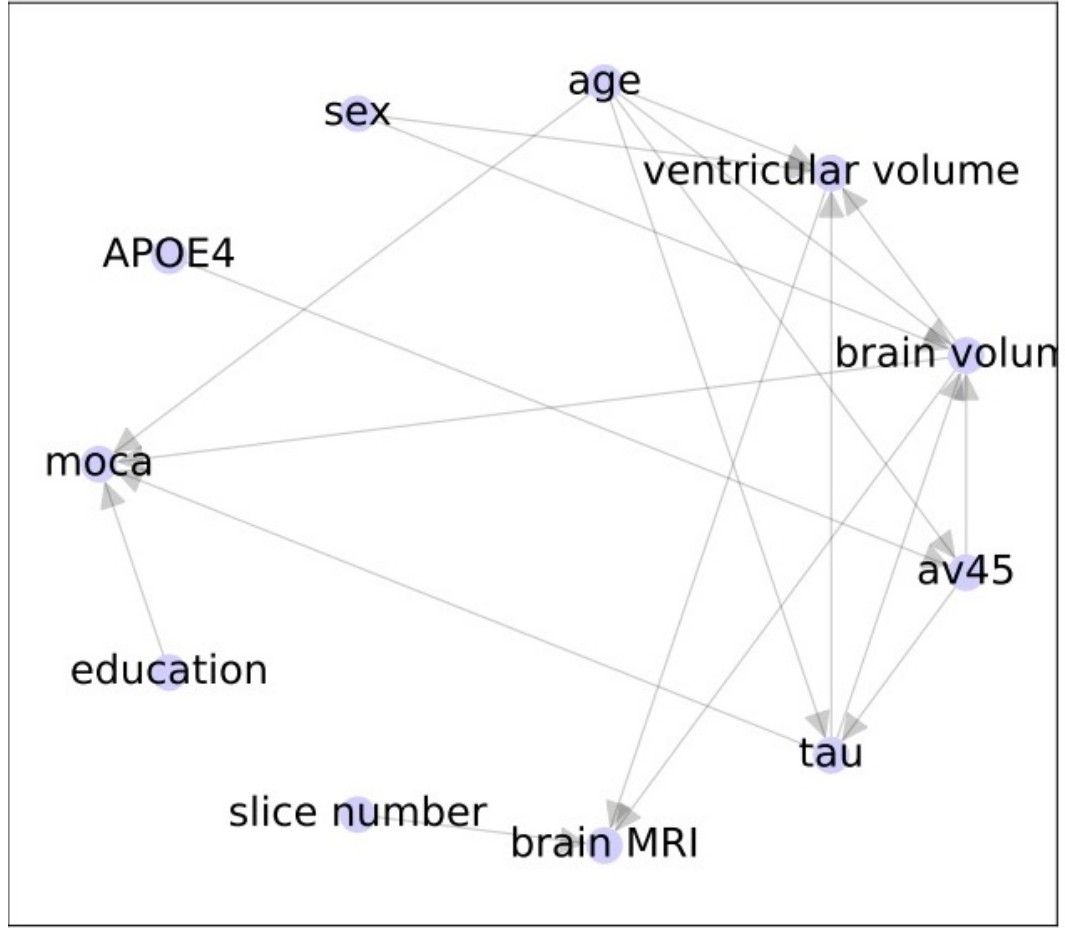

Figure A9: The **Alzheimer's dataset** is a Bayesian Network developed by Abdulaal, Ahmed, et al. in collaboration with five domain experts, as detailed in their paper "Causal Modelling Agents: Causal Graph Discovery through Synergising Metadata-and Data-driven Reasoning" (ICLR 2024). The dataset includes the following variables: **age**, which represents the age of the patient; **sex**, indicating the biological sex of the patient; **APOE4**, which measures the expression level of the APOE4 gene; **education**, reflecting the patient's educational attainment in years; **av45**, measuring the beta amyloid protein level using Florbetapir F 18; **tau**, indicating phosphorylated-tau deposition; **brain volume**, representing the total brain matter volume of the patient; **Ventricular Volume**, indicating the total ventricular volume of the patient; and **moca**, which is the Montreal Cognitive Assessment Score.

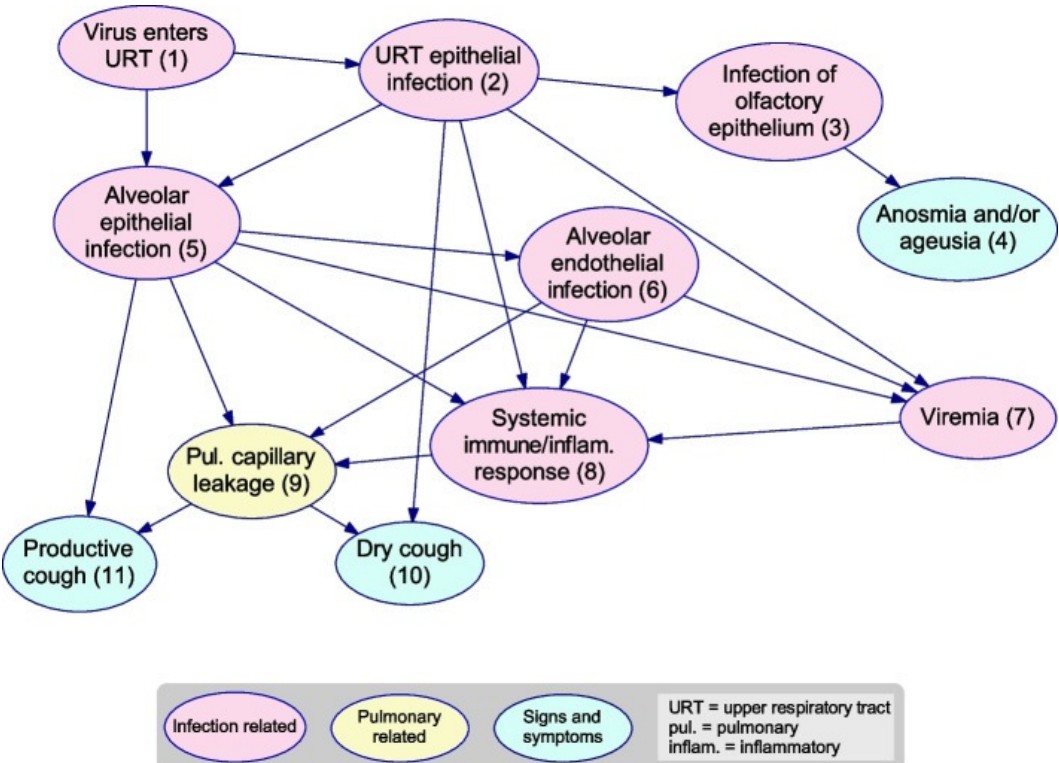

Figure A10: **Respiratory causal DAG** taken from Mascaro S, Wu Y, Woodberry O, et al. Modeling COVID-19 disease processes by remote elicitation of causal Bayesian networks from medical experts. BMC Med Res Methodol. Here, **Virus enters upper respiratory tract (URT)**: SARS-CoV-2 viral particles inhaled and attach to upper respiratory tract mucosal surface. The size of the viral inoculum is dependent on exposure related factors, not included in the current model, **Upper respiratory tract (URT) epithelial infection**: Viral infection of upper respiratory tract epithelial cells $+/-$ signaling an immune response and leading to local inflammation, **Infection of olfactory epithelium**: Viral infection of the olfactory epithelial cells $+/-$ leading to impaired olfaction, **Ageusia and/or anosmia**: Loss of the ability to taste and/or smell, **Alveolar epithelial infection**: Viral infection of the alveolar cells, $+/-$ inducing an immune response which leads to local inflammation. **Alveolar endothelial infection**: Viral infection of the endothelial cells of the capillaries of the terminal airways, $+/-$ inducing an immune response which leads to local inflammation., **Viremi**: Presence of SARS-CoV-2 in blood allowing for systemic dissemination of the virus., **Systemic immune/inflammatory (inflam.) response**: Activation of innate and/or adaptive immune system by the presence of virus at one or more body site/s. Manifest by the release of pro- +/- anti-inflammatory markers in blood by immune-related cells, **Pulmonary capillary leakage**: Leakage of plasma from the pulmonary capillaries into the terminal airways., leakage **Dry cough**: Cough without the production of mucus or phlegm., **Productive cough**: Cough with the production of mucus or phlegm.

Question: For a causal graph used to model relationship of various factors and outcomes related to cancer with the following nodes: ['Pollution', 'Cancer', 'Smoker', 'Xray', 'Dyspnoea'],
Which cause-and-effect relationship is more likely between nodes 'smoker' and 'cancer'?

A. changing the state of node 'smoker' causally affects a change in another node 'cancer'.

B. changing the state of node 'cancer' causally affects a change in another node 'smoker'.

C. There is no causal relation between the nodes 'cancer' and 'smoker'.

Make sure to first provide a grounded reasoning for your answer and then provide the answer in the following format: ⟨Answer⟩A/B/C⟨/Answer⟩. It is very important that you output the final Causal graph within the tags like ⟨Answer⟩A/B/C⟨/Answer⟩ otherwise your answer will not be processed.

Answer: The causal effect of "smoker" directing to "cancer" is based on the strong evidence from epidemiological studies linking smoking to an increased risk of developing cancer. Smoking introduces harmful substances into the respiratory system, leading to cellular damage and mutation, which significantly raises the likelihood of cancer development in the lungs or respiratory tract, subsequently impacting the occurrence of respiratory problems like shortness of breath. Therefore answer is ⟨Answer⟩A⟨/Answer⟩

Question: For a causal graph used to model relationship of various factors and outcomes related to cancer with the following nodes: ['Pollution', 'Cancer', 'Smoker', 'Xray', 'Dyspnoea'], Which cause-and-effect relationship is more likely between nodes 'xray' and 'dyspnoea'?

A. changing the state of node 'xray' causally affects a change in another node 'dyspnoea'.

B. changing the state of node 'dyspnoea' causally affects a change in another node 'xray'.

C. There is no causal relation between the nodes 'xray' and 'dyspnoea'.

Make sure to first provide a grounded reasoning for your answer and then provide the answer in the following format: ⟨Answer⟩A/B/C⟨/Answer⟩. It is very important that you output the final Causal graph within the tags like ⟨Answer⟩A/B/C⟨/Answer⟩ otherwise your answer will not be processed.

Answer: Reasoning behind the lack of causal relation between X-ray and dyspnoea is that X-ray and dyspnoea are both effects of having cancer, but they do not directly cause or affect each other. X-ray is a diagnostic test that can help detect cancer in the lungs or other organs, while dyspnoea is a symptom of cancer that involves feeling short of breath. Therefore, X-ray and dyspnoea are not causally related, but they are both associated with cancer. Therefore answer is ⟨Answer⟩C⟨/Answer⟩

Question: For a causal graph used to model relationship of various factors and outcomes related to cancer with the following nodes: ['Pollution', 'Cancer', 'Smoker', 'Xray', 'Dyspnoea'], Which cause-and-effect relationship is more likely between nodes 'xray' and 'cancer'?

A. changing the state of node 'xray' causally affects a change in another node 'cancer'.

B. changing the state of node 'cancer' causally affects a change in another node 'xray'.

C. There is no causal relation between the nodes 'xray' and 'cancer'.

Make sure to first provide a grounded reasoning for your answer and then provide the answer in the following format: ⟨Answer⟩A/B/C⟨/Answer⟩. It is very important that you output the final Causal graph within the tags like ⟨Answer⟩A/B/C⟨/Answer⟩ otherwise your answer will not be processed.

Table A20: Chain of Thought Prompt

Answer: The causal effect of cancer on X-ray is that X-rays are often used to diagnose or detect cancer in different parts of the body, such as the bones, lungs, breasts, or kidneys123. X-rays use low doses of radiation to create pictures of the inside of the body and show the presence, size, and location of tumors. X-rays can also help monitor the response to treatment or check for signs of recurrence. Therefore, having cancer may increase the likelihood of getting an X-ray as part of the diagnostic process or follow-up care. Therefore answer is ⟨Answer⟩B⟨/Answer⟩

Question: For a causal graph used to model relationship of various factors and outcomes related to cancer with the following nodes: ['Pollution', 'Cancer', 'Smoker', 'Xray', 'Dyspnoea'],         Which cause-and-effect relationship is more likely between nodes 'pollution' and 'cancer'?

A. changing the state of node 'pollution' causally affects a change in another node 'cancer'.

B. changing the state of node 'cancer' causally affects a change in another node 'pollution'.

C. There is no causal relation between the nodes 'pollution' and 'cancer'.

Make sure to first provide a grounded reasoning for your answer and then provide the answer in the following format: ⟨Answer⟩A/B/C⟨/Answer⟩.         It is very important that you output the final Causal graph within the tags like ⟨Answer⟩A/B/C⟨/Answer⟩ otherwise your answer will not be processed.

Answer: The causal effect of pollution on cancer is that air pollution contains carcinogens (cancer-causing substances) that may be absorbed into the body when inhaled and damage the DNA of cells. Another possible reasoning is that particulate matter (tiny dust-like particles) in air pollution may cause physical damage to the cells in the lungs, leading to inflammation and oxidative stress and eventually cell mutations. A third possible reasoning is that air pollution may create an inflamed environment in the lungs that encourages the proliferation of cells with existing cancer-driving mutations. These are some of the hypotheses that researchers have proposed to explain how air pollution may cause cancer, but more studies are needed to confirm them. Therefore answer is ⟨Answer⟩A⟨/Answer⟩

Question: For a causal graph used to model relationship of various factors and outcomes related to cancer with the following nodes: ['Pollution', 'Cancer', 'Smoker', 'Xray', 'Dyspnoea'],         Which cause-and-effect relationship is more likely between nodes 'pollution' and 'smoker'?

A. changing the state of node 'pollution' causally affects a change in another node 'smoker'.

B. changing the state of node 'smoker' causally affects a change in another node 'pollution'.

C. There is no causal relation between the nodes 'pollution' and 'smoker'.

Make sure to first provide a grounded reasoning for your answer and then provide the answer in the following format: ⟨Answer⟩A/B/C⟨/Answer⟩.         It is very important that you output the final Causal graph within the tags like ⟨Answer⟩A/B/C⟨/Answer⟩ otherwise your answer will not be processed.

Answer: Reason behind the lack of causal relation between pollution and smoker is that pollution and smoking are both independent risk factors for respiratory problems, but they do not directly cause or affect each other. Pollution and smoking both contribute to air pollution, which can harm the health of people and the environment. However, pollution is mainly caused by human activities such as burning fossil fuels, deforestation, or industrial processes, while smoking is a personal choice that involves inhaling tobacco smoke. Therefore, pollution and smoker are not causally related, but they are both associated with respiratory problems. Therefore answer is ⟨Answer⟩C⟨/Answer⟩.

Table A21: Chain of Thought Prompt (continued..)

Question: For a causal graph used for modeling factors causing Coronary Heart Diseases with the following nodes: ['Family Disease', 'Gene', 'Smoking', 'Blood Pressure', 'Coronary Heart Disease', 'Headache'], Which cause-and-effect relationship is more likely between nodes 'Family Disease' and 'Gene'?

A. changing the state of node 'Family Disease' causally affects a change in another node 'Gene'.

B. changing the state of node 'Gene' causally affects a change in another node 'Family Disease'.

C. There is no causal relation between the nodes 'Family Disease' and 'Gene'.

Make sure to first provide a grounded reasoning for your answer and then provide the answer in the following format: ⟨Answer⟩A/B/C⟨/Answer⟩. It is very important that you output the final Causal graph within the tags like ⟨Answer⟩A/B/C⟨/Answer⟩ otherwise your answer will not be processed.

Answer: Reason behind the causal effect of family disease on gene is that family disease is a term that refers to diseases or health conditions that run in the family, meaning that they are influenced by genetic factors. Gene is a term that refers to the basic unit of heredity that carries information for a specific trait or function. Family disease can affect gene by altering the type or frequency of genes that are inherited by the offspring from their parents. For example, some family diseases are caused by deterministic genes, which are genes that guarantee the development of a disease if they are present in a person's genome. Other family diseases are influenced by risk genes, which are genes that increase the likelihood of developing a disease but do not guarantee it. Therefore, family disease can causally direct to gene by changing the genetic makeup of a person or a population. Therefore answer is ⟨Answer⟩A⟨/Answer⟩.

Question: For a causal graph used for modeling factors causing Coronary Heart Diseases with the following nodes: ['Family Disease', 'Gene', 'Smoking', 'Blood Pressure', 'Coronary Heart Disease', 'Headache'], Which cause-and-effect relationship is more likely between nodes 'Coronary Heart Disease' and 'Gene'?

 A. changing the state of node 'Coronary Heart Disease' causally affects a change in another node 'Gene'.

B. changing the state of node 'Gene' causally affects a change in another node 'Coronary Heart Disease'.

C. There is no causal relation between the nodes 'Coronary Heart Disease' and 'Gene'.

Make sure to first provide a grounded reasoning for your answer and then provide the answer in the following format: ⟨Answer⟩A/B/C⟨/Answer⟩. It is very important that you output the final Causal graph within the tags like ⟨Answer⟩A/B/C⟨/Answer⟩ otherwise your answer will not be processed.

Answer: Possible reasoning behind the causal effect of gene on coronary heart disease is that gene is a term that refers to the basic unit of heredity that carries information for a specific trait or function. Gene can affect coronary heart disease by influencing the structure and function of the blood vessels, the metabolism and transport of lipids (fats) in the blood, the inflammation and clotting processes, or the response to environmental factors such as smoking or diet. For example, some genes code for proteins that regulate the cell cycle and growth of the cells that line the arteries, which can affect their susceptibility to damage or plaque formation. Other genes code for proteins that control the synthesis and clearance of cholesterol or other lipids, which can affect their levels and deposition in the arteries. Therefore, gene can causally direct to coronary heart disease by modifying the biological pathways that contribute to the development or progression of the disease. Therefore answer is ⟨Answer⟩B⟨/Answer⟩

Table A22: Chain of Thought Prompt (continued..)

---

Question: For a causal graph used for modeling factors causing Coronary Heart Diseases with the following nodes: ['Family Disease', 'Gene', 'Smoking', 'Blood Pressure', 'Coronary Heart Disease', 'Headache'], Which cause-and-effect relationship is more likely between nodes 'Blood Pressure' and 'Smoking'?

A. changing the state of node 'Blood Pressure' causally affects a change in another node 'Smoking'.

B. changing the state of node 'Smoking' causally affects a change in another node 'Blood Pressure'.

C. There is no causal relation between the nodes 'Blood Pressure' and 'Smoking'.

Make sure to first provide a grounded reasoning for your answer and then provide the answer in the following format: ⟨Answer⟩A/B/C⟨/Answer⟩. It is very important that you output the final Causal graph within the tags like ⟨Answer⟩A/B/C⟨/Answer⟩ otherwise your answer will not be processed.

Answer: Possible reasoning behind the causal effect of smoking on blood pressure is that smoking is a habit that involves inhaling tobacco smoke, which contains nicotine and other harmful chemicals. Smoking can affect blood pressure by activating the sympathetic nervous system (SNS), which is the part of the nervous system that controls the body's response to stress or danger. When the SNS is activated, it releases hormones such as adrenaline and noradrenaline, which cause the heart to beat faster and harder, and the blood vessels to constrict. This results in a temporary increase in blood pressure, which can last for 15 to 20 minutes after each cigarette. Therefore, smoking can causally direct to blood pressure by stimulating the SNS and increasing the cardiac output and vascular resistance. Therefore answer is ⟨Answer⟩B⟨/Answer⟩.

Question: For a causal graph used for modeling factors causing Coronary Heart Diseases with the following nodes: ['Family Disease', 'Gene', 'Smoking', 'Blood Pressure', 'Coronary Heart Disease', 'Headache'], Which cause-and-effect relationship is more likely between nodes 'Headache' and 'Smoking'?

A. changing the state of node 'Headache' causally affects a change in another node 'Smoking'.

B. changing the state of node 'Smoking' causally affects a change in another node 'Headache'.

C. There is no causal relation between the nodes 'Headache' and 'Smoking'.

Make sure to first provide a grounded reasoning for your answer and then provide the answer in the following format: ⟨Answer⟩A/B/C⟨/Answer⟩. It is very important that you output the final Causal graph within the tags like ⟨Answer⟩A/B/C⟨/Answer⟩ otherwise your answer will not be processed.

Answer: One possible reasoning behind the lack of causal relation between headache and smoking is that headache and smoking are both associated with various health conditions, but they do not directly cause or affect each other12. Headache is a term that refers to pain or discomfort in the head, scalp, or neck, which can have many possible causes, such as stress, dehydration, infection, injury, or medication. Smoking is a habit that involves inhaling tobacco smoke, which contains nicotine and other harmful chemicals, which can increase the risk of diseases such as cancer, heart disease, stroke, and lung disease. Therefore, headache and smoking are not causally related, but they are both linked to different health problems. Therefore the answer is ⟨Answer⟩C⟨/Answer⟩

Table A23: Chain of Thought Prompt (continued..)

Question: For a causal graph used for modeling factors causing Coronary Heart Diseases with the following nodes: ['Family Disease', 'Gene', 'Smoking', 'Blood Pressure', 'Coronary Heart Disease', 'Headache'], Which cause-and-effect relationship is more likely between nodes 'Headache' and 'Smoking'?

A. changing the state of node 'Headache' causally affects a change in another node 'Smoking'.

B. changing the state of node 'Smoking' causally affects a change in another node 'Headache'.

C. There is no causal relation between the nodes 'Headache' and 'Smoking'.

Make sure to first provide a grounded reasoning for your answer and then provide the answer in the following format: ⟨Answer⟩A/B/C⟨/Answer⟩. It is very important that you output the final Causal graph within the tags like ⟨Answer⟩A/B/C⟨/Answer⟩ otherwise your answer will not be processed.

Answer: One possible reasoning behind the lack of causal relation between headache and smoking is that headache and smoking are both associated with various health conditions, but they do not directly cause or affect each other. Headache is a term that refers to pain or discomfort in the head, scalp, or neck, which can have many possible causes, such as stress, dehydration, infection, injury, or medication. Smoking is a habit that involves inhaling tobacco smoke, which contains nicotine and other harmful chemicals, which can increase the risk of diseases such as cancer, heart disease, stroke, and lung disease. Therefore, headache and smoking are not causally related, but they are both linked to different health problems. Therefore the answer is ⟨Answer⟩C⟨/Answer⟩

Question: For a causal graph used for modeling factors causing Coronary Heart Diseases with the following nodes: ['Family Disease', 'Gene', 'Smoking', 'Blood Pressure', 'Coronary Heart Disease', 'Headache'], Which cause-and-effect relationship is more likely between nodes 'Coronary Heart Disease' and 'Smoking'?

A. changing the state of node 'Smoking' causally affects a change in another node 'Coronary Heart Disease'.

B. changing the state of node 'Coronary Heart Disease' causally affects a change in another node 'Smoking'.

C. There is no causal relation between the nodes 'Coronary Heart Disease' and 'Smoking'.

Make sure to first provide a grounded reasoning for your answer and then provide the answer in the following format: ⟨Answer⟩A/B/C⟨/Answer⟩. It is very important that you output the final Causal graph within the tags like ⟨Answer⟩A/B/C⟨/Answer⟩ otherwise your answer will not be processed.

Answer: Possible reasoning behind the causal effect of smoking on coronary heart disease is smoking damages the heart and blood vessels by raising triglycerides, lowering HDL, increasing blood clotting, and impairing blood flow to the heart. This can lead to plaque buildup, heart attacks, and death. Therefore answer is ⟨Answer⟩A⟨/Answer⟩.

Question: For a causal graph used for context with the following nodes: nodes, Which cause-and-effect relationship is more likely between nodes X and Y?

A. changing the state of node X causally affects a change in another node Y.

B. changing the state of node Y causally affects a change in another node X.

C. There is no causal relation between the nodes X and Y.

Make sure to first provide a grounded reasoning for your answer and then provide the answer in the following format: ⟨Answer⟩A/B/C⟨/Answer⟩. It is very important that you output the final Causal graph within the tags like ⟨Answer⟩A/B/C⟨/Answer⟩ otherwise your answer will not be processed.

Table A24: Chain of Thought Queries (continued..)

Which cause-and-effect relationship is more likely?

A. changing the state of node which says X causally affects a change in another node which says Y.

B. changing the state of node which says Y causally affects a change in another node which says X.

C. There is no causal relationship between node X and Y.

Make sure to first output a factually grounded reasoning for your answer. X and Y are nodes of a Causal Graph. The causal graph is sparse and acyclic in nature. So option C could be chosen if there is some uncertainity about causal relationship between X and Y.

First give your reasoning and after that please make sure to provide your final answer within the tags ⟨Answer⟩A/B/C⟨/Answer⟩.
It is very important that you output your final answer between the tags like ⟨Answer⟩A/B/C⟨/Answer⟩ otherwise your response will not be processed.

Table A25: Base Queries

For the nodes X and Y which form an edge in a Causal Graph, you have to identify which cause-and-effect relationship is more likely between the nodes of the edge. This will be used to rearrange the nodes in the edge to create a directed edge which accounts for causal relation from one node to another in the edge.

A. changing the state of node X causally affects a change in another node Y.

B. changing the state of node Y causally affects a change in another node X.

C. There is no causal relation between the nodes X and Y.

You can also take the edges from the skeleton which have been rearranged to create a directed edge to account for causal relationship between the nodes: directed_edges.
Make sure to first output a factually grounded reasoning for your answer. First give your reasoning and after that please make sure to provide your final answer within the tags ⟨Answer⟩A/B/C⟨/Answer⟩.
It is very important that you output your final answer between the tags like ⟨Answer⟩A/B/C⟨/Answer⟩ otherwise your response will not be processed.

Table A26: Iterative orientation Queries

For the following undirected edge in a Causal Graph made of nodes X and Y, you have to identify which cause-and-effect relationship is more likely between the nodes of the edge. This will be used to rearrange the nodes in the edge to create a directed edge which accounts for causal relation from one node to another in the edge.

A. changing the state of node X causally affects a change in another node Y.

B. changing the state of node Y causally affects a change in another node X.

C. There is no causal relation between the nodes X and Y.

You can also take the other directed edges of nodes X: X_edges and Y: Y_edges of the Causal graph as context to redirect the edge to account for causal effect.
Make sure to first output a factually grounded reasoning for your answer. First give your reasoning and after that please make sure to provide your final answer within the tags ⟨Answer⟩A/B/C⟨/Answer⟩.
It is very important that you output your final answer between the tags like ⟨Answer⟩A/B/C⟨/Answer⟩ otherwise your response will not be processed.

Table A27: Iterative One Hop Queries

*Identify the causal relationships between the given variables and create a directed acyclic graph to {context}. Make sure to give a reasoning for your answer and then output the directed graph in the form of a list of tuples, where each tuple is a directed edge. The desired output should be in the following form: [('A','B'), ('B','C')] where first tuple represents a directed edge from Node 'A' to Node 'B', second tuple represents a directed edge from Node 'B' to Node 'C'and so on.*

*If a node should not form any causal relationship with other nodes, then you can add it as an isolated node of the graph by adding it seperately. For example, if 'C' should be an isolated node in a graph with nodes 'A', 'B', 'C', then the final DAG representation should be like [('A','B'), ('C')].*
*Use the description about the node provided with the nodes in brackets to form a better decision about the causal direction orientation between the nodes.*

*It is very important that you output the final Causal graph within the tags <Answer></Answer>otherwise your answer will not be processed.*

*Example:*
*Input: Nodes: ['A', 'B', 'C', 'D'];*
*Description of Nodes: [(description of Node A), (description of Node B), (description of Node C), (description of Node D)]*
*Output: <Answer>[('A','B'),('C','D')]</Answer>*
*Question:*
*Input: Nodes: {Triplet Nodes Input}*
*Description of Nodes: {Description of Each Node from the Triplet}*
*Output:*

Table A28: The *triplet* query template, which includes a concise context of the graph, the input triplet nodes and their respective descriptions. As an example, for the Child graph, the context is *"to model congenital heart disease in babies"*, the three nodes may be *('HypoxiaInO2', 'Grunting', 'GruntingReport')*; and their node descriptions are *["hypoxia when breathing oxygen", "grunting in infants", "report of infant grunting"]* respectively.

---

Input: **('HypDistrib', 'LowerBodyO2')**

Answer: Low oxygen areas equally distributed around the body can affect the level of oxygen in the lower body by reducing the amount of oxygen available for circulation. Therefore, the answer is ⟨Answer⟩A⟨/Answer⟩.

Input: **('Disease', 'LungFlow')**

Answer: Infant methemoglobinemia is a condition where the blood is unable to carry enough oxygen, which can lead to low blood flow in the lungs. Therefore, the answer is ⟨Answer⟩A⟨/Answer⟩.

Input: **('CardiacMixing', 'HypDistrib')**

Answer: Mixing of oxygenated and deoxygenated blood is a common cause of congenital heart disease in babies. This mixing can occur due to structural defects in the heart, such as a hole between the heart chambers or abnormal connections between the blood vessels. This mixing can lead to low oxygen areas equally distributed around the body. Therefore, the answer is ⟨Answer⟩B⟨/Answer⟩.

---

Table A29: Example LLM (GPT-3.5-turbo) outputs for estimating the causal direction between a given pair of nodes (with their descriptions) using the CoT Prompt (refer Table A20 in Appendix for the prompt). Given a pair (X,Y), Option $A$ refers to X causing Y, option $B$ refers to Y causing X, and option $C$ refers to no causal relationship between the variables. The pairs shown are from the *Child* graph from BNLearn repository (Scutari & Denis, 2014), that is used for modeling Congenital Heart Disease in infants.

---

Input: **('Right C7','Discoligment injury C6-C7')**

Answer: Discoligment injury C6-C7 can cause compression of the nerve roots that exit the spinal cord at the C7 level, which can lead to symptoms such as pain, numbness, and weakness in the right C7 dermatome. Therefore, the answer is ⟨Answer⟩B⟨/Answer⟩.

Input: **('Right C7', 'Left C7')**

Answer: Right C7 and left C7 are both parts of the cervical spine and are not known to directly influence each other. Therefore, the answer is ⟨Answer⟩C⟨/Answer⟩.

Input: **('Right elbow trouble', 'Left shoulder trouble')**

Answer: There is no direct causal relationship between right elbow trouble and left shoulder trouble. They may both be symptoms of a larger underlying condition, but they do not directly cause or affect each other. Therefore the answer is ⟨Answer⟩C⟨/Answer⟩.

---

Table A30: Example LLM (GPT-3.5-turbo) reasoning outputs for estimating causal directionality between different pairs of nodes using CoT queries (refer Table A20 for the query) for Neuropathic subgraph (used for pain diagnosis).

