# OpenReview forum: "Causal Order: The Key to Leveraging Imperfect Experts in Causal Inference"
_ICLR.cc/2025/Conference — ICLR 2025 Poster_

### Official Review · Reviewer_JgEV · 2024-10-19

**Soundness:** 2
**Presentation:** 3
**Contribution:** 2
**Rating:** 6
**Confidence:** 3

**Summary:**

This paper explores the use of large language models (LLMs) for causal inference tasks, particularly causal discovery. The authors present two key arguments: (1) For both LLMs (termed as imperfect experts) or perfect experts, it is more effective to identify the topological order of causal variables rather than directly attempting to discover the full causal graph. They demonstrate that this approach yields more robust results and remains valuable for downstream tasks such as causal discovery or treatment effect estimation. (2) Since LLMs are imperfect experts, relying on existing pairwise prompts to determine causal order may still result in cycles. To address this, the authors propose a triplet-based prompt design. They empirically validate their claims on several datasets.

**Claim**: I have research experience in causality. However, I have limited experience in LLMs or LLMs for causality. Hence, I might miss something especially when it comes to evaluation of novelty or comparison with existing works.

**Strengths:**

1. Regarding the first key contribution, I find the proposal of identifying causal orders rather than full causal graphs to be both useful and insightful.
2. Overall, the paper is easy to follow. While there are some typos, the writing remains generally clear.

**Weaknesses:**

I will present my main concerns here. Minor concerns or questions can be found in the Question section.

**Significance of contribution**

I have several concerns regarding the methodology section:
1. 3.1 - While I agree with the intuition that identifying causal order is easier and leads to fewer errors, I’m not sure the result in this subsection is particularly interesting by itself. It seems almost obvious, as knowing the correct causal order is a prerequisite for identifying the causal graph. What matters more is whether the reduced chance of errors still preserves the usefulness of the method.
2. 3.2 - This section should have provided evidence addressing the question above. However, I find it difficult to assess the significance of the method presented. Proposition 3.2 offers a sufficient condition for backdoor adjustment, but applying this condition broadly to include everything that satisfies it seems impractical.
3. 4.1 - Should these be considered as contributions of the paper?
4. 4.2 - The authors try to justify why triplet prompt is better than pairwise prompt here. One thing unclear to me is that, as more variables are provided in the context (as the authors also mentioned), there is a higher chance of LLMs making mistakes. In practice, does that mean an $\epsilon$-expert LLM would become an $\epsilon’$-expert where $\epsilon’>\epsilon$? If so, would this become a trade-off and how should we interpret this tradeoff?
5. 4.2 - Another question regarding this theorem: It seems that the major problem the authors try to resolve is to avoid cycles. However, Proposition 4.1 focuses on the error of edge prediction given the assumption of acyclicity. Hence, I am not sure if this theorem provides insight on how to resolve the key problem of this paper.

**Experiment**

My main concern is the lack of comparison with existing LLM-based methods. For example, while it’s valuable to verify that providing causal order helps with downstream causal tasks, do we know if integrating causal order into causal discovery outperforms directly using LLMs to identify the causal graph or providing other types of inputs to a causal discovery algorithm? Could the authors justify their choice not to compare against existing baselines, such as those mentioned in Section 2?

**Questions:**

1. Experiment - Missing reference to Table 2 in the main paper.
2. Line 114 - Could the authors provide some justification on why the redundancy leads to a more reliable order?
3. Line 52 - Would using triplet prompt lead to a more efficient method in this sense?

---

> ### Author Response · Authors · 2024-11-21
> **Response to reviews by reviewer JgEV**
>
> We thank the reviewer for their insightful comments, we have tried our best to incorporate their suggestions and answer their queries.
>
> **Response to weaknesses:**
>
> **>3.1 - While I agree with the intuition that identifying causal .... chance of errors still preserves the usefulness of the method.**
>
> **Response:** The key result of this section is that even with a Perfect Expert (e.g., human domain expert), the inferred graph using pairwise queries can be incorrect (Prop 3.2). Given that inferring edges using pairwise queries is the dominant method in the LLM-based discovery literature, we believe that this is result of significance.
>
> **>3.2 - This section should have provided evidence addressing the ..... this condition broadly to include everything that satisfies it seems impractical.**
>
> **Response:**
> Section 3.2 provides 3 key results showing the usefulness of causal order.
>
> 1. Prop 3.1 shows that causal order can be used to find a valid, unbiased backdoor set. For example, in Table A14, we compare the estimation error when using the minimal backdoor set obtained using the full graph, to the "maximal" backdoor sets obtained using the causal order. Across sample sizes from 250-10000 for the Asia dataset, we find the difference in estimation error between the two kinds of adjustment sets is minimal (see also Table 5). However, as Cinelli et al. (2022) argue, such a maximal set may have high variance. Note that we consider the causal order as output only because it is a structure that we can obtain accurately from an expert. To obtain both unbiased estimation and low variance, we can use the causal order as input to a data-based graph discovery algorithm (as described in pt 3 below) and then derive the optimal backdoor set using the obtained graph.
>
> 2. In  addition to Prop 3.2, the key result of this section is that the causal order metric, $D_{top}$ is a more accurate measure of effect estimation error than SHD. This result assumes significance since SHD is a popular method for evaluating graph quality especially for LLM-based methods; we show that if the downstream task is effect inference, measuring $D_{top}$ is more suitable.
>
> 3. We also provide algorithms on how causal order can be used to improve accuracy of data-based graph discovery algorithms. Empirically, we show that the accuracy of incorporating (expert-predicted) causal order in discovery algorithms is higher than that of the discovery algorithms alone.
>
> **>4.1 - Should these be considered as contributions of the paper?**
>
> **Response:** Yes, our extensions of pairwise strategies can be considered as contributions of the paper. To evaluate the benefit of the Triplet method, we create stronger versions of the standard pairwise method that utilize additional graph context and the latest advances in LLM prompting. The objective is to make sure that we have explored possible extensions of the pairwise method as much as possible, before proposing the Triplet method. One of these methods, Pairwise (CoT), obtains significantly better results than the standard pairwise method used in the literature (see Table 3), but is still less accurate than the Triplet method.
>
> **>4.2 - The authors try to justify why triplet prompt is ... would become an ϵ′-expert where ϵ′>ϵ? If so, would this become a trade-off and how should we interpret this tradeoff?**
>
> **Response:** Yes, as we increase the number of nodes in the prompt, there is a tradeoff: Adding more nodes provides more context and thus is beneficial, but more nodes in the LLM's prompt can also lead to higher error (Levy et al., 2024) and higher computational cost. Therefore, we tackled this question empirically by comparing pairwise, triplet, and quadruplet-based prompts. As Table A9 shows, using a quadruplet prompt slightly increases accuracy but leads to a significant increase in the number of LLM calls. In contrast, the increase in accuracy (especially cycle avoidance) is substantial when moving from pairwise to the triplet method.
>
> Given these considerations, we decided to go with the Triplet prompt, as it allows for adding more context with minimal increase in prompt complexity and total number of LLM calls. Note that future iterations of language models might be able to handle longer context better with more improvements, therefore the ϵ′ will vary with model size, architecture and data the model is trained on. Since we don't have the information about this, it will be difficult to model ϵ′ accurately. However, with the LLMs that we have tried (GPT-4, GPT-3.5, Phi-3 and LLama3), we do not see an increased error when using the triplet prompt compared to the pairwise prompt.

---

> ### Author Response · Authors · 2024-11-21
> **Continuation of response to reviewer JgEV**
>
> **>4.2 - Another question regarding this theorem: It seems that ... to resolve the key problem of this paper.**
>
> **Response:**
> The key difference is between local acyclicity for a triplet (assumption) and global acyclicity for the graph (empirical result). Based on our experiments with LLMs such as GPT-3.5 and 4, we find that they can obey the acyclicity constraint over 3 variables when specifically instructed to do so. Therefore, we assume that imperfect expert's output over 3 variables will not contain a cycle. Based on this, Prop 4.1 shows that error in predicting causal relationship between pairs of nodes is lower for the triplet prompt compared to the pairwise prompt. (For context, in the triplet method, all nodes are divided into all possible groups of three, and the expert (like LLMs) is specifically prompted to create a Directed Acyclic Graph which models the causal relationship between the nodes for each subgroup (refer Table A24 for the prompt template used for triplet subgraph)).
>
> This result, however, does not ensure that the global graph has no cycles. This is what the triplet method achieves, through aggregation over triplets and further steps. Overall, since the triplet method ensures a higher quality prediction of the edge direction, we believe this leads to lesser cycle formations empirically as compared to pairwise.
>
> >My main concern is the lack of comparison with existing LLM-based methods. ... against existing baselines, such as those mentioned in Section 2?
>
> **Response**:
>
> We thank the reviewer for this point. Our work focused on tackling the shortcomings of the standard pairwise LLM-based approaches for causal discovery. We compare our method against the pairwise prompting strategy proposed by (Kiciman et al., 2022). For a fair comparison, we further propose stronger variants of the pairwise method based on additional context and advances in LLM prompting (All Directed Edges, One Hop Iteration and Chain of Thought, refer Table A5).
>
>
> As an additional baseline reflecting the state-of-the-art in LLM-based methods, we have implemented two more methods from Jiralerspong et al., 2024 ("Efficient Causal Graph Discovery using Large Language Models") on 4 complex graphs used in our analysis: Covid, Alzheimers, Child and Asia. We evaluate the methods over two LLMs: GPT-3.5-Turbo and GPT-4.
> This paper presents an efficient breadth-first approach (BFS) to causal discovery using LLMs. The pipeline operates level-wise, starting by querying the LLM to identify all nodes independent of others in the graph. From these independent nodes, the LLM is queried to determine dependent nodes, adding edges accordingly. The dependent nodes are then added to a queue, and the process is repeated iteratively for each node in the queue until it is empty. For BFS+Stats method, the pearson corelation coefficient for the independent nodes to all the other nodes is added in the prompt as additional context for graph discovery.
>
> As the results below show, BFS and BFS-Stats methods obtain lower accuracy than the Triplet method. In particular, except BFS with GPT-4, all configurations lead to cycles in Child and/or Covid datasets. SHD and Dtop for BFS and BFS-Stats methods (especially with GPT-3.5) are also higher than the Triplet method.
>
>
> BFS (GPT-3.5-turbo)
>
> | Dataset | Dtop | SHD | IN | Cycles |
> | -------- | -------- | -------- | -------- | -------- |
> | Asia     | 2     | 7     | 0     | 0     |
> | Alzheimers     | 5     | 17     | 2     | 0     |
> | Child     | -     | 40     | 0     | 6     |
> | Covid     | -     | 28     | 0     | 4     |
>
> BFS (GPT-4)
>
> | Dataset | Dtop | SHD | IN | Cycles |
> | -------- | -------- | -------- | -------- | -------- |
> | Asia     | 0     | 1     | 0     | 0     |
> | Alzheimers     | 0     | 34     | 0     | 0     |
> | Child     | 11     | 30     | 0     | 0     |
> | Covid     | 5     | 20     | 0     | 0     |
>
> BFS + Statistics (GPT-3.5-Turbo)
>
> | Dataset | Dtop | SHD | IN | Cycles |
> | -------- | -------- | -------- | -------- | -------- |
> | Asia     | -     | 23     | 0     | 33     |
> | Alzheimers     | -     | 27     | 1     | 17     |
> | Child     | -     | 52     | 2     | 21     |
> | Covid     | -     | 30     | 0     | 15     |
>
> BFS + Statistics (GPT-4)
>
> | Dataset | Dtop | SHD | IN | Cycles |
> | -------- | -------- | -------- | -------- | -------- |
> | Asia     | 0     | 3     | 0     | 0     |
> | Alzheimers     | -     | 14     | 0     | 1     |
> | Child     | 2     | 27     | 4     | 0     |
> | Covid     | -     | 32     | 1     | 10     |
>
> For reference these are the results of triplet with GPT-3.5-turbo
>
> | Dataset | Dtop | SHD | IN | Cycles |
> | -------- | -------- | -------- | -------- | -------- |
> | Asia     | 1     | 14     | 0     | 0     |
> | Alzheimers     | 4     | 28     | 0     | 0     |
> | Child     | 1     | 28     | 10     | 0     |
> | Covid     | 0     | 30     | 0     | 0     |

---

> > ### Author Response · Authors · 2024-11-21
> > **Continuation of response to reviewer JgEV**
> >
> > While BFS and BFS+stats fail to give 0 cycles across all graphs even with GPT-4, triplet consistently gives acyclic causal graphs, with lower Dtop.
> >
> > **PC + BFS**. We present another ablation which explores a hybrid approach where the PC algorithm integrates an LLM-derived prior (GPT-4) obtained via BFS for Alzheimer's and COVID graphs. The prior directly provides edge orientations, which guide the initial graph structure, while PC subsequently orients remaining edges. Unlike triplet that used only causal order, this approach incorporates the full graph as a prior. The PC algorithm is further supported by a large observational dataset of 10,000 samples.
> >
> > The results show that PC + BFS (GPT-4) is also outperformed by Triplet method (GPT-3.5). Specifically, PC+BFS yields  1 cycle and higher SHD on Covid dataset. On the Alzheimers dataset, PC+BFS is comparable: it yields higher Dtop but a lower SHD.
> >
> > | Dataset | Dtop | SHD | IN | Cycles |
> > | -------- | -------- | -------- | -------- | -------- |
> > | Alzheimers     | 5     | 14     | 0     | 0     |
> > | Covid     | -     | 36     | 0     | 1     |
> >
> >
> > **Response to Questions:**
> >
> > >1. Experiment - Missing reference to Table 2 in the main paper.
> >
> > Thanks for pointing it out. We will fix this.
> >
> > >2. Line 114 - Could the authors provide some justification on why the redundancy leads to a more reliable order?
> >
> > **Response**: In the triplet method, for each pair of nodes, we have multiple answers from the LLM, each considering a different auxiliary node as context.
> > To aggregate the final graph, we take majority vote on the answers from each edge, further leading to robustness (See our explanation in Section 4.2 (line 334)).
> > Redundancy thus makes the causal order more reliable,  in comparison to the pairwise method where each pair is queried only once and without any additional context about other nodes in the graph. See our explanation in
> > Sec 4.2 (line 334).
> > >3. Line 52 - Would using triplet prompt lead to a more efficient method in this sense?
> >
> > **Response**: Yes, rather than providing all the nodes as context, the triplet prompt only adds a single node as additional context. As a result, the LLM handles only 3 nodes at a time, even for large graphs.

---

> > > ### Comment · Reviewer_JgEV · 2024-11-25
> > > **Follow-up**
> > >
> > > Thank you for the detailed rebuttals. However, a few of my questions in the section **Significance of contribution** have not been fully addressed. Could you please provide further clarification?
> > > 1. W1 (regarding 3.1) - The authors comment that "The key result of this section is that even with a Perfect Expert (e.g., human domain expert), the inferred graph using pairwise queries can be incorrect (Prop 3.2)." I don't quite get this point for the following reasons: (1) The authors dedicated more than 2 pages (Section 3) with a great emphasis on the importance and use case of causal orders, whose significance is what I was questioning about (2) It is unclear to me how Prop 3.2 explains why pairwise query is not good enough (3) Prop 3.2 is not novel result
> > > 2. W2 (regarding 3.2) - Could the authors answer my original question that "Proposition 3.2 offers a sufficient condition for backdoor adjustment, but applying this condition broadly to include everything that satisfies it seems impractical."?
> > > 3. W4 (regarding 4.2) - I was mainly asking about after involving a third variable, if $\epsilon$ would change accordingly. If so, I wonder if that still leads to a fair comparison between pairwise and triplet prompt.
> > >
> > > Overall I like the idea of the paper but I am kind of doubtful about the significance and usefulness of the results in Section 3 and 4.

---

> ### Author Response · Authors · 2024-11-25
>
> Thanks for engaging on the rebuttal. We clarify on the importance and usecases of causal order below.
>
> > W2 (regarding 3.2) - Could the authors answer my original question that "Proposition 3.2 offers a sufficient condition for backdoor adjustment, but applying this condition broadly to include everything that satisfies it seems impractical."?
>
> Including variables appearing before treatment (in the causal order) is actually **a widespread practice in biomedical and social science empirical studies**. In these studies, such variables are called "pre-treatment variables" and a common practice is to condition on all of them. For this reason, we do not think that our proposal is impractical. The importance of Prop 3.2 is to show the utility of the causal order to identify such a commonly used adjustment set.
>
> For example, refer to the Covariate selection chapter [1] by Sauer, Brookhart, Roy and Vanderwheele in a User Guide ("Developing a Protocol for Observational Comparative Effectiveness Research"). In the section on "Adjustment for all observed pre-treatment covariates", **they mention the widely used propensity score adjustment and write, _"The greatest importance is often placed on balancing all pretreatment covariates."_** They also add that while theoretically colliders can bias the result, _"in practice, pretreatment colliders are likely rarer than ordinary confounding variables."._
>
> Further, when unobserved confounding cannot be ruled out (as is the case with most observational studies), evidence is not clear on whether we should include all pre-treatment covariates or select a few, especially because the true graph may be unknown. _"Strong arguments exist for error on the side of overadjustment (adjusting for instruments and colliders) rather than failing to adjust for measured confounders (underadjustment). Nevertheless, adjustments for instrumental variables have been found to amplify bias in practice"._ As the last sentence suggests, note that **we are not claiming that adjusting for all pre-treatment variables (variables before treatment in causal order) is always the correct approach; but rather showing that it can be practical in many situations.**
>
> Theoretically, of course, improvements to this causal order criterion are possible. Vanderweele and Shpitser (2011) [2] cite the popular practice of using "all pre-treatment variables" and propose the Disjunctive Cause criterion as an improvement. This criterion states that if a pre-treatment variable causes the treatment, outcome, or both; then it should be included in the adjustment set. Note that this criterion---effectively including all pre-treatment ancestors of treatment and/or outcome---is quite close to the causal order-based criterion in our paper. Except for possibly conditioning on a collider in cases where there are unobserved variables in the graph (see Fig. 1 from [2]), additional variables in the causal order adjustment superset will not have a significant effect on the estimate.
>
>
> [1] Sauer, Brockhart, Roy, Vanderweele. Chapter 7: Covariate Selection https://www.ncbi.nlm.nih.gov/books/NBK126194/ in Developing a Protocol for Observational Comparative Effectiveness Research: A User's Guide (2013).
>
> [2] Vanderweele, Shpitser (2011).  A new criterion for confounder selection. *Biometrics.* https://pmc.ncbi.nlm.nih.gov/articles/PMC3166439/

---

> > ### Author Response · Authors · 2024-11-25
> >
> > > 1) The authors comment that "The key result of this section is that even with a Perfect Expert (e.g., human domain expert), the inferred graph using pairwise queries can be incorrect (Prop 3.2)." I don't quite get this point for the following reasons: (1) The authors dedicated more than 2 pages (Section 3) with a great emphasis on the importance and use case of causal orders, whose significance is what I was questioning about (
> >
> > Apologies, we meant to say Prop 3.1 here. Let us reiterate the two important usecases of causal order.
> > 1) **Identifying a suitable adjustment set for effect inference**. We already discussed it above.
> > 2) **Providing a prior or constraint to causal discovery methods**. As discussed in Section 3.2, causal order provides an accurate interface between domain experts and discovery algorithms. (Obtaining a causal graph from the domain expert is not suitable, as Prop 3.1 shows that even a perfect expert can output the wrong graph.) Moreover, causal order is not just a simpler structure, it also helps improve the accuracy of discovery methods: we show how causal order provides non-trivial improvements to existing discovery algorithms, both algorithmically (Sec 3.2) and empirically (Section 5).
> >
> > > W4 (regarding 4.2) - I was mainly asking about after involving a third variable, if would change accordingly. If so, I wonder if that still leads to a fair comparison between pairwise and triplet prompt.
> >
> > Ah, thanks for clarifying. At least in our experiments, we did not see an increase in error rate ($\epsilon$) as we move from pairwise to triplet prompt. Perhaps the effects of higher error due to longer context lengths may kick in at larger context sizes than what triplet prompt provides. Also, note that Prop 4.1 shows theoretically that _effective_ pairwise error rate for the triplet prompt is lower than that of a pairwise prompt (assuming an Expert that given a triplet of nodes, predicts causal relationship between all pairs of nodes sequentially).

---

> > > ### Comment · Reviewer_JgEV · 2024-11-29
> > > **Response**
> > >
> > > Thank you for the explanation.
> > >
> > > Overall, most of my concerns in the review have been addressed in the rebuttal and following discussion. While I am still kind of doubtful about the significance of the theoretical contribution of this paper, I think it could provide some interesting perspectives to the community of LLMs and causality.
> > >
> > > I have adjusted my score accordingly.

---

### Official Review · Reviewer_W4GZ · 2024-10-28

**Soundness:** 2
**Presentation:** 2
**Contribution:** 2
**Rating:** 6
**Confidence:** 3

**Summary:**

This paper proposes a new technique for causal discovery algorithms by leveraging triplets. The authors argue that traditional causal discovery algorithms based on pairwise variable relationships cannot handle situations involving mediating variables and can lead to circular structures in causal graphs. Thus, they emphasize the importance of utilizing causal order, meaning modeling based on the topological order among variables. Causal order is effective for downstream tasks, and the triplet approach can efficiently identify causal order. The authors discuss the error of estimated causal order from both empirical and theoretical perspectives, especially in the presence of imperfect experts.

**Strengths:**

The approach introduces the use of causal order as the output for causal discovery algorithms employed by LLMs, with D_{top} topological divergence as a metric. This novel approach brings a fresh perspective and insight into the fields of LLMs and causal discovery.

**Weaknesses:**

1. Typo error: Lines 333-334 seem to have missed a closing parenthesis in O(V^3).
2. Complexity considerations: According to the paper, the triplet-based method has a complexity of O(V^3), whereas the traditional pairwise method has a complexity of O(V^2). For larger-scale graphs, the time complexity of these methods poses significant limitations for causal discovery.
3. The approach of using LLMs to simply determine causal relationships between variables is challenging to implement effectively in real-world applications. It requires the large model itself to possess a high level of expert knowledge, which is often difficult to achieve, as real causal inference typically demands deep domain-specific expertise and nuanced understanding beyond general-purpose language models.

**Questions:**

1. The current method produces a causal order rather than a full causal graph. While the authors suggest that, in the absence of confounders, this causal order can offer a superset of backdoor adjustment sets, the presence of “imperfect experts” in real-world applications may lead to additional error accumulation in the causal order. This raises uncertainty about whether the derived causal order is sufficiently reliable for downstream tasks, such as causal effect estimation, and how significantly these biases could impact the results. Experimental validation would be valuable to assess the practicality and risks of this trade-off.
2. In the triplet method, the authors employ a four-step process to determine causal order. Unlike the pairwise approach, each iteration uses a third auxiliary variable \(C\) to determine the causal edge direction between \(A\) and \(B\), followed by aggregation across all variables. My understanding is that the pairwise approach is simply a special case of this method where the number of auxiliary variables equals zero. The authors could further clarify the distinctions and differences between the pairwise approach under this four-step method and the current approach.

---

> ### Author Response · Authors · 2024-11-21
> **Response to reviewer W4GZ's comments**
>
> We thank the reviewer for their insightful comments, we have tried our best to incorporate their suggestions and answer their queries.
>
> **Response to weaknesses**
>
> **>1. Typo error: Lines 333-334 seem to have missed a closing parenthesis in O(V^3).**
>
> **Response**: Thanks for pointing it out. We will fix it.
>
> **>2. Complexity considerations: According to the paper, the triplet-based method ... methods poses significant limitations for causal discovery.**
>
> **Response**: For larger graphs, we propose a $O(kV^2)$ variant of the triplet method below. Before that, however, we want to highlight the significant **increase in both accuracy and efficiency** that the triplet method provides compared to the pairwise method.
>
> With the triplet method, our main aim was to develop a method that significantly improves the accuracy of LLM-based causal discovery. A second aim was to build a robust method such that even smaller LMs can be used and they provide better accuracy than the pairwise method. As a result, the triplet method leads to a higher accuracy and (cost and time) efficiency compared to the pairwise method, even though its theoretical complexity is $O(V^3)$. For a large graph with many nodes, using the triplet method with GPT-3.5 can obtain significantly higher accuracy than pairwise method using GPT-4, and costs significantly less: based on OpenAI's pricing if we want to query for a 100 node graph, pairwise (GPT-4) would cost `$574` while the triplet method costs `$55` (see Appendix E, line 1371). Further, for larger graphs, we can even run Triplet method with much smaller models such as Phi-3 (3.8B params) and Llama3 8B, leading to further efficiency gains in both wall clock time and costs (while obtaining better accuracy than pairwise with GPT-4, see Table 2).
>
> That said, we believe modified approaches of triplet could be adopted for improved scalability. Rather than incorporating votes from all triplet subgraphs while deciding on edge direction between a pair of nodes, we can sample a fixed number of triplets per variable pair.  So if number of triplets used per pair is *k*, then the overall time complexity becomes O(kV^2), where *k* can be constant. Identifying optimal ways of choosing the subset can be a future extension of the work.
>
> **> 3. The approach of using LLMs to simply determine ... understanding beyond general-purpose language models.**
>
> **Response:** We agree that using LLMs for causal discovery is challenging in real-world scenarios. However, many real-world graph discovery problems involve a combination of inferring novel and widely known relationships, and we believe that LLMs can save significant effort in extracting the known relationships.
>
> Therefore, the focus of this work is to develop a robust method to extract causal relationships from LLMs, at least the ones that are known to a general-purpose language model. As can be seen from our experiments in highly domain-specific datasets such as Neuropathic and Covid-19, the triplet method substantially improves the accuracy of obtaining such causal relationships.
>
> That said, for complex real-world scenarios, a combination of LLM and data-based algorithms is more suited. Therefore, we proposed two variants that combine LLMs with data-based algorithms such as PC and CaMML (for example, LLMs may be used for providing a prior on the known relationships, which may help algorithms to learn the remaining novel relationships). In a similar direction, a key benefit of the triplet prompt is that it can also provide a measure of LLM's uncertainty (for each variable pair $<A, B>$, fraction of triplets that predict A causes B, B causes A, or no relationship) that can be useful for weighting the LLM-based prior for data-based algorithms.

---

> ### Author Response · Authors · 2024-11-21
> **Response to Questions by reviewer W4GZ**
>
> **>1. The current method produces a causal order rather than a full causal graph .... would be valuable to assess the practicality and risks of this trade-off.**
> **Response:**
>
> This is a great question. Contrary to the intuition above, we find that the triplet order method obtains more accurate effect estimates than the pairwise method that outputs a graph. Below we show an analysis using the Survey dataset and five combinations of treatment and outcome. The backdoor set computed from the pairwise method's graph results in a higher error than the "maximal" backdoor set computed from the triplet method's order.
>
>
> | Treatment, Target | $\epsilon_{ATE}$ for Pairwise Prompt | $\epsilon_{ATE}$ for Triplet Prompt |
> | ----------------- | ------------------------------------ | ----------------------------------- |
> | A, E              | 0.07                                 | **0.00**                            |
> | S, E              | 0.03                                 | **0.00**                            |
> | A, T              | 0.02                                 | **0.00**                            |
> | A, R              | 0.02                                 | **0.00**                            |
> | T, E                  |  0.04                                    |                                   **0.00**  |
>
>
>
> The result can also be understood analytically. In Proposition 3.3, we show that causal order($D_{top}$ metric) can be a suitable measure for the downstream error in causal effect estimation. Recall that from the Table 3 of main paper, for survey causal graph, the causal order obtained by pairwise prompt gets $D_{top}=3$ and triplet prompt gets $D_{top}=0$.  This difference in $D_{top}$ directly impacts the estimated causal effects as shown in the table.
>
> That said, the triplet method produces a causal order only because of the limitation of expert knowledge extraction through prompts. In practice, the triplet method can be combined with a data-based discovery algorithm (such as PC or CaMML) to obtain a causal graph and then compute the optimal backdoor set for causal effect inference. We use DoWhy libray for estimating the causal effects and linear regression as the estimator, and the sample size is 1000.
>
> **>2. In the triplet method, the authors employ a four-step process to ..... differences between the pairwise approach under this four-step method and the current approach.**
>
> **Response:** If we consider only the prompts provided to an LLM, then yes, pairwise approach can be considered as a special case of the triplet approach (with auxiliary variables being the null set).
> However, the inclusion of auxiliary variables is the key ingredient that makes the triplet _method_ substantially more accurate. Specifically, the auxiliary variable provides context for determining the relationship between a variable pair; and enables querying the LLM multiple times for the same variable pair, leading to a more robust answer. Also, selective use of GPT-4, for resolving clashes ensures further robustness for node pairs where the model might face difficulty.
> With reference to the four steps of Triplet method, the second and the third step are the key for the accuracy of the Triplet method. In comparison, for the pairwise method, the third step is absent and the second step does not provide additional context.
>
> As a result, even using smaller models such as Llama3 or Phi-3 with the Triplet method leads to better accuracy than the Pairwise method with GPT-4.

---

> > ### Author Response · Authors · 2024-11-26
> > **Requesting feedback on the rebuttal**
> >
> > Thank you again for your helpful feedback. We tried our best to address your concerns and have added additional experiments and clarifications, especially wrt. the time complexity and real-life applicability of the triplet causal order method.
> >
> > Please let us know if you have any further questions or comments. We would be happy to provide additional clarifications.

---

> > > ### Comment · Reviewer_W4GZ · 2024-11-27
> > >
> > > Thank you for your response, I have raised my score.

---

### Official Review · Reviewer_HLpf · 2024-11-02

**Soundness:** 3
**Presentation:** 4
**Contribution:** 3
**Rating:** 6
**Confidence:** 3

**Summary:**

LLMs are often used as an expert for finding causal graphs, but they can’t distinguish between direct and indirect edges. Authors proposed to use causal order as a more stable version of it, along with a triplet prompt approach.
- Their proposed approach facilitates the recovery of causal order despite imperfect experts.
- Prompt strategy that introduces auxiliary variables that introduces an auxiliary variable for every variable pair and instructs the LLM to avoid cycles within this triplet.
- Triplet prompt leads to fewer cycles than pairwise
- Causal order is a simpler structure that can still encode helpful information for down-stream tasks;

**Strengths:**

Well-written motivation and discussion of limitations;

Good discussion on the utility of the causal order, and related works.

Sound experiments with empirical evidence of the effectiveness of the proposed method.

**Weaknesses:**

Despite a sound experimental analysis, there a few details missing, i.e., how many repetitions were done in Tables 1 and 2,

One potential area not explored/mentioned is what happens with a large number of features. As the proposed method decreases the feature space, would the proposed method also facilitate the adoption of causal discovery areas that contains larger graphs?

**Questions:**

Q1: Can you clarify the experimental setup in Table 4? The title shows ‘ours - < causal discovery > + LMM’. But in line 279 it is described that “causal order is used to reduce search space of causal discovery methods”. In that sense, should it be LLM + < causal discovery > instead? As the LLM with triplet is used to reduce the search space inside the causal discovery method.

Q2: Can you describe in more details the causal effect component? How is the causal order used in that context? Is it used to drop variables (aka, no informative features or down-stream features)? How are the counterfactuals estimated?

Q3: Another ablation study that would be helpful is to see the robustness to the number of variables. Most causal discovery methods struggle to scale for larger graphs / number of features. How LLMs + triplet could be adopted to reduce search space and improve this current causal discovery bottleneck?

---

> ### Author Response · Authors · 2024-11-21
> **Response to Reviewer HLpf's comments**
>
> We thank the reviewer for their insightful comments, we try our best to incorporate their suggestions and answer their queries. We will update our final version to reflect the same as well.
>
> **Response to weaknesses**
>
> **> Despite a sound experimental analysis, there a few details missing, i.e., how many repetitions were done in Tables 1 and 2**
>
> **Response**: Each LLM based experiment (Table 2)  was run three times, and the average of the final score was reported.
> Similarly, for the human-based graph construction (Table 1),  for each dataset, three human annotators were asked to annotate the final graph and the aggregate of that was reported. Each annotator was randomly allotted a graph for both pairwise and triplet query strategies while ensuring no annotator got the same graph to query with both strategies. To get an estimate of the upper bound of human performance, for resolving tie-breaking conflicts in the triplet method, we used a ground truth-based oracle (proxy for a human domain expert). We will make sure to add this detail in the final version of the paper.
>
> **Response to questions**
>
> **> Can you clarify the experimental setup in Table 4? .... search space inside the causal discovery method.**
>
> **Response:** We thank the reviewer for this point. We agree, LLM +  < causal discovery > is a more appropriate title for our CamML based hybrid method, since we use LLM triplet's output to reduce the discovery algorithm’s search space for getting to the most optimal causal graph. However for PC hybrid approach, we use LLM triplet's output to identify the most optimal graph from the MEC obtained from PC, therefore < causal discovery > + LLM might be more suitable for this case (that said, we also propose an LLM+PC algorithm at the end of this response that may be interesting!). We will make updates in the paper to make this more clear.
>
> **> Can you describe in more details the causal effect component? .... are the counterfactuals estimated?**
>
> **Response:** Following Proposition 3.2, we use all the variables that precede the treatment variable in the estimated topological order as the adjustment set. This set qualifies as a valid backdoor adjustment set. Once the adjustment set is identified, the causal effect is estimated using the DoWhy library and linear regeression as the estimator. Appendix Table A14 presents an analysis comparing the causal effects estimated using this approach to those obtained by adjusting for the (minimal) backdoor set in the Asia dataset. The results show almost no differences in the estimated causal effects.
>
> **> Another ablation study that would be helpful is to ... improve this current causal discovery bottleneck?**
>
> **Response:** Focusing on scalability for larger graphs would be a great direction of research---thanks for suggesting this. Below we discuss how incorporating LLM triplet-based order in causal discovery algorithms can reduce the search space over graphs.
>
> **LLM Triplet + Score-based methods**: In the paper, we presented how causal order from the Triplet method can be used to provide a level order prior for the CaMML algorithm. In general, score-based methods sample different graphs, evaluate a score function for each graph, and iteratively select the graph with the best score value. Causal order from LLMs can be used to significantly decrease this search space. For instance, we can pre-compute the "forbidden" edges that violate the causal order (as a hard constraint); consequently graphs with those edges are no longer explored by a score-based discovery algorithm.
>
> For example, the GES score-based algorithm runs in two phases, forward and backward, and uses a score function to greedily search for the best graph. In the forward phase, it starts with an empty graph. Next, it adds a single edge to the graph among all possible edges that could be added that maximizes the score of the new graph. This process is repeated multiple times until no additional edges can be added.
> To reduce the search space, we propose our modification:  Instead of scoring every possible edge addition, the algorithm can rule out those edges that violate the LLM Triplet order (this can be precomputed), thus helping reduce the search space at each iteration.
>
> **LLM Triplet + Constraint-based methods**: Constraint-based methods such as the PC algorithm depend on conditional independence tests. The PC algorithm starts with complete undirected graph and then for every pair of nodes connected by an edge, it removes the edge if the two nodes are independent or conditionally independent given other subsets of nodes. In the second stage, edges are oriented. The main computational burden in the PC algorithm is the skeleton building. However, as the skeleton is undirected, it is unclear how causal order-based constraints can help in this step. Therefore, in this work, once the skeleton is obtained, we propose using the Triplet method to help orient the edges (Sec. 3 (line 282)).
>
> _(continued...)_

---

> > ### Author Response · Authors · 2024-11-22
> > **Continuation of Response to Reviewer HLpf**
> >
> > _(continued...)_
> >
> > Note that while we discussed a hard constraint based on the triplet method above, probabilistic variants of the order constraints which are weighted based on confidence of the Triplet method can also be developed. A key benefit of the triplet method is that it can provide a measure of LLM's uncertainty (for each variable pair $<A, B>$, fraction of triplets that predict $A$ causes $B$, $B$ causes $A$, or no relationship) that can be useful for weighting the LLM-based prior for discovery algorithms.

---

> > > ### Author Response · Authors · 2024-11-26
> > > **Requesting feedback on the rebuttal**
> > >
> > > Thank you again for your helpful feedback. We tried our best to address your questions and have included additional clarifications. We also discuss how LLM-based Triplet method can help reduce the search space for discovery algorithms.
> > >
> > > Please let us know if you have any questions or comments. We would be happy to engage further.

---

> > > > ### Comment · Reviewer_HLpf · 2024-12-01
> > > > **Thanks for your feedback**
> > > >
> > > > Thanks for the author's response. I choose to keep my score.

---

### Official Review · Reviewer_5DRM · 2024-11-03

**Soundness:** 3
**Presentation:** 2
**Contribution:** 3
**Rating:** 8
**Confidence:** 4

**Summary:**

The authors aimed to find a better interface to use domain knowledge for causal discovery, including LLM, but not restricted. For LLM, this is done via a pairwise prompting strategy. The authors found that expert knowledge often falls short in distinguishing direct and indirect causal relations between pairwise variables, correspondingly many cycles in causal graphs, but their causal order is relatively precise. Given this finding, the authors suggest using expert knowledge of causal order instead of causal relation. While integrating the strategy of using causal order and LLM prompting, the authors propose to introduce an auxiliary variable to improve the conventional pairwise prompting strategy by avoiding cycles within the triplet. Experimental results support that the proposed prompting methods increase the robustness and performance.

**Strengths:**

- The paper is well-written. Motivation, method, and experiments are logically straightforward.
- The attempts to utilize LLM for causal discovery have identified certain problems, such as the difficulty in distinguishing between direct and indirect causal relationships, which consequently lead to cycles in the causal graph. The authors convincingly argue that these issues are inherent to the pairwise prompting approach.
- The authors propose using causal order to leverage domain expert knowledge in causal discovery without encountering the issues associated with pairwise prompting. Additionally, it presents a novel prompting strategy to support this approach.
- When the proposed method was utilized, performance improvement was consistently observed across experiments. Additionally, the experimental design accommodates node size scaling from small to large datasets, allowing for a comprehensive understanding of how the method is affected by node size.

**Weaknesses:**

Overall, I found this work very interesting, though a few questions remain unanswered. Addressing these issues could potentially lead to a higher score.

- It is hard to determine whether triplet prompting is effective for high-performance LLMs (such as GPT-4, Claude3-opus). It appears that the prompting strategy and LLM can be applied orthogonally, so it is weird the corresponding results are omitted. Could you add results for Triplet prompting + high-performance LLMs in Table 2, to clarify this matter?
- In the experimental design related to the downstream causal discovery algorithm, it is difficult to determine how the proposed method's advantages change with varying observations. Could you add corresponding experiments with varying sample sizes to clarify this matter?

**Questions:**

- While domain expert knowledge is independent of the number of observations, it significantly affects the causal discovery algorithm. Therefore, it raises curiosity about how much performance improvement the proposed method can achieve in situations of data scarcity. Could you conduct an experiment with a very small dataset as a showcase for a data scarcity regime? It would be very interesting.

---

> ### Author Response · Authors · 2024-11-21
> **Response to Reviewer 5DRM's comments**
>
> We thank the reviewer for their insightful comments, we have tried our best to incorporate their suggestions and answer their queries.
>
> **Reply to weaknesses:**
>
> **> 1. It is hard to determine whether triplet prompting is effective for high-performance LLMs ..... LLMs in Table 2, to clarify this matter?**
>
> **Response:**
> We thank the reviewer for this point. We did not run Triplet with GPT-4 due to efficiency reasons.  We have now run the experiments with GPT-4 as expert for orienting subgraphs and then re-using GPT-4 for resolving clashes during merging phase. Following are the results for graph discovery on Asia, Alzheimers and Child graphs. Upgrading to a superior model (GPT-4) leads to better results for all three graphs.
>
>
>
> | Dataset       | Metric     | Triplet GPT-4       | Triplet GPT-3.5-Turbo     |
> |---------------|------------|----------------------|--------------------|
> | **Asia**      | $D_{top}$ | 0                    | 1                  |
> |               | SHD        | 10                   | 14                  |
> |               | Cycles     | 0                    | 0                  |
> |               | IN/TN      | 0/8                  | 0/8                |
> | **Alzheimers**| $D_{top}$ | 4                    | 4                  |
> |               | SHD        | 23                   | 28                 |
> |               | Cycles     | 0                    | 0                  |
> |               | IN/TN      | 0/11                 | 0/11                |
> | **Child**     | $D_{top}$ | 1                    | 1                  |
> |               | SHD        | 24                   | 28                 |
> |               | Cycles     | 0                    | 0                  |
> |               | IN/TN      | 6/20                 | 10/20               |
>
> **> 2. In the experimental design related to the downstream ..... experiment with a very small dataset as a showcase for a data scarcity regime? It would be very interesting.**
>
> **Response:**
> We thank the reviewer for their insightful question. As shown in Table 4 (main paper) and Table A3 (appendix), we analyzed the impact of observational dataset sizes on performance, ranging from data-scarce settings (250 instances) to data-rich ones (up to 10,000 instances) covering varying sample sizes (N=250, 500, 5000, 10000). Our results reveal that incorporating causal order using the triplet method consistently improves causal discovery, regardless of dataset size. These results are across graphs of varying sizes (4–5 nodes to 24+ nodes), including domain-specific contexts like healthcare.
>
> Based on your suggestion, we also ran an experiment for even more data-scarce setting, N = 100. We observe that incorporating triplet prior has a stronger positive impact on graph discovery performance for data scarce settings. In particular, for the medium-size graphs,  at N=100, $D_{top}$ for PC reduces from 6.3 to 2.3 for Child, and $D_{top}$ for CaMML reduces from 12.5 to 5, a significant improvement.
>
> N=100
>
> | Dataset | PC | SCORE | ICA Lingam | Direct Lingam | NOTEARS | CaMML | PC+LLM | CaMML+LLM | PC+Human | CaMML+Human |
> |----------|----------|----------|----------|----------|----------|----------|----------|----------|-----------|-----------|
> | Earthquake    | $0.5\pm 0.5$      | $4.00\pm 0.00$     | $0.66\pm0.94$     | $0.00\pm0.00$     | $1.33\pm0.47$     | $2.00\pm0.00$     | $0.00\pm 0.00$     | $0.00\pm 0.00$     | $0.00\pm 0.00$      | $1.00\pm0.00$      |
> | Cancer    | $0.00\pm 0.00$      | $2.66 \pm 0.47$     | $1.33\pm0.47$     | $2.00\pm0.00$     | $1.66\pm0.47$     | $3.00\pm0.00$     | $0.00\pm 0.00$     | $0.33\pm0.00$     | $0.0\pm 0.0$      | $0.00\pm0.00$      |
> | Asia    | $1.75\pm1.25$     | $6.33\pm0.47$     | $3.00\pm0.81$     | $0.66\pm0.94$     | $3.33\pm0.47$     | $2.33 \pm 0.14$     | $0.33\pm0.57$     | $0.97\pm0.62$     | N/A      | N/A       |
> | Asia-M    | $1.00\pm0.00$     | $6.00\pm0.00$     | $0.66\pm0.47$     | $2.33\pm1.69$     | $3.00\pm0.81$     | $1.55 \pm 0.00$     | $0.00\pm0.00$     | $1.00\pm0.00$     | $1.00\pm 0.00$      | $2.00\pm0.00$      |
> | Child    | $6.33 \pm 0.86$     | $13.33 \pm 1.24$     | $13.66\pm1.24$     | $15.33\pm1.24$     | $14.33\pm0.47$     | $3.00 \pm 0.00$     | $2.33 \pm 1.15$     | $4.00\pm0.00$     | N/A      | N/A       |
> | Neuropathic    | $1.00\pm0.00$     | $6.00\pm0.00$     | $13.33\pm2.81$     | $13.33\pm1.54$     | $9.66\pm0.00$     | $12.50\pm0.00$     | $1.00\pm0.00$     | $5.00\pm0.00$     | N/A      | N/A       |
>
> We plan to extend the results in paper to show how LLM priors help in data scarce settings for causal discovery.

---

> > ### Author Response · Authors · 2024-11-25
> > **Requesting feedback on the rebuttal**
> >
> > Thank you again for your helpful feedback. We tried our best to address your concerns and have added additional experiments with GPT-4 and with data-scarce settings.
> >
> > Please let us know if you have any further questions or comments. We would be happy to provide additional clarifications.

---

> > > ### Comment · Reviewer_5DRM · 2024-11-26
> > >
> > > Thanks for the author's response. I appreciate the efforts of the authors' for the additional experiments.
> > >
> > > - Regarding weakness 1, my concern is partially resolved, for those experimental results seem not to be completely conclusive. The chosen pool of graphs for the experiment is not representative. In addition, metrics other than SHD (even the most highlighted metric, $D_{top}$) are almost identical between GPT-4 and GPT-3.5, so they seem not to provide enough information. However, I understand that the difference in general performance between GPT-4 and GPT -3.5 is not necessarily proven through single experiments. I expect future works might analyze this point more appropreiately.
> > > - Regarding weakness 2, it seems that the authors tried to directly address my concern, but the chosen benchmark datasets are not appropriate for the analysis; there is no transparent trend along the change of the number of observations (N=100, 250, 10000).
> > >
> > > Overall, I think the benchmark datasets are not optimal for analyzing the proposed method.
> > > The benchmarks are not challenging enough for the proposed method, so in many cases, it results in performance near the upper bound, 0.00. This does not harm the novelty of the proposed method itself but inhibits the comprehensive understanding and highlighting of the characteristic behavior of the LLM integrated framework.
> > > Taking these points together, I choose to maintain the current score.

---

> > > > ### Author Response · Authors · 2024-12-01
> > > > **Follow up to Reviewer 5DRM's points**
> > > >
> > > > Thanks again for your response and feedback on our work.
> > > >
> > > >
> > > >
> > > > **Weakness 1:**
> > > >
> > > > For the results above, we chose the same datasets as in Table 2.
> > > >
> > > > While using GPT-4 with Triplet provides slightly improved performance compared to using GPT-3.5, the main result is that under both LLMs (GPT-3.5 and 4), Triplet method leads to substantially better results than the pairwise method (see expanded table below).
> > > >
> > > > As noted in Table 2, another benefit of Triplet is that even when using it with smaller models such as Phi3 and Llama3, it leads to better graph metrics than the pairwise method with GPT-4. This is an important advantage of the Triplet method since using these smaller models can lead to substantial efficiency gains.
> > > >
> > > > For completeness, below we provide Table 2 with the updated results including Triplet (GPT-4); with bolded best metrics per LLM model (GPT-3.5 and 4).
> > > >
> > > >
> > > >
> > > > | Dataset       | Metric    | A:Pairwise GPT-3.5-Turbo | A: Triplet GPT-3.5-Turbo | B: Pairwise GPT-4 | B: Triplet GPT-4 | Triplet Phi-3 | Triplet Llama3 |
> > > > |---------------|-----------|------------------------|-----------------------|----------------|---------------|---------------|----------------|
> > > > | **Asia**      | D_top     | -                      | **1**                     | 1              | **0**             | 0             | 2              |
> > > > |               | SHD       | 21                     | **14**                    | 18             | **10**            | 13            | 17             |
> > > > |               | Cycles    | 1                      | **0**                     | **0**              | **0**             | 0             | 0              |
> > > > |               | IN/TN     | **0/8**                    | **0/8**                   | **0/8**            | **0/8**           | 1/8           | 0/8            |
> > > > | **Alzheimers**| D_top     | -                      | **4**                     | -              | **4**             | 7             | 5              |
> > > > |               | SHD       | 42                     | **28**                    | 30              | **23**            | 25             |   24            |
> > > > |               | Cycles    | 684                    | **0**                     | 1              | **0**             | 0             | 0              |
> > > > |               | IN/TN     | **0/11**                   | **0/11**                  | **0/11**           | **0/11**          | 0/11          | 0/11           |
> > > > | **Child**     | D_top     | -                      | **1**                     | -              | **1**             | 17            | 12             |
> > > > |               | SHD       | 177                    | **28**                    | 148            | **24**            | 69            | 129            |
> > > > |               | Cycles    | >10k                   | 0                     | >10k           | 0             | 0             | 0              |
> > > > |               | IN/TN     | **0/20**                   | 10/20                 | **0/20**           | 6/20          | 0/20          | 0/20           |
> > > >
> > > >
> > > >
> > > >
> > > >
> > > > **Weakness 2**:
> > > >
> > > > Overall, Table 4 and the results for N=100 (above) show that adding LLM Triplet results helps improve accuracy of discovery algorithms. As noted above, for N=100, the impact is significant for medium-sized graphs such as Child (incorporating the Triplet output reduces $D_{top}$ of PC from 6.33 to 2.33) and Neuropathic (incorporating the Triplet output reduces $D_{top}$ of CaMML from 12.5 to 5). That said, we agree that extending to additional complex graphs may provide more comprehensive understanding wrt. sample size and will be useful as future work.

---

### Official Review · Reviewer_WUxn · 2024-11-04

**Soundness:** 4
**Presentation:** 3
**Contribution:** 3
**Rating:** 8
**Confidence:** 4

**Summary:**

This paper considers optimal ways of querying imperfect experts (e.g., LLMs) when the aim is to discover a causal DAG over a set of variables. The key proposed idea is to query imperfect experts about the causal ordering over variables instead of graphical relationships like edges. The paper shows that certain ways of orienting edges based on queries to experts can lead to errors in the overall DAG, while the causal ordering from a perfect expert will never be incorrect. The paper proposes ways of integrating causal ordering into known causal discovery algorithms, and then focuses on extracting orderings from imperfect experts. To minimize errors by imperfect experts, the paper proposes querying for the order over triplets of variables while enforcing acyclicity, proving that this makes fewer errors than querying about pairs of variables (under some assumptions). The work then studies this querying strategy empirically using a variety of hand-curated known causal DAGs. They study both human and LLM imperfect experts, and find evidence that the triplet querying strategy improves causal discovery and effect estimation performance over both the pairwise querying strategy and using no experts at all.

**Strengths:**

+ The technical as well as written clarity of the paper is excellent. Technically, the notation is clear, all preliminaries and results are well-defined and well-explained. The paper is also written in a way that is easy to read and follow.

+ The technical result about the optimality of the triplet querying strategy over the pairwise querying strategy is novel, to the best of my knowledge, and could have an impact on causal discovery algorithms more broadly.

+ The empirical studies are comprehensive, and the initiative to contrast both humans and LLM imperfect experts is impressive since studies with humans are time-consuming and costly.

**Weaknesses:**

+ In the broader context of causal discovery, I question the idea that LLMs are $\epsilon$-imperfect experts. Existing results including those in this work consistently consider causal discovery problems that involve standard medical knowledge, or general knowledge about the world that's fairly well-established by now. However, causal discovery as a field ought to be about discovering new knowledge, in domains where we have very only faint hypotheses about variables relate, e.g., consider abstract variables that represent aspects of human behavior like "trust" or "aggression." In these settings, we should expect $\epsilon$ to be so large as to render LLM experts unreliable. I'm curious to see the authors better justify and contextualize the role of LLMs in genuinely challenging causal discovery problems.

+ The premise in this paper is that querying experts with questions of the form "does A cause B" and directing the edge A->B if the expert says yes leads to incorrect graphs. However, the query "does A cause B" isn't by itself inherently flawed -- it's already a question about causal ordering -- the policy to orient edges based on ancestral causal relations is what's flawed. Nevertheless, the introduction and parts of section 3 (e.g., definition 3.1) seem to critique the query itself, when really, the issue is that responses to this query should not be used to directly orient edges (e.g., the perfect expert in 3.1 is defined based on a flawed policy). Therefore, the argumentation strategy in the paper feels like setting up a strawman argument. Note, though, that this is just a critique about the storytelling; the technical results are valid and make sense regardless. One could pivot the story slightly to argue that the flaw isn't with the "does A cause B" style of query, but in ensuring that this is interpreted only as a causal ordering.

+ There is a bit of extra context that I would have liked in the empirical studies: 1) can you contextualize the datasets and graphs that are studied further and discuss whether they are semi-synthetic, purely realistic, etc.? 2) Despite the results in Figure 3 on synthetic data, it would be useful in the broader empirical studies to compare the causal ordering queries to the flawed policy of orienting edges based no responses to "does A cause B"-style questions. (I apologize if this comparison is indeed there and I missed it -- if that ended up being the case, please point me to the correct figure or table in the empirical studies.)

**Questions:**

+ Why do you think the premise of LLMs as $\epsilon$-imperfect experts makes sense more generally in causal discovery, given that $\epsilon$ *should* large if we're solving a truly interesting causal discovery problem?

+ Can you the details about the empirical studies that I mention above?


Edit: The authors satisfactorily addressed my questions and concerns in their response; I'm happy to thus raise my score and support their work.

---

> ### Author Response · Authors · 2024-11-21
> **Reply to Reviewer WUxn's comments**
>
> We thank the reviewer for their insightful comments, we have tried our best to incorporate their suggestions and answer their queries.
>
> **Reply to weaknesses**:
>
> **1. "In the broader context of causal discovery ..... causal discovery problems."**
>
> **Response**: We agree that in truly novel situations, $\epsilon$ error of LLMs in causal discovery can be high. As in previous work on LLM-based discovery, we instead focus our attention on known relationships and LLM's capabilities on extracting them. This can be practically useful; in real-world problems, building a full causal graph often involves a combination of novel and previously known relationships, and we hope that LLMs can save significant effort in extracting the known relationships.
>
> For the genuinely challenging causal discovery problems, we think a combination of LLM and data-based algorithms is more suited. That is why we proposed two variants that combine LLMs with data-based algorithms such as PC and CaMML (for example, LLMs may be used for providing a prior on the known relationships, which may help algorithms to learn the remaining novel relationships). In a similar direction, a key benefit of the triplet prompt is that it can also provide a measure of LLM's uncertainty (for each variable pair $<A, B>$, fraction of triplets that predict A causes B, B causes A, or no relationship) that can be useful for weighting the LLM-based prior for data-based algorithms.
>
> **2. "The premise in this paper is that querying experts with questions ......this is interpreted only as a causal ordering."**
>
> **Response**: That's a great point. Existing LLM-based methods (Kiciman et al. 2023, Long et al. 2022, and others) used pairwise queries to infer an edge (which motivated our story), but we agree that it is not a necessary feature of a pairwise method. We will refine our introduction and Section 3 to reflect this.
>
> **3. "There is a bit of extra context that I would have liked in the empirical studies:
> > 1) can you contextualize the datasets and graphs that are studied further and discuss whether they are semi-synthetic, purely realistic, etc.?"**
>
> **Response**: All graphs are real-world graphs constructed by human experts. The data is generated using different methods. We summarize the datasets below (also see Table A15).
>
> | Dataset                                                      | Graph                                                                                                                                                                                                | Data for Variables                                  |
> |--------------------------------------------------------------|------------------------------------------------------------------------------------------------------------------------------------------------------------------------------------------------------|-----------------------------------------------------|
> | BN Learn Datasets (Asia, Cancer, Earthquake, Survey, Child)  | Real-world graphs from scientific studies                                                                                                                                                            | Synthetic data generation based on bnlearn library  |
> | Neuropathic Pain                                             | Real-world graph constructed with consensus from medical experts (Tue et al. 2019). Includes domain-specific variables as Right L1 Radiculopathy, Toracal Dysfunction, DLS L5-S1, etc. (see Fig. A8) | Synthetic data generation based on Tu et al. (2019) |
> | Alzheimers Dataset                                           | Real-world graph constructed with consensus from medical experts (Abadullah et al. 2023). Constructed in 2023, after the training cutoff date of GPT-3.5 and GPT-4 models used.                     | No data is available                                |
> | Covid-19 Respiratory Dataset                                 | Real-world graph constructed by experts to understand effect of Covid-19 on respiratory system (Mascaro et al. 2022). Constructed in 2022, after the training cutoff date of GPT-3.5 and GPT-4 models used.                | No data is available.                               |

---

> > ### Author Response · Authors · 2024-11-21
> > **Continuation of the reply to reviewer WUxn's comments**
> >
> > **> 2) Despite the results in Figure 3 on synthetic data, it would be useful in the broader empirical studies to compare the causal ordering queries to the flawed policy of orienting edges based no responses to "does A cause B"-style questions.**
> >
> > **Response**: Thanks for this suggestion. In addition to the _Pairwise (Edge)_ method, we developed a order variant of the pairwise method: Rather than using a pairwise query's answer to infer an edge, we infer a partial/relative order for each pair of nodes. We then aggregate all pairwise orders to get the final graph order. However, as in the case of _Pairwise (Edge)_ method where cycles were obtained for many graph datasets (such as Asia, Alzheimers), as we progress to build a final aggregated causal order for the full graph, we find pairwise responses that violate causal order (acyclicity) consistency for the same datasets. As a result, we get cycles in the obtained causal order and we could not proceed further to compare _Pairwise (Order)_ to the Triplet-based method.  We conclude that similar to edge orientations for pairwise level, inferring order based on pairwise queries is also susceptible to erroneous and cyclic structures.

---

> > > ### Author Response · Authors · 2024-11-25
> > > **Requesting feedback on the rebuttal**
> > >
> > > Thank you again for your helpful feedback. We tried our best to address your concerns with additional experiments and details about the benchmark datasets.
> > >
> > > Please let us know if you have any further questions or comments. We would be happy to provide additional clarifications.

---

> > > > ### Comment · Reviewer_WUxn · 2024-11-26
> > > > **Raised score**
> > > >
> > > > Dear authors, thanks for engaging and providing responses to my questions and concerns. Given that you've addressed them to a good extent, I went ahead and raised my score in support of your work.

---

### Author Response · Authors · 2024-12-04
**Rebuttal Summary**

We sincerely appreciate all the reviewers for their time, insightful comments and positive feedback. Reviewers appreciated the technical and writing clarity, novelty of our contribution on inferring causal order, and the comprehensive experiments including both LLMs and humans as imperfect experts.

Reviewers' suggestions have significantly contributed to improving our work, and we thank them for increasing their scores based on our response and additional experiments. The majority of the suggestions pertain to clarifications of our approach and ablations to provide a better understanding of the proposed Triplet method and its effectiveness (for example comparison with recent LLM based methods, or how causal order from our pipeline would serve as an effective prior for tasks like causal effect estimation). These ablations have further strengthened the impact of our approach.

1. Additional results of our triplet method using a stronger model such as GPT-4 (in response to reviewer 5DRM) complete a comprehensive comparison across a spectrum of models---from smaller ones like Phi3 and Llama3 to larger ones like GPT-4---wrt. the baseline pairwise approach. Across all models, we observe a consistent trend that the triplet method outperforms existing pairwise-based methods for causal discovery. Moreover, as reported in Table 2 of the paper, triplet method using smaller models such as Phi3 and Llama3 obtains better accuracy than the pairwise method with GPT-4.

2. We highlight how the triplet method yields computational efficiency while ensuring high accuracy for causal discovery, as answered to reviewer W4GZ. As requested, we have also implemented additional baselines from recent work (LLM-BFS, LLM-BFS+Stats) and find that the triplet method outperforms both methods.

3. We have provided clarifications on the practical utility of causal order for downstream tasks such as graph discovery and effect inference; and more generally, on the utility of LLM-based methods for causal discovery. We thank the reviewers for these questions and will add this discussion to the final paper.

We again thank the reviewers for engaging with us and helping to clarify the contributions of the work.

---

### Meta-Review · Area_Chair_Nxik · 2024-12-21

**Metareview:**

The paper introduces a strategy for querying imperfect experts (such as LLMs) for causal discovery, focusing on causal ordering over graphical relationships.


Strengths:

+ The paper proposed triplet querying strategy for causal discovery.


+ The inclusion of both human and LLM experts in experiments provides valuable insights, especially given the cost and time involved in human studies.

Weaknesses:

+ The paper lacks justification of the use of LLMs as imperfect experts, especially in challenging causal discovery tasks where LLMs might be unreliable.

+ The critique of the "does A cause B" question feels misplaced, as the issue lies more in how edge orientation is determined, not the query itself.

+ Some key details in the empirical studies, such as the dataset context and variations in experimental setups, are missing or unclear, limiting the robustness of the findings.

**Additional Comments On Reviewer Discussion:**

The reviewers unanimously lean toward acceptance.

---

### Decision · Program_Chairs · 2025-01-22

Accept (Poster)